# Residual Layer Ozone, Mixing, and the Nocturnal Jet in California's San Joaquin Valley

Dani J. Caputi[1], Ian Faloona[1], Justin Trousdell[1], Jeanelle Smoot[1], Nicholas Falk[1], Stephen Conley[2]

[1]Department of Land, Air, and Water Resources, University of California Davis, Davis, 95616, USA
[2]Scientific Aviation, Inc., Boulder, 80301, USA

*Correspondence to*: Dani J. Caputi (djcaputi@ucdavis.edu)

**Abstract:** The San Joaquin valley is known for excessive secondary air pollution owing to local production combined with terrain-induced flow patterns that channel air in from the highly populated San Francisco Bay area and stagnate it against the surrounding mountains. During the summer, ozone violations of the National Ambient Air Quality Standards (NAAQS) are notoriously common, with the San Joaquin Valley having an average of 115 violations of the recent 70 ppb standard each year between 2012 and 2016. The nocturnal dynamics that contribute to these summertime high ozone events have yet to be fully elucidated. Here we investigate the hypothesis that on nights with a strong low-level jet (LLJ) ozone in the residual layer is more effectively mixed down into the stable boundary layer. There it is subject to dry deposition to the surface, the rate of which is itself enhanced by the strength of the LLJ, resulting in lower ozone levels the following day. Conversely, nights with a weaker jet will sustain residual layers that are more decoupled from the surface and thus lead to more fumigation of ozone in the mornings giving rise to higher ozone concentrations the following afternoon. We analyse aircraft data from a study sponsored by the California Air Resources Board (CARB) aimed at quantifying the role of residual layer ozone in the high ozone episode events in this area. By formulating nocturnal scalar budgets based on flights around midnight and just after sunrise the following days, we estimate the rate of vertical mixing between the residual layer (RL) and the nocturnal boundary layer (NBL), and thereby infer eddy diffusion coefficients in the top half of the NBL. The average depth of the NBL observed on the 12 pairs of flights was 210 ($\pm$ 50) m. Of the average -1.3 ppb h$^{-1}$ loss of the $O_x$ family (here $[O_x] \equiv [O_3] + [NO_2]$) in the NBL during the overnight hours from midnight to 06:00 PST, -0.2 ppb h$^{-1}$ was found to be due to horizontal advection, -1.2 ppb h$^{-1}$ due to dry deposition, -2.7 ppb h$^{-1}$ to chemical loss via nitrate production, and +2.8 ppb h$^{-1}$ from mixing into the NBL from the residual layer overnight. Based on the observed gradients of $O_x$ in the top half of the NBL these mixing rates yield eddy diffusivity estimates ranging from $1.1 - 3.5$ m$^2$ s$^{-1}$ that are found to inversely correlate with the following afternoon's ozone levels, and provide support for our hypothesis. The diffusivity values are approximately an order of magnitude larger than the few others reported in the extant literature for the NBL, which further suggests that the vigorous nature of nocturnal mixing in this region due to the LLJ has an important control on ozone. Additionally, we investigate the synoptic conditions that are associated with strong nocturnal jets and find that on average, deeper troughs along the California coastline are associated with stronger jets. The LLJ had an average height of 340 m, an average speed of 9.9 m s$^{-1}$ (SD = 3.1 m s$^{-1}$) and a typical peak timing around 23:00 PST. Seven years of 915 MHz radio-acoustic sounding system and surface air quality network data show an inverse correlation between the jet strength and ozone the following day, suggesting that air quality models need to forecast the strength of this nocturnal feature in order to more accurately predict ozone violations.

## 1. Introduction

Under typical fair weather conditions over the continents, thermals are generated near the surface beginning shortly after sunrise, forcing a convectively mixed layer, known more generally as the convective atmospheric boundary layer. As solar heating of the Earth's surface increases throughout the day, this layer reaches its maximum height by late afternoon, typically between 700 and 900 m in California's central valley during summer months (Bianco et al., 2011; Trousdell et al., in preparation). Around sunset, when the solar heating of the surface ends, the convective thermals are cut off and can no longer power turbulent mixing in the boundary layer. The result of the radiative cooling of the ground throughout the night forms a stable, nocturnal boundary layer (NBL) near the surface, typically extending between 100 and 500 m (Stull, 1988). The convective layer from the daytime, after spinning down and no longer actively mixing, functions as a residual reservoir for pollutants and other trace gases from daytime emissions and photochemical production. This layer overlying the NBL is known as the residual layer (RL).

During both daytime and nighttime, mixing can occur between the boundary layer and the layer of air above. In the daytime, this process of entrainment is driven by convective thermals that penetrate into the laminar free troposphere, which then sink back into the convective layer, and may be augmented by wind shear near the top of the boundary layer (Conzemius and Fedorovich, 2006). Entrainment has been shown to be a significant factor for surface pollution, and more generally scalar budgets, as the two interacting layers usually have different trace gas concentrations

(Lehning et al., 1998; Trousdell et al., 2016; Vilà-Guerau de Arellano et al., 2011). At night, another type of gas exchange can occur between the aforementioned stable boundary layer and the residual layer by shear-induced mixing. Extensive observations of the structure of the NBL indicate that a localized wind maximum near the top of the NBL, known as a low level jet (LLJ), is often present (Banta et al., 2002; Garratt, 1985; Kraus et al., 1985). This low level jet is able to drive sheer production of turbulence in an intermittent, cyclical manner, powering the mixing between these layers.

California's complex terrain amplifies the challenge of both studying and managing air pollution in this area. The main source of air for California's Southern San Joaquin Valley (SSJV) is incoming maritime flow from the San Francisco Bay area, which gets accelerated toward the southern end of the valley as a consequence of the valley-mountain circulation (Rampanelli et al., 2004; Schmidli and Rottuno, 2010). The local sources of ozone are scattered along this primary inflow path to the SSJV. The ozone buildup in the SSJV results from both the large amount of local upwind sources and the Tehachapi Mountains to the south which block the flow, preventing advection out of the region (Dabdub et al., 1999; Pun et al., 2000). Because of this tendency for the air to stagnate, both daytime and nocturnal vertical mixing are likely important in the phenomenology of ozone pollution in this area.

The complex nocturnal wind patterns in the SSJV contribute to the challenges of understanding and forecasting ozone pollution in our study region. The LLJ in the SSJV is known to contribute to the formation of a commonly observed late night and early morning mesoscale wind feature known as the Fresno Eddy, which can drive both vertical mixing and regional horizontal advection. The aforementioned daytime northwesterly valley wind continues into the late evening, decoupling from the surface and forming a LLJ (Davies 2000). The Tehachapi Mountains act as a barrier to the jet if the Froude number is lower than about 0.2 (Lin and Jao, 1995). The eddy feature is formed during the hours before dawn when this northwesterly flow interacts with southeasterly nocturnal downslope flow coming from the high southern Sierra Nevada Mountains, although there is some question as to the extent to which the southeasterly flow observed in the morning hours is merely the result of a topographic deflection and recirculation of the nocturnal jet. The Coriolis force helps to circulate this flow; however, a mesoscale low is not thought to develop (Bao et al. 2007, Lin and Jao, 1995). It is worth noting that the valley flow peaks shortly after sunset, while the katabatic drainage flow peaks shortly before dawn, so these two components of the Fresno eddy are not time-coherent. The initial northwesterly wind and a low Froude number are both critical for determining whether or not the eddy will form on a given night (Lin and Jao, 1995). Monthly averaged wind speeds from June through August of the low level jet near the Fresno eddy up to 12 m s$^{-1}$ have been reported (Bianco et al., 2011). Beaver and Palazoglu (2009) found that ozone pollution in the central San Joaquin Valley is particularly high on days where the preceding nocturnal Fresno Eddy is strong, even when strong ventilation is occurring. They also found that the early morning downslope flow through the Tehachapi pass is a strong predictor of ozone pollution in the SSJV. However, mixing induced by nocturnal jets has been shown to decrease ozone levels the following day in other parts of the world (Hu et al., 2013; Neu et al., 1995), so one might suspect that a Fresno eddy that creates a particularly strong LLJ on a given night may decrease ozone the following day if the recirculation of ozone does not compensate for its loss due to vertical mixing.

It has been previously shown that residual layer ozone can have a substantial correlation with ground-level ozone the following day (Aneja et al. 2000; Zhang and Rao, 1999). Neu et al. (1995) estimated that about 75 % of the contribution

to the difference in afternoon ozone concentrations from one day to the next is from residual layer depletion. This study was done in complex terrain of Switzerland and primarily used SODAR data. They also found a strong correlation ($r^2 = 0.74$) between weaker turbulence in the RL, inferred from the amount of time the wind maximum at night was below 150 m, and the aforementioned ozone difference. Coupling of the RL and NBL via intermittent turbulence has also been shown to correlate with ozone spikes at ground-level monitoring stations (Salmond and McKendry, 2005). Because of the complexity of intermittent nocturnal turbulence, the spatial and temporal distributions of these spikes are unknown, and thus it is not known the extent to which these ozone spikes help to deplete the residual layer ozone or contribute to the following day's ozone. One advantage of our study is that we are using airborne data to sample a large area, which overcomes the limitations of studies using ground monitoring stations that may be influenced by the intermittent bursts of turbulence and confounded by uncertain horizontal advection.

A study from Southern Taiwan also found that residual layer ozone plays an important role in the following day's ozone concentrations, with fumigation of this ozone into the developing daytime boundary layer accounting for 19 % of the variance (Lin 2012). As the ozone problems in Southern Taiwan are not heavily driven by local sources, a more extensively mixed daytime boundary layer can in fact contribute to a buildup, rather than ventilation, of ozone, because the daytime intrusion into the formal residual layer outcompetes the local production. This, and the fact that many ozone forecasts currently made only take into account local daytime boundary layer dynamics, highlights the need for studying the effects of residual layer ozone in more areas that have ozone problems.

Bao et al. (2008) reports that while the Weather Research and Forecasting (WRF) model is able to qualitatively capture the LLJ, systematic errors up to 2 m s$^{-1}$ are observed, with root mean square errors of $4 - 5$ m s$^{-1}$. Above 2000 m, a similar magnitude of errors in the model's ability to forecast wind is observed, and since the LLJ is influenced by this upper level synoptic forcing, there is a need for more systematic study of the background synoptic conditions associated with strong and weak LLJ. The authors also note that apart from the 915 MHz Radio Acoustic Sounding Systems (RASS), observations of the LLJ in the SSJV are lacking in spatial coverage. This further highlights the need for an observational-based study of low level winds in the SSJV during high ozone episodes.

At the core of our observational method, we acknowledge that most scalar budgets are driven by horizontal advection, vertical mixing (primarily entrainment), and local emissions/uptake and net production (including chemical gains and/or losses). Conley et al. (2011) and Faloona et al. (2009) have shown that on any given day, advection can be a relatively large and significant term in the daytime scalar budget. However, when averaged over numerous flight days, the advection is often close to zero. Studies performing daytime scalar budgets of ozone (Conley et al., 2011; Lehning et al., 1998; Lenschow et al., 1981; Trousdell et al., 2016) have shown that chemical production is important, and similarly, we expect the chemical loss of ozone to be important at night.. The nocturnal ozone chemistry is driven primarily by its reaction with $NO_2$ forming the nitrate radical. The nitrate radical can equilibrate with $N_2O_5$, react with VOCs, and rapidly react with NO (Brown et al., 2006, 2007; Wood et al., 2004). As we will attempt to show, the fate of the nitrate radical is highly uncertain and this plays an important role in the net overnight chemical loss of ozone. Additionally, dry deposition of the chemical species of interest cannot be ignored (Conley et al., 2011; Faloona et al., 2009). While the aforementioned studies focused on daytime scalar budgets, to our knowledge, no attempts have been made at nocturnal scalar budgets using aircraft data. Our goal is to test whether more nocturnal mixing between the

residual layer and stable boundary layer, induced by wind-shear turbulence beneath a strong low level jet, will effectively "deplete" ozone in the residual layer, making less available to fumigate the following morning and seed further photochemical production. We will proceed with this in three ways: first, we introduce a method for analysing nocturnal scalar budgets of flight data, which is similar to that of the daytime scalar budgets, and attempt to estimate the eddy diffusivity of $O_x$ in the NBL on each night of the field campaign (sections 3.1 and 3.2). Second, we analyse

synoptic conditions around the LLJ, and look at a broader dataset of LLJ strength and the following afternoon's ozone concentrations (sections 3.3 and 3.4). Lastly, we look at other metrics of NBL turbulence in our campaign data such as Turbulent Kinetic Energy (TKE), Bulk Richardson Number (BRN), and elevated mixed layers in order to bolster confidence in our findings (sections 3.5 and 3.6).

## 2. Nocturnal $O_x$ Budgeting Method

### 2.1. Data Collection

Aircraft data was collected by a Mooney Bravo and Mooney Ovation, which are fixed-wing single engine airplanes operated by Scientific Aviation Inc. The wings are modified to sample air through inlets, which flow to the on-board analyzers. Temperature and relative humidity data were collected by a Visalia HMP60 Humidity and Temperature

Probe, ozone was measured with a dual beam ozone absorption monitor (2B Technologies Model 205), and NO was measured by chemiluminescence (ECO PHYSICS Model CLD 88). $NO_x$ was measured by utilizing a photolytic converter (model 42i BLC2-395 manufactured by Air Quality Design, Inc.). For flights performed in 2016, a pre-reaction chamber was also installed to monitor and subtract the changing background signal, reducing the detection threshold to < 50 ppt. Frequent calibrations were performed in the field, generally once per deployment, with zero and

span checks daily. Calibrations for NO measurements were performed with a NIST-traceable standard by Scott-Marrin, Inc. Calibrations for $NO_x$ measurements were performed by titrating the NO standard with an ozone generator (2B Technologies, Model 206 Ozone Calibration Source.) During routine operation on the aircraft, the lamp of the photolytic converter was toggled on and off at 20-second intervals during the flights (corresponding to approximately 1.5 km horizontal and 50 m vertical displacements by the aircraft), requiring linear interpolation for continuous NO

and $NO_2$ data. The pre-reaction chamber was toggled on for a 40 second period every 10 minutes in order to measure the background signals of NO and $NO_x$, and the background signals were subtracted from the measurement. The interpolated $NO_2$ signal was noted to decay approximately exponentially after powering up, which sometimes affected the first 15-30 minutes of flight. The presumed artifact was successfully replicated in the lab with a constant $NO_2$ concentration, and was removed by exponential detrending (see Trousdell et al., in preparation for a detailed

discussion.)
Winds are measured using a Duel-Hemisphere Global Positioning System combined with direct airspeed measurements, as described in Conley et al. (2014). The winds are measured at 1 Hz, and the power spectra is observed to fit the Kolmogorov Scaling Law within the inertial subrange (approximately from 0.12 - 0.5 Hz in the daytime convective boundary layer corresponding to roughly 150 – 600 m spatial scales). At night, the -5/3 slope is observed

from 0.02 – 0.5 Hz (Fig. 1), corresponding to length scales of 150 – 3700 m, the largest of which are likely contributions from buoyancy waves. This is evident by the calculated Brunt–Väisälä frequencies (Fig. 2), which have an average value of 0.023 Hz in the NBL. For simplicity sake, we consider anything smaller than this buoyancy frequency to be "turbulence", and use $1/N_{BV} \sim 50$ seconds to be the sampling time to observe wind variances, though we recognize that this cutoff is somewhat arbitrary. The turbulent kinetic energy (TKE) is estimated by correcting the

observed wind variance of a given detrended 50-second signal with the integrated nocturnal power spectra beyond the Nyquist frequency (0.5 Hz) using a -5/3 extrapolation, which indicates that approximately 11 % of the total variance is not directly captured by the system. Only horizontal winds are measured, thus similarity assumptions are required to estimate vertical wind variance ($\sigma_w^2$). While some similarity relationships have been reported for the stable boundary layer (Nieuwstadt, 1984), we were not able to measure the governing parameters. However, Banta et al.

(2006) reported a meta-analysis of stable boundary layer studies with an average $\sigma_w^2/\sigma_u^2$ of 0.39, where $\sigma_u^2$ is the streamwise variance. We applied this correction to our TKE measurements to account for the missing vertical wind variance.

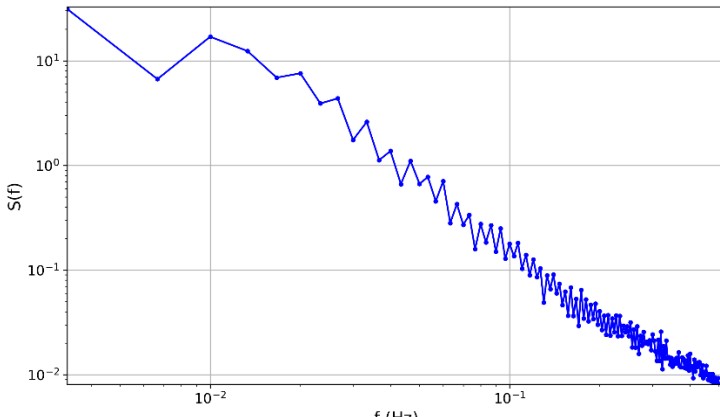

**Figure 1.** Power spectra for nighttime winds averaged over 309 5-minute samples. The average airspeed was 76.6 m
s$^{-1}$.

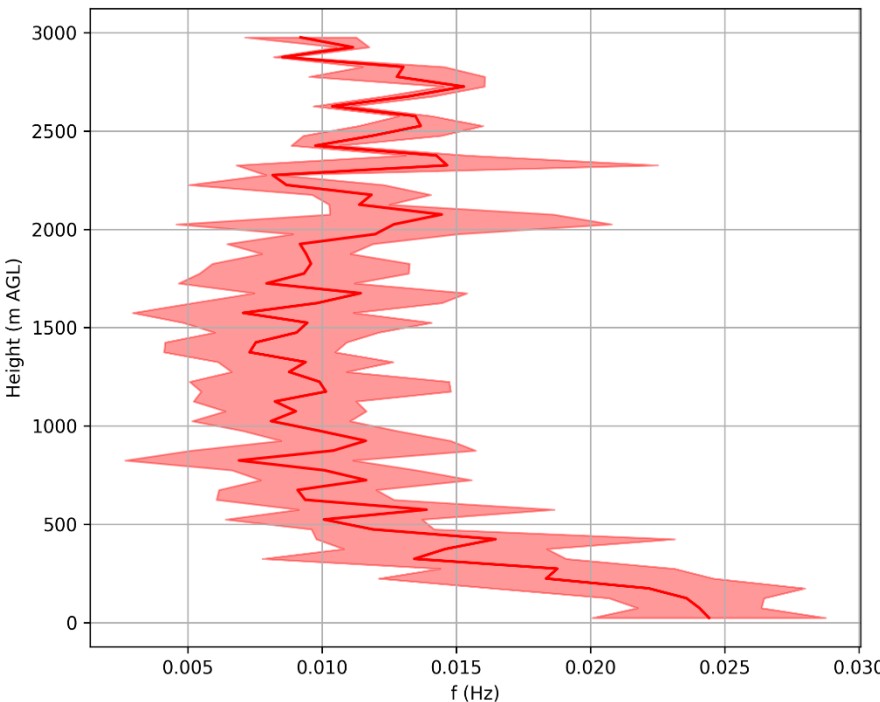

**Figure 2.** Mean and standard deviation profile of Brunt–Väisälä frequencies for all late night flights. The mean value within the stable boundary layers is 0.023 s$^{-1}$.

Data was collected on 5 separate deployments (10-12 September 2015, 2-4 June 2016, 28-29 June 2016, 24-26 July

2016, 12-18 August 2016). During a given deployment, 4 flights per day were conducted (7, 11, 15, and 22 PST). Each deployment consisted of stationing the airplane at Fresno Yosemite International Airport (FAT), with each flight comprising a transect to Bakersfield Meadows Field Airport (BFL) and back spanning approximately 2 hours and 15 minutes (Fig. 3). Profiles of the full boundary layer and above were taken at Fresno and Bakersfield. Along the Fresno-Bakersfield transect, altitude legs of 500, 1000, and 1500 m AGL were conducted in a randomized order. Low passes

were also flown over the Tulare (TLR), Delano (DLO), and Bakersfield airports, but in 2016 we replaced the low approaches at Tulare with Visalia (VIS) to coincide with the NOAA LIDAR deployment (Langford et al., submitted). All of these airports are within a few hundred meters of California Highway 99, or in the case of Fresno and Bakersfield within an urban center. If time was remaining on any given flight, we typically utilized it by either completing an extra profile at Visalia, or flying west toward Hanson to better sample the nocturnal LLJ.

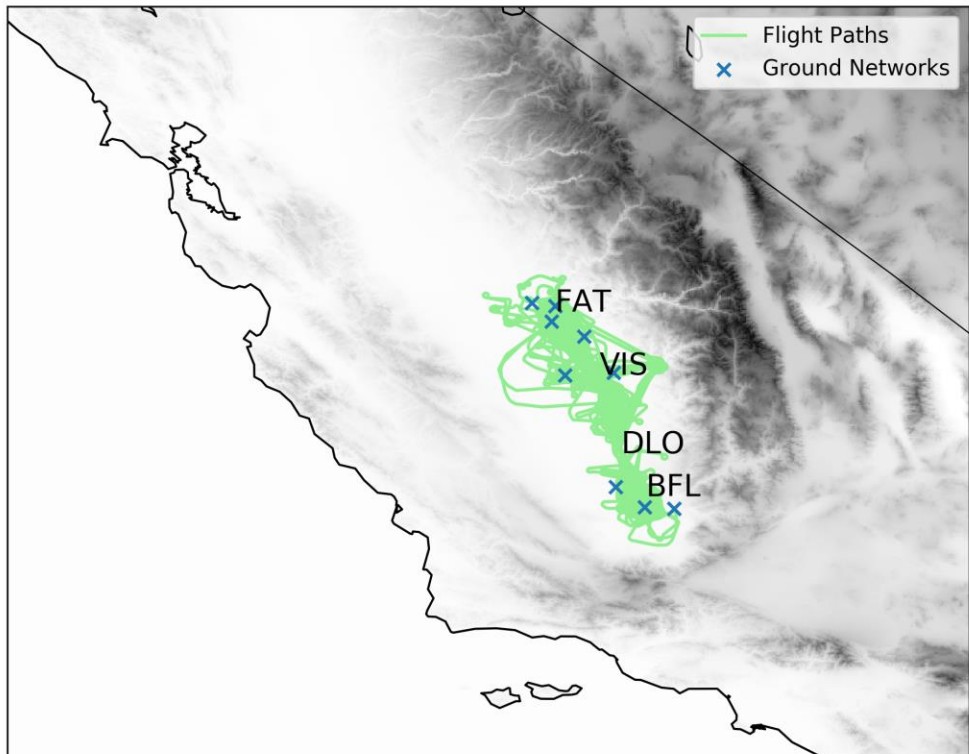


**Figure 3.** Ground tracks of all flights of the Residual Layer Ozone project. Airports with low approaches and ground ozone monitors are shown. The ground network stations (blue crosses) used were Bakersfield-5558 California Avenue, Bakersfield-Municipal Airport, Clovis-N Villa Avenue, Edison, Fresno-Drummond Street, Fresno-Garland, Fresno-Sierra Skypark #2, Hanford-S Irwin Street, Parlier, Shafter-Walker Street, and Visalia-N
Church Street.

The nocturnal scalar budget analyses presented here utilizes all late night (~ 21:45 – 00:00 PST) flights in which a subsequent flight was conducted the following morning (~ 06:15 – 08:30 PST). The dates (before midnight PST) of the late night flights for the 12 overnight periods are shown in Table 1. Additionally, late night flights without a subsequent morning flight were flown on 12 September 2015 and 26 July 2016, and morning flights without a
preceding late night flight were flown on 10 September 2015, 24 July 2016, 12 August 2016, and 14 August 2016. These additional flights are included in the analyses here that refer exclusively to either the late night or morning flights, but were not used for the scalar budgets.

**2.2. Budget Conceptual Framework**

Here we aim to test the importance of the aforementioned nocturnal mixing on the ozone budget in this region by applying a scalar budgeting technique to the aircraft data in order to estimate an eddy diffusivity between the stable boundary layer and the residual layer. This objective aims to use a similar method that has been presented with daytime scalar budgets (Conley et al., 2011; Faloona et al., 2009; Trousdell et al., 2016) to further demonstrate the overall practicality of this methodology.

The nocturnal budget equation is formulated by the Reynolds-averaged conservation equation for a scalar – in this case $O_x$ – in a turbulent medium. $O_x$ is defined here as $NO_2 + O_3$ in order to avoid the effects of titration of $O_3$ by NO. If not depleted by chemical oxidation to $NO_3$ and further reaction products, $NO_2$ will photolyze the following day to reproduce ozone in photostationary state, so it can act as an overnight reservoir of ozone. The chemical loss of $O_x$ is then computed by the reaction between $O_3$ and $NO_2$ to form nitrate, and the ultimate fate of nitrate will affect the

overall $O_x$ loss. In the stable nighttime environment we will treat the mixing between the RL and NBL by using an eddy diffusivity. The NBL $O_x$ budget can thus be represented as:

$$\frac{\partial [O_x]}{\partial t} = -\alpha k_{O3+NO2}[O_3][NO_2] - \bar{u}\frac{\overline{\Delta[O_x]}}{\Delta x} - \bar{v}\frac{\overline{\Delta[O_x]}}{\Delta y} + \frac{-[O_3]_{SFC}*|v_d|}{h} + \frac{K_z\frac{\Delta[O_x]}{\Delta z}}{h} \qquad (1)$$

Where the term on the left represents the change in concentration with respect to time within the flight volume. The leftmost term on the right side of Eq. 1 represents the net loss of $O_x$ due to chemical reaction of the resultant $NO_3$ and

contains an unknown constant of proportionality, α, which depends on the subsequent reaction pathway of $NO_3$, and can range from $0 - 3$. For reasons later discussed, α is assumed to be ~ 1.5 for this analysis. The next two terms represent changes due to advection by the horizontal wind, followed by terms representing the dry deposition of ozone to the surface, and finally the vertical turbulent mixing term that uses the vertical gradient and the eddy diffusivity, $K_z$ – a number that encapsulates the strength of the overnight mixing. The storage term, chemical loss, advection, surface

ozone, and stable boundary layer height can be calculated using the aircraft data. Combining those measurements with an estimated 0.2 cm s$^{-1}$ nighttime dry deposition velocity of ozone at night in the SSJV (Padro, 1996), we can indirectly estimate $K_z$.

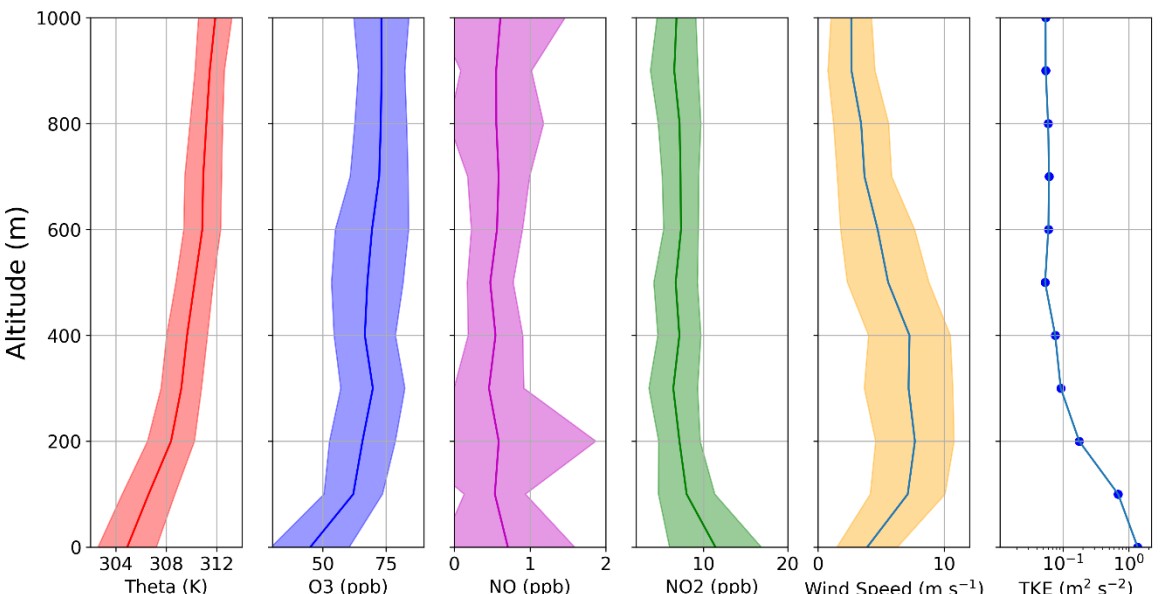

**Figure 4.** Mean and ±1 standard deviation (swatches) of potential temperature, ozone, NO, NO$_2$, and wind speed
from all late night flights.

Profiles of wind speed, potential temperature, NO$_2$, and O$_3$ from each night and morning flight were analyzed to make a best guess of the NBL height, $h$. Figure 4 shows the average scalar profiles from all 15 late night flights to illustrate

the typical gradients in the lower portion of the atmosphere. One method of determining $h$ is to observe the lowest point where $\partial\theta/\partial z$ becomes close to adiabatic, as the layer below that physically represents air that is in thermodynamic communication with the radiatively cooled surface (Stull, 1988). Another method is to use the level of wind maximum, or LLJ height, when one is present. The drop in momentum above the jet is similar to the jump in other scalars (humidity, methane, etc) often observed at the top of either the NBL or daytime atmospheric boundary layer. In our case, the vertical jump (or sharp gradient) of $O_x$ in this height region should be considered, as this likely points to a region of maximum mixing. All three of these methods were used in tandem for both the late night and corresponding morning flight to determine an average $h$ for each night. All of the aforementioned factors lead to an estimated uncertainty of $\pm$ 100 m for all of the NBL heights obtained. The average conditions from the late night and morning flights are presented in Table 1.

| Flight Date | NBL Height h (m) | NBL $O_3$ (ppbv) | NBL $NO_2$ (ppbv) | MDA8 (ppbv) | BV Frequency N ($s^{-1}$) | BRN | TKE ($m^2\ s^{-2}$) | LLJ Max $U_x$ (m $s^{-1}$) | $\sigma_u/U_x$ |
|---|---|---|---|---|---|---|---|---|---|
| 9 Sep 2015 | 250 | 45.4 | 16.5 | 82.7 | 0.025 | 0.68 | 0.35 | 8.1 | 0.09 |
| 12 Sep 2015 | 130 | 31.2 | 18.5 | 67.2 | 0.018 | 0.89 | 0.70 | 4.0 | 0.22 |
| 3 Jun 2016 | 260 | 52.7 | 6.0 | 87.8 | 0.021 | 0.23 | 0.35 | 12.0 | 0.05 |
| 4 Jun 2016 | 220 | 59.0 | 6.1 | 92.3 | 0.026 | 0.80 | 0.50 | 5.9 | 0.12 |
| 29 Jun 2016 | 150 | 43.0 | 9.9 | 91.9 | 0.022 | 0.28 | 0.41 | 10.0 | 0.08 |
| 25 Jul 2016 | 190 | 44.2 | 12.0 | 85.5 | 0.022 | 0.71 | 0.43 | 6.4 | 0.10 |
| 26 Jul 2016 | 320 | 51.6 | 8.7 | 94.8 | 0.023 | 0.99 | 0.56 | 8.0 | 0.08 |
| 13 Aug 2016 | 150 | 49.8 | 13.9 | 92.1 | 0.017 | 0.41 | 0.61 | 9.1 | 0.08 |
| 15 Aug 2016 | 250 | 42.5 | 11.6 | 74.3 | 0.023 | 0.37 | 1.02 | 10.3 | 0.08 |
| 16 Aug 2016 | 210 | 44.8 | 14.1 | 86.8 | 0.025 | 0.52 | 0.71 | 9.4 | 0.10 |
| 17 Aug 2016 | 170 | 48.3 | 15.9 | 91.5 | 0.024 | 1.35 | 0.74 | 6.2 | 0.12 |
| 18 Aug 2016 | 190 | 48.8 | 12.6 | 92.2 | 0.025 | 1.00 | 0.71 | 5.6 | 0.17 |
| **Average** | **208** | **46.8** | **12.1** | **86.6** | **0.023** | **0.69** | **0.59** | **7.9** | **0.11** |
| **Stdev.** | **53** | **6.5** | **3.8** | **7.9** | **0.003** | **0.32** | **0.19** | **2.2** | **0.04** |

**Table 1.** NBL heights, ozone, $NO_2$, Brunt–Väisälä (BV) frequencies, Bulk Richardson Number (BRN), Turbulent Kinetic Energy (TKE), and LLJ maximum wind speeds observed during the late night / morning flight pairs. Maximum daily 8-hour average ozone (MDA8) values are from the following day and are an average of the 11 ground networks in our flight region.

For the domain of interest, all measured $NO_2$ and $O_3$ data was averaged for each 20 m altitude bin in order to generate mean vertical profiles of $O_x$. Separate profiles were created for the late night flight and the subsequent morning flight. The height of the stable boundary layer for each night ($h$) was used as the upper altitude limit when averaging observations to obtain advection, chemical loss, and time rate of change (storage) terms for the budget equation, since the budget equation is meant to be applied to the NBL. The overnight average $O_x$ profile was subtracted from the Sunrise profile and divided by the time difference between the midpoints of each flight to compute the storage term.

### 2.2.1. Ozone and $NO_x$ Chemistry

As previously mentioned, the chemical loss term in Equation 1 is expected to be an important component of the NBL $O_x$ budget. Both $NO_2$ and $NO_3$ are able to regenerate ozone in the presence of sunlight and participate in the same sequence of reactions, which are grouped together into a family referred to as odd oxygen ($O_x$). $O_x$ is usually defined

as $O_3+NO_2+2NO_3+3N_2O_5$ (Brown et al., 2006; Wood et al., 2004); however, since we were unable to measure the higher oxidation state $NO_y$ species, we will define $O_x$ as $O_3+NO_2$, as these are by far the dominant species of $O_x$. Considering $O_x$ is useful in this case because the family is conserved in the rapid oxidation of NO by $O_3$ (R1), yielding $NO_2$ that is quickly photolyzed back to $O_3$ once the sun rises as part of the standard daytime photostationary state. Aside from dry deposition to the Earth's surface, $NO_x$ chemistry is the main loss of ozone at night, counteracting its role in production during the daytime (Brown et al., 2006, 2007). The chemical loss of ozone at night begins with the production of the nitrate radical (R2):

**(R1)** $NO + O_3 \rightarrow NO_2 + O_2$
**(R2)** $NO_2 + O_3 \rightarrow NO_3 + O_2$

$NO_3$ photolyzes rapidly once the sun rises, so the ultimate net loss of ozone depends on the loss of nitrate in the dark. The loss occurs mainly via three general channels. In one channel, dinitrogen pentoxide is formed (R3), which has a backwards reaction and can be a source of $NO_2$ if not deposited onto moist surfaces or aerosols to form nitric acid via hydrolysis (R4):

**(R3)** $NO_3 + NO_2 + M \leftrightarrow N_2O_5 + M$
**(R4)** $N_2O_5 + H_2O \rightarrow 2HNO_3$
**Net (R1-R4)**: $NO + 2O_3 + NO_2 \rightarrow 2NO_z$

where $NO_z = NO_y - NO_x$ to represent the family of products of $NO_x$ oxidation. In another channel, nitrate is lost by reaction with a wide array of organic compounds:

**(R5)** $NO_3 + (VOC, etc.) \rightarrow$ organic nitrates

**Net (R1, R2, R5)**: $NO + 2O_3 \rightarrow NO_z$

However, in urban environments with nocturnal sources of NO, nitrate is reduced back to $NO_2$ by the very rapid reaction:

**(R6)** $NO + NO_3 \rightarrow 2NO_2$

**Net (R1, R2, R6)**: $2NO + 2O_3 \rightarrow 2NO_2$

If the hydrolysis of $N_2O_5$ (R4) is the dominant $NO_3$ sink, then the net reaction leads to a loss of 3 $O_x$ molecules per nitrate produced (R2). However, if the dominant loss is reaction with VOC's (R5) then the net reaction leads to 2 $O_x$ molecules lost per R2. And if there is sufficient NO, R6 will dominate the nitrate loss leading to no net $O_x$ loss. Thus, determining the dominant loss of nitrate is crucial for our analysis.

Reaction (R6) has often been ignored at night under the presumption that local sources of NO are sparse and reaction (R1) will outcompete reaction (R6) (Brown et al., 2007; Stutz et al., 2010); however, at 30 ppb of $O_3$ and 20 ppt of $NO_3$ the lifetimes of NO to (R1) and (R6) are nearly equivalent (~80s). Our measurements indicate ground-level NO of about 0.6 ppb at midnight (SD = 1 ppb), corroborated by the surface air quality network, increasing in the early morning hours to 2-4 ppb. However, both the ground network and aircraft observations may be biased high to the regional average because of their proximity to California Highway 99 and other urban centers (Fig. 3). Nevertheless, the rate of reaction (R6) is $2.6 \times 10^{-11}$ cm$^3$ s$^{-1}$ molec$^{-1}$ (Sander et al., 2006), extremely rapid relative to the others, such

that even 60 ppt of NO, an order of magnitude lower than what our measurements indicate, results in an $NO_3$ lifetime of only 25 seconds. Hence, we conclude that (R6) should not be ignored.

There is then a further question as to whether any VOCs would be able to compete with this channel of $NO_3$
consumption. An investigation into the faster VOC reactions with $NO_3$ per Atkinson et al. (2006) and Gentner et al. (2014a) is presented in Table 2. The estimated lifetime of $NO_3$ due to the VOC reactions in Table 2 is 9.5 seconds, about four times the lifetime of $NO_3$ with respect to the presence of 0.6 ppb of NO (2.5 seconds). We note that although there are few direct observations of $NO_3$ in the SSJV, the CALNEX campaign conducted one flight that measured concentrations of about 10-40 ppt shortly after sunset on 24 May 2010
(https://esrl.noaa.gov/csd/groups/csd7/measurements/2010calnex/P3/DataDownload/index.php). Smith et al. (1995) present DOAS measurements from 15 nights in July and August 1990 (their Figure 6a) from a site 32 km southeast of Bakersfield suggesting that $NO_3$ concentrations in the SSJV peak around 30 pptv within an hour or two after sunset and plateau in the middle of the night around 10 ppt, then decline to zero by sunrise. The variability of $NO_3$ reported in that study is high, with nocturnal values ranging from near zero to over 50 ppt. Under a simplified, steady-state
model, the expected lifetime of $NO_3$ can be estimated using the second-order reaction rate for (R2) for the formation of the nitrate radical, and combining all of the loss channels into a single lifetime ($\tau_{NO3}$):

$$\tau_{NO3} = \frac{[NO_3]}{k_2[NO_2][O_3]} \qquad \textbf{(2)}$$

Using the average NBL ozone and $NO_2$ from Table 1, a $NO_3$ concentration of 10 ppt would imply its lifetime to be about 25 seconds, which is about twice as large as our estimate from Table 2. Based on these direct measurements of $NO_3$, our lifetime calculations likely represent a lower bound and further illustrate the uncertainty given the sensitivity to the unconstrained VOCs and our NO measurements, which have an envelope of error that spans a large range of possible nitrate loss lifetimes.

With longer lifetimes of nitrate loss with respect to the VOC and NO reactions, we are faced with the possibility that hydrolysis of $N_2O_5$ is also an important loss channel, increasing the amount of $O_x$ molecules lost per nitrate molecule formation in (R2). Smith et al. (1995) report that the lifetime of $NO_3$ was found to be highly dependent on relative humidity, with lifetimes ranging from seconds to 10 minutes when the relative humidity is above 45 % (presumably due to $N_2O_5$ hydrolysis), but between 10 and 60 minutes when below the 45 % threshold. Figure 5 shows the diurnal
cycle of temperature and relative humidity observed at the airports in our flight region during the days of our campaign, compared with the 2015-2016 1 June – 30 September averages. The > 45 % relative humilities observed at FAT and VIS imply that the hydrolysis of $N_2O_5$ is an important sink for $NO_3$.

Given the obvious importance of the nitrate loss to VOCs and NO, but some importance of the $N_2O_5$ hydrolysis, we use a best estimate that each effective collision of $NO_2$ and $O_3$ will lead to the net loss of approximately 1.5 ($\pm0.5$)
molecules of $O_x$ from the net effects of the entire series of reactions outlined above. This is a "center of the envelope" estimate for the possible range of $0 - 3$, and best accounts for the lack of certainty as to which (if any) nitrate loss channel is dominant. Although our measurements are unable to constrain this coefficient, the ultimate fate of the

nitrate radical can be seen to have a very important role in quantifying the net loss of $O_x$ overnight, and without a greater understanding of the nitrate budget, predicting this loss rate is highly uncertain.

| VOC | k (cm³/mlc*s) cm³ mlc⁻¹ s⁻¹ | Best Guess ppb | τ$_{NO3}$ s | Source |
|---|---|---|---|---|
| o-Cresol | 3.33 ($10^{-11}$) | 0.01 | 12.0 | Estimate |
| linalool | 2.22 ($10^{-11}$) | 0.05 | 36.1 | Arey et al. 1990 |
| 3-methylfuran | 1.90 ($10^{-11}$) | 0.009 | 234.9 | Steiner et al. 2008 |
| b-caryophyllene | 1.67 ($10^{-11}$) | 0.005 | 4.8 | Gentner et al. 2014b |
| 6-Methyl-5-hepten-2-one | 1.67 ($10^{-11}$) | 0.05 | 48.2 | Estimate |
| limonene | 1.33 ($10^{-11}$) | 0.026 | 116.6 | CalNex |
| myrcene | 1.11 ($10^{-11}$) | 0.01 | 378.8 | Gentner et al. 2014b |
| sabinene | 9.52 ($10^{-12}$) | 0.003 | 1284.2 | CalNex |
| b-phellandrene | 8.33 ($10^{-12}$) | 0.05 | 96.4 | Estimate |
| Phenol | 7.41 ($10^{-12}$) | 0.05 | 108.4 | Estimate |
| a-pinene | 6.06 ($10^{-12}$) | 0.047 | 142.3 | CalNex |
| b-pinene | 2.67 ($10^{-12}$) | 0.003 | 4653.8 | CalNex |
| Trans 2-butene | 7.94 ($10^{-13}$) | 0.13 | 389.3 | Steiner et al. 2008 |
| isoprene | 6.94 ($10^{-13}$) | 0.068 | 853.2 | CalNex |
| camphene | 6.54 ($10^{-13}$) | 0.007 | 8502.4 | CalNex |
| **NET** | | | **9.5** | |


**Table 2.** Estimations of VOC reactions with nitrate in the summertime nocturnal boundary layer for the SSJV. Reaction rates from Atkinson & Arey (1998), Table 2 & Atkinson (2006). The measurements in some of the studies above were taken in specific crop fields. Since the aim of this analysis was merely to obtain an order of magnitude estimate, we predicted whether a valley-averaged concentration may be slightly higher or lower than what was reported in the study. Thus, values here may not exactly match literature.

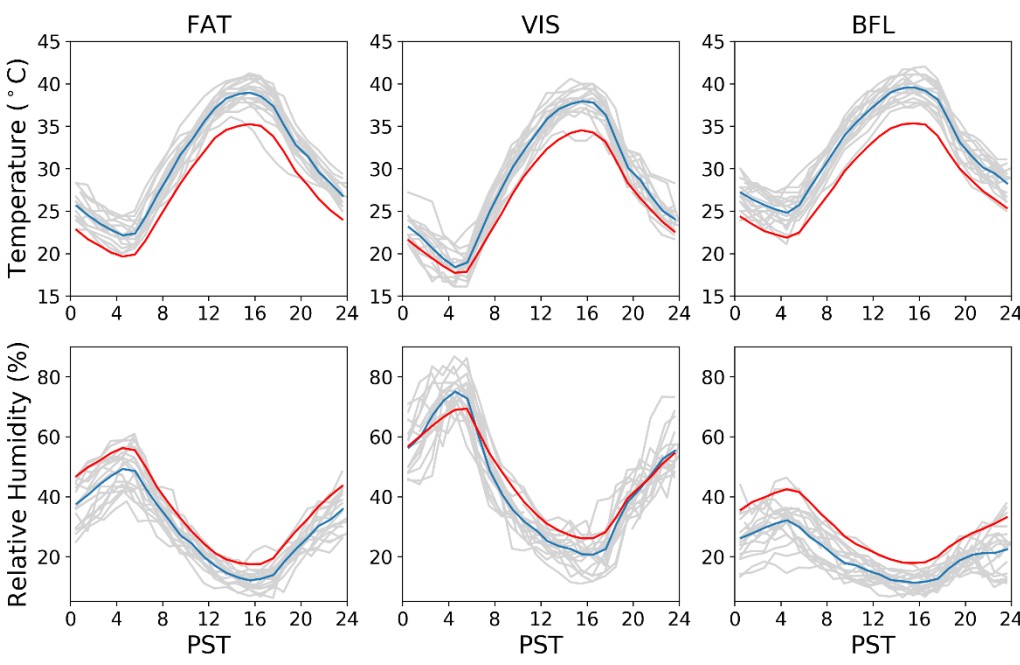

**Figure 5.** Diurnal plots of temperature and relative humidity during flight days of the Residual Layer Ozone campaign (individual days = grey lines, campaign average = blue lines), compared to 1 June – 30 September 2015

and 2016 averages (red lines) at the Fresno, Visalia, and Bakersfield airports Automated Weather Observing System (AWOS) network.

Consequently, we calculate the net reaction (R1-R6) for the nocturnal chemical loss rate of $O_x$ as a constant multiple of (R2). The 2nd order rate equation for the net chemical loss of $O_x$ is calculated by:

$$\frac{dO_x}{dt}\Big|_{chemical\ loss} = -\alpha k_{O3+NO2}[O_3][NO_2] \qquad \textbf{(3)}$$

Where α can range from 0 – 3, and per the discussion above, is estimated to be $1.5 \pm 0.5$ (uncertainty discussed in section 3.2). To estimate a value for the second order rate constant ($k_{O3+NO2}$), we start with the temperature dependent function for this reaction (Sander et al., 2006):

$$k_{O3+NO2} = 1.2(10^{-13}) * e^{\frac{-2450}{T}} \qquad \textbf{(4)}$$

Where T is the temperature in Kelvin. For the domain being analyzed, an instantaneous value of $k_{O3+NO2}$ is determined at each data point. These values of $k_{O3+NO2}$ are then averaged to obtain a constant value for the given night. It should be noted that small errors in the value of $k$ that would be within the order of our temperature fluctuations were found to not have a measurable impact on the chemical loss term. To estimate the chemical loss of $O_x$, the initial 20 m altitude bins for $NO_2$ and $O_3$ are taken from the late night and morning profiles. In each bin, the concentrations are linearly interpolated between the late night and morning values, so that there is an estimation of the current average concentration within that bin at every time during the night.

### 2.2.2. Horizontal Advection by Mean Wind

The advection term in Equation 1 is calculated by first collecting all 1-second $O_x$ data points for the late night and morning flights separately. A multiple linear regression is fit through the $O_x$ data for latitude ($y$), longitude ($x$), and altitude ($z$), allowing estimations for $\partial[O_x]/\partial x$ and $\partial[O_x]/\partial y$ in the horizontal advection terms. The total advection term within the NBL on a given flight is:

$$A_{Ox} = -\left[\left(\frac{\partial[O_x]}{\partial x} * \bar{u}\right) + \left(\frac{\partial[O_x]}{\partial y} * \bar{v}\right)\right] \qquad \textbf{(5)}$$

Where $u$ is the mean $x$-component (zonal) wind and $v$ is the mean $y$-component (meridional) wind. The same procedure is repeated for the morning flights, and the advection terms from the late night and morning flights are averaged together.

### 2.2.3. Dry deposition of $O_x$

Deposition of ozone is presumed to be the main sink of $O_x$ at the surface, the flux of which can be parameterized as the product of the surface ozone values (measured directly from the aircraft) and the deposition velocity for ozone. There are reports of ozone deposition in the area of this field campaign from a 1994 study using the eddy covariance technique (Padro, 1996). The findings of their study suggest nocturnal ozone deposition velocities are a few times smaller than the daytime counterpart, but still important for the budgeting technique presented here. Results from a European field study in a flat grass field corroborates this finding (Pio et al., 2000). We thus estimate a dry deposition

velocity of 0.2 cm s$^{-1}$ ± 0.1 cm s$^{-1}$ for ozone at night in the SSJV based on these, as well as other (Pederson et al.,
1995; Meszaros et al., 2009; Neirynck et al., 2012; Lin et al., 2010), literature values.

We purposefully ignore $NO_2$ deposition on the basis that crop canopies can be either a small source or sink of $NO_2$ at
the surface (Walton et al., 1997). The amount of $O_x$ lost overnight due to deposition would be within our stated
uncertainty (± 0.86 ppb h$^{-1}$) as long as $|v_{d\ NO2}| < \sim 2.5$ cm s$^{-1}$, an assumption supported by the literature (Pilegaard et
al., 1998; Walton et al., 1997).

### 2.2.4. Vertical Turbulent Mixing between the NBL and the RL

Finally, a vertical flux divergence for $O_x$ must be estimated for Equation 1, which is represented by the last two terms.
For the top part of the stable boundary layer, the flux of $O_x$ can be interpreted as an eddy diffusivity ($K_z$) multiplied
by the vertical gradient of $O_x$ between the NBL and RL. A linear regression through the 20 m resolution vertical $O_x$
profile is used to determine $\partial[O_x]/\partial z$ (for the last term in Equation 1) in the upper portion of the NBL that appeared to
contain the strongest gradient. The layers used for the regression fit were 100 - 200 m thick and did not extend below
70 m AGL on any given night to avoid capturing the region where the $O_x$ sink due to surface deposition is likely to
account for the vertical gradient (Fig. 6). The eddy diffusivity can now be solved for with all of the other terms
estimated.

### 3. Results and Discussion

### 3.1. Overnight Mixing and the $O_x$ Budget

Figure 6 shows an example of the observed profiles of $O_x$ on the late night and morning flights, for the series performed
on 2016-06-04. The height of the NBL is shown (green), and the lower bound of the layer used in the vertical gradient
fit is shown (yellow). The dashed profiles show the expected profile that would have been observed on the morning
flight if only advection (blue), chemical loss (green), or both advection and chemical loss (red) processes were
occurring. The observed morning $O_x$ (magenta) is inferred to exceed the predicted morning $O_x$ (red) due to the vertical
mixing term in the scalar budget equation.

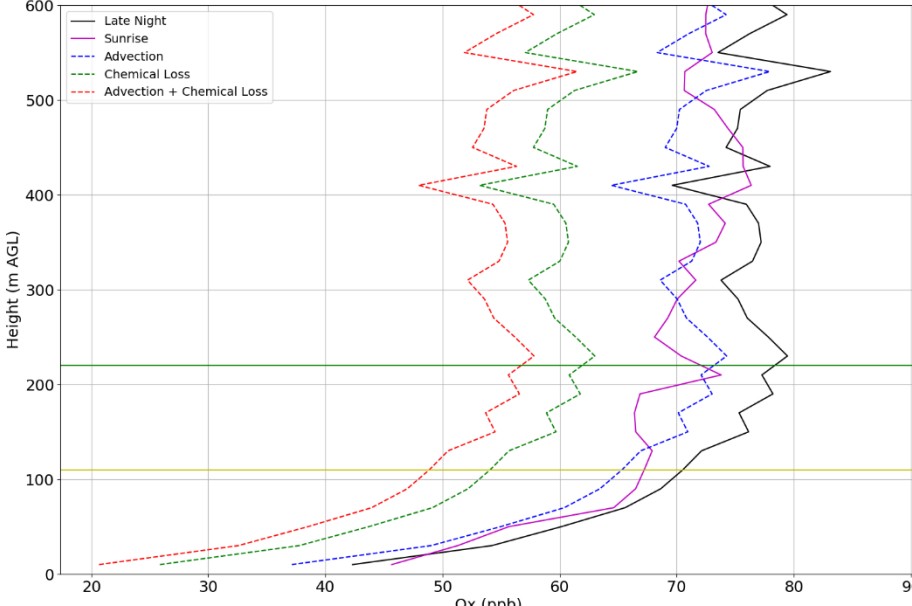

**Figure 6.** $O_x$ profiles from 2016-06-04 overnight analysis, NBL height (green line), and lower bound to vertical mixing gradient (yellow line). The solid lines are observations and the dashed lines are inferred.

450     Results of the scalar budget analysis for all 12 paired flights are presented in Table 3. An error propagation analysis (discussed in section 3.2) is presented for each term in the budget, as well as for the ultimately calculated $K_z$ values.

| Flight Date | Storage ppb h$^{-1}$ | Advection ppb h$^{-1}$ | Chemical Loss ppb h$^{-1}$ | Vertical Mixing ppb h$^{-1}$ | Deposition ppb h$^{-1}$ | Eddy Diffusivity m$^2$ s$^{-1}$ |
|---|---|---|---|---|---|---|
| 9 Sep 2015 | -2.3(0.2) | -3.2(0.2) | -3.6(1.3) | 5.1(3.1) | -0.6(0.4) | 3.0(1.3) |
| 12 Sep 2015 | -0.2(0.2) | -0.0(0.1) | -2.9(0.9) | 4.0(5.2) | -1.4(1.3) | 3.5(3.0) |
| 3 Jun 2016 | -0.7(0.1) | 0.3(0.2) | -1.5(0.4) | 1.5(0.9) | -1.0(0.6) | 2.9(1.4) |
| 4 Jun 2016 | -0.5(0.2) | -0.6(0.1) | -1.9(0.6) | 3.2(2.0) | -1.2(0.8) | 2.9(1.2) |
| 29 Jun 2016 | -1.3(0.2) | -1.0(0.1) | -2.2(0.6) | 3.4(3.1) | -1.6(1.3) | 2.0(1.1) |
| 25 Jul 2016 | -1.2(0.2) | 0.6(0.2) | -2.7(0.8) | 2.0(1.5) | -1.2(0.9) | 1.1(0.6) |
| 26 Jul 2016 | -1.4(0.2) | 0.2(0.2) | -2.2(0.8) | 1.3(1.0) | -0.7(0.4) | 1.5(1.1) |
| 13 Aug 2016 | -1.4(0.2) | -0.3(0.2) | -3.4(1.1) | 4.1(3.6) | -1.8(1.5) | 2.3(1.2) |
| 15 Aug 2016 | -1.1(0.1) | 0.6(0.2) | -2.5(0.9) | 1.8(1.3) | -0.9(0.6) | 2.6(1.6) |
| 16 Aug 2016 | -1.9(0.2) | -0.1(0.1) | -3.0(1.1) | 2.3(1.9) | -1.0(0.7) | 2.2(1.4) |
| 17 Aug 2016 | -2.0(0.2) | 0.1(0.1) | -3.7(1.4) | 2.8(2.5) | -1.2(0.9) | 1.5(1.0) |
| 18 Aug 2016 | -1.6(0.2) | 0.5(0.2) | -3.1(1.2) | 2.2(2.0) | -1.2(0.9) | 1.9(1.3) |
| **Average** | **-1.30 (0.18)** | **-0.24 (0.16)** | **-2.73 (0.93)** | **2.81 (2.34)** | **-1.15 (0.86)** | **2.28 (1.35)** |
| **Standard Dev.** | **0.59** | **1.00** | **0.66** | **1.12** | **0.33** | **0.69** |

**Table 3.** Results from the nocturnal scalar budget for all terms. Estimated error (see section 3.2) in parenthesis.

Of note is the fact that on average the chemical loss is expected to be a little more than twice as large as the physical
455     loss from dry deposition. For dry deposition the average lifetime of ozone is 28 h (200 m / 0.002 m s$^{-1}$), and for chemical loss it is 12 h. Both losses of $O_x$ added together are about triple the observed time rate of change, and thus the physical and chemical losses are largely ($\sim$ 2/3) compensated by vertical mixing. Because the RL consistently contains more ozone than the stable NBL, turbulent mixing will result in a transfer of ozone into the NBL. While $NO_2$

is observed to be higher in the NBL than in the RL (by about 3-5 ppbv), it is a much smaller contribution to the $O_x$ ($O_3$ is less than $NO_2$ by anywhere from 10-20 ppbv.) Thus, vertical mixing at the top of the stable boundary layer, influenced by the strength of the LLJ, is inherently a source term of $O_x$ to the lower NBL. It is also worth noting that the chemical loss of $O_x$ does not vary significantly between the RL and NBL because the increase of $NO_2$ in the NBL is compensated by the decrease of $O_3$, although this assumes that there are not other chemical differences that alter the ultimate reaction fate of nitrate (altering the coefficient α in Eq. 1.)

### 3.2. Error Analysis

Here we estimate the uncertainty for each term in the budget equation, as well as the ultimately calculated eddy diffusivities. The storage term error is computed by first taking the standard deviation of 1-second ozone measurements divided by the square root of the number of samples, then the standard error of the means for both the late night and morning profiles are combined. This analysis is carried out in 20 m altitude bins separately and then averaged together because there is more uncertainty at lower altitudes due to fewer measurements. The advection term error is computed from the standard error of the slopes of the regression fit, with errors propagating for each of the 4 advection components for both the $u$ and $v$ components of wind. To compute the chemical loss error, the large uncertainty of the α coefficient must be taken into consideration. Based on our analysis concluding that all channels of nitrate loss are probably non-negligible, we infer that α is between 0.5 and 2.5 with a 95 % confidence interval. Thus, one standard error for the α coefficient is about 0.5. An error propagation is then carried out for each 20 m altitude bin, using the standard deviations of the $O_3$ and $NO_2$ measurements divided by the square root of the sample size. As previously stated, the estimated standard errors of the stable boundary layer height and surface deposition of ozone are taken to be 100 m and 0.1 cm s$^{-1}$, respectively. The surface ozone standard error is computed as the standard deviation of the aircraft measurements divided by the square root of the sample size, and the vertical $O_x$ gradient uncertainty is computed by the standard error of the regression slope. The uncertainties in the vertical mixing, deposition, and diffusivity values can then be computed by standard error propagation. The resultant relative error estimates of the nighttime diffusivities are about 50 %, and errors of this order seem reasonable based on a technique that assumes the closure of 4 independently measured terms. Past studies using similar airborne budgeting methods have estimated relative uncertainties ranging from 15-75 % (Conley et al., 2011; Faloona et al., 2009; Kawa & Pearson, 1989; Trousdell et al., 2016).

### 3.3 The Fresno Eddy and Low-Level Jet

One complicating factor that remains for this particular analysis is the presence of the Fresno eddy and its influence on our measurements of advection. If an eddy is recirculating a scalar quantity, using a simple linear fit model as we did to estimate advection would be questionable, especially if the flight area only covered a small portion of the larger mesoscale circulation. Zhong et al. (2004) uses a series of 915 MHz RASS to analyze low-level winds in the SSJV. Their Figure 4 shows that at night, the northwesterly low level jet is formed in the San Joaquin Valley, and a weak katabatic southerly flow is observed in the foothills to the east at the Trimmer site. As the night progresses, the eddy becomes more coherent as the northwesterly jet relaxes while the southerly flow strengthens and expands westward. After daybreak, the eddy appears to deform and disintegrate with much of the SSJV experiencing a strong southerly wind.

This pattern is roughly consistent with our aircraft observations, suggesting the presence of a Fresno eddy during our flights. An analysis of the average wind vectors and their consistency for all nocturnal and morning flights in the approximate stable boundary layer (0-300 m AGL) and residual layer (300-700 m AGL) are shown in Figure 7. The wind consistency is defined as the ratio of the vector-averaged wind speed to the magnitude-averaged wind speed, with values close to 1 indicating a consistent wind direction (Stewart et al., 2002; Zhong et al., 2004). The nocturnal low-level jet can be seen clearly to fill most of the SSJV in both the NBL and RL. In the morning residual layer level, there is localized consistent southerly flow closest to the foothills, some of which may be regarded as surprisingly strong. The lower level winds in the morning are consistent with the deformed eddy. We note that caution should be exercised in directly comparing our flight data to the analysis from Zhong et al. as our flights specifically targeted high ozone episode events, which we based primarily on high temperature stagnation conditions, so they may be subject to a meteorological bias (see Fig. 5).

From this analysis we conclude that it is likely that our dataset captures the bulk of the dominant flow (and thus advection) on both the late night and morning flights, which are averaged and interpolated. It is noted that the average advection term for the 12 nights presented is -0.24 ppb h$^{-1}$, which is nearly an order of magnitude smaller than the chemical loss and storage terms. The small average contribution from advection is consistent with previous findings from daytime scalar budgets performed over the oceans (Conley et al., 2011; Faloona et al., 2009) and in the SJV (Trousdell et al., 2016) and what might be expected in the presence of a recirculating eddy. Lastly, it is noted that individually adjusting each flight to have an advection term of zero (to assume full eddy recirculation) results in only a 3 % change to the average of the diffusivity values.

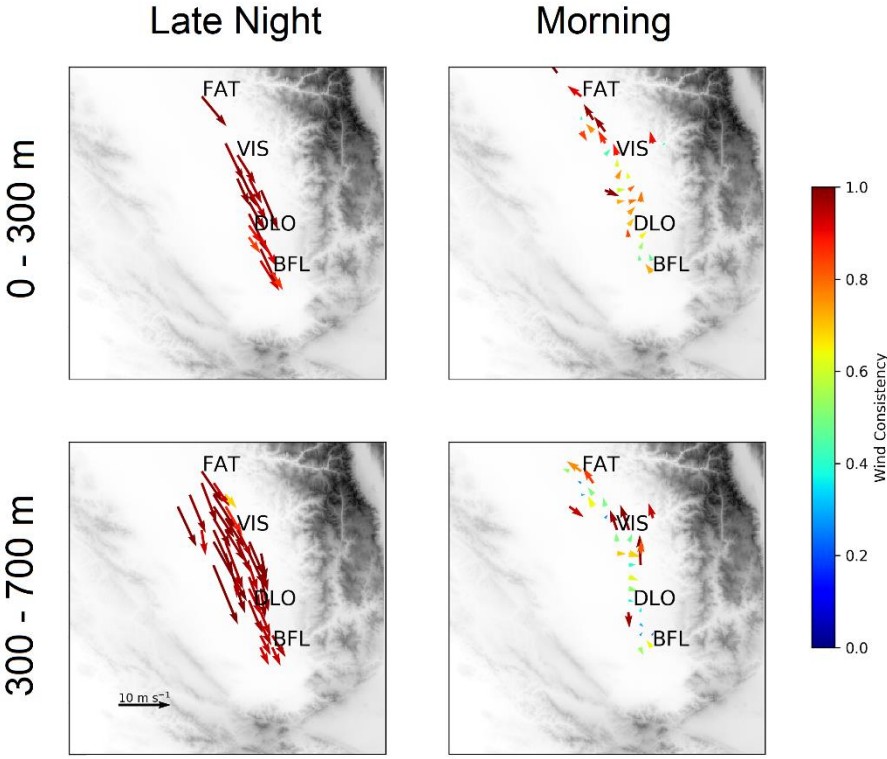

**Figure 7.** Wind consistency for late night flights and morning flights in the NBL (0 – 300 m) and the RL (300 – 700 m).

Since the low-level jet is hypothesized to contribute to the variability of maximum daytime ozone concentration, we explored the synoptic patterns that are associated with differing strengths of the LLJ. Seven years of data (2010-2016) from the 915 MHz sounder located in Visalia, CA, is compiled to obtain the low-level jet speed and the height at which it was observed. For this analysis we define the nocturnal low-level jet speed as the maximum hourly-averaged wind speed observed below 1000 m averaged in 100 m vertical bins from 23 PST to 7 PST, specifically during the summer months (defined here as 1 June – 30 September). The 1000 m cutoff is used to ensure that the wind maximum that is captured is related to the LLJ at the top of the NBL rather than free-tropospheric wind. Using this definition, the low-level jet had an average height of 340 m, an average speed of 9.9 m s$^{-1}$ (SD = 3.1 m s$^{-1}$) and a typical peak timing around 23 PST. The 700 mb level corresponds to approximately 3000 m, well above the Pacific Coast Range but approximately in line with the top of the Southern Sierras.

To analyze variability of the jet strength, daily average synoptic charts from the North American Regional Reanalysis (NARR) are created in Figures 8 and 9 for days when the low-level jet strength was less than 7 m s$^{-1}$ (N=147 nights), and greater than 12 m s$^{-1}$ (N=165 nights). Both the strong and weak low-level jets show a climatological trough pattern, but the mean trough axis is situated about 100 km to the east for the strong cases. We also note that the pressure gradient is at least twice as strong for the stronger low-level jets, and that the synoptic pattern of the weak jets favors southerly geostrophic wind aloft, which directly opposes the up-valley northwesterly thermally driven flow. We also note the positive correlation found between the LLJ strength and the upwelling index ($r^2$ = 0.3018, p < 10$^{-5}$), which is primarily driven by the North Pacific High, which when strong, acts to push the trough farther eastward as seen in Figure 8. These findings are consistent with the Lin and Jao (1995) modeling study where the Fresno Eddy (and thus LLJ) did not form when the synoptic flow over the coastal range was westerly. Beaver and Palazoglu (2009) found that maximum daily 8-hour average ozone (MDA8) exceedances were more frequent in the central and southern San Joaquin Valley when an offshore ridge or onshore high were present, consistent with Figure 8 (right). The results of our study suggest that this may be at least partially explained by the presence of a weaker LLJ under those synoptic conditions.

It is important to note that the LLJ and Fresno Eddy are not exactly the same thing, rather, the LLJ is part of the northwesterly flow that is an important precursor to the Fresno Eddy. When the eddy is present, the LLJ is essentially the strongest branch. Nevertheless, Beaver and Palazoglu (2009) conclude that recirculation from the Fresno Eddy contributes to a buildup of ozone, while we conclude that a strong jet may lead to lower ozone. Future research may attempt to further establish the degree to which the LLJ and Fresno Eddy are linked, as well as which of these two nocturnal mechanisms will dominate the ozone budget under different synoptic conditions. As a provisional synthesis of these seemingly conflicting findings, we suggest that the Fresno Eddy, when present, will act to recirculate pollutants regardless of the strength of the LLJ (the strongest branch of the eddy). That is, a stronger eddy will not recirculate pollutants any more than a weaker eddy will. Thus, the optimal nighttime dynamics for ozone pollution the following day may consist of a Fresno Eddy just coherent enough to effectively recirculate pollutants, but without its strongest branch too strong as to deplete the RL ozone by vertical mixing.

In addition to the synoptic patters discussed above, slightly lower surface temperatures across the entire region during stronger low-level jets are observed. This could either be a consequence of the synoptic flow (southerly geostrophic

flow will generally result in warmer temperatures) or itself be an underlying precursor to the LLJ (a colder delta region will lead to more up-valley thermal forcing resulting in stronger winds that decouple from the surface at night). The higher temperatures associated with the weak nocturnal jets may make for a twofold mechanism for high ozone: the high temperatures either causing increased photochemical production or resulting from increased meteorological stagnation, and a lack of mixing overnight induced by the low level jet causing less depletion of the RL ozone. Warmer

nights may also result in less dry deposition of $O_x$ through stomatal pores. It is worth noting that this relationship with temperature is only apparent with the NARR climatology, as ambient overnight low temperature at Visalia yields only a very weak relationship with the jet strength ($r^2 = 0.035$, $p < 10^{-5}$).

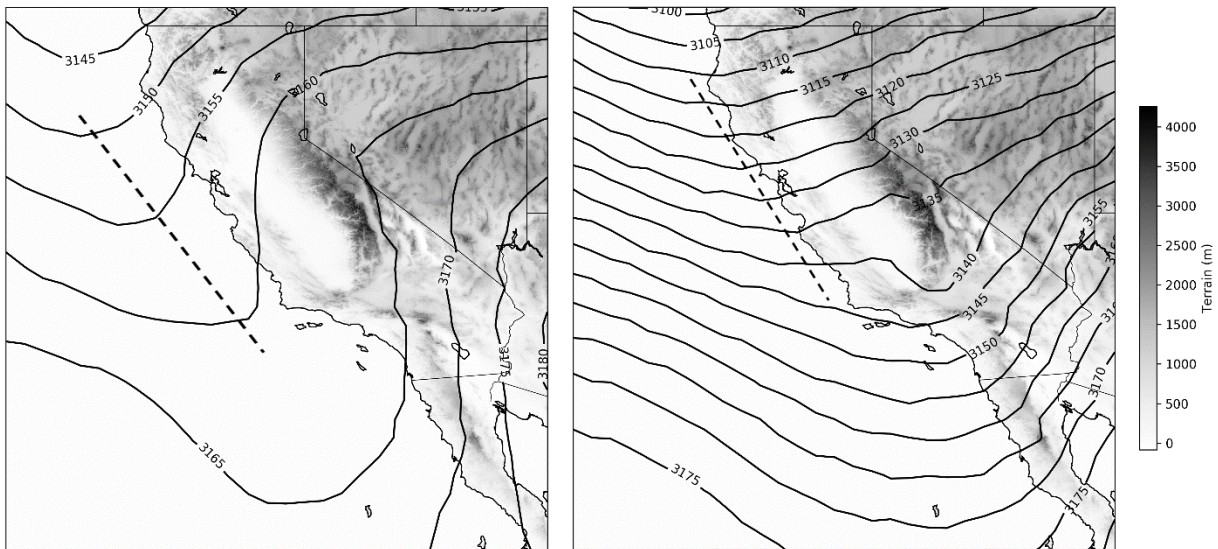

**Figure 8.** North American Regional Reanalysis 700 mb Geopotential Height (m) for low-level jet speeds exceeding
12 m s$^{-1}$ (left) and below 7 m s$^{-1}$ (right).

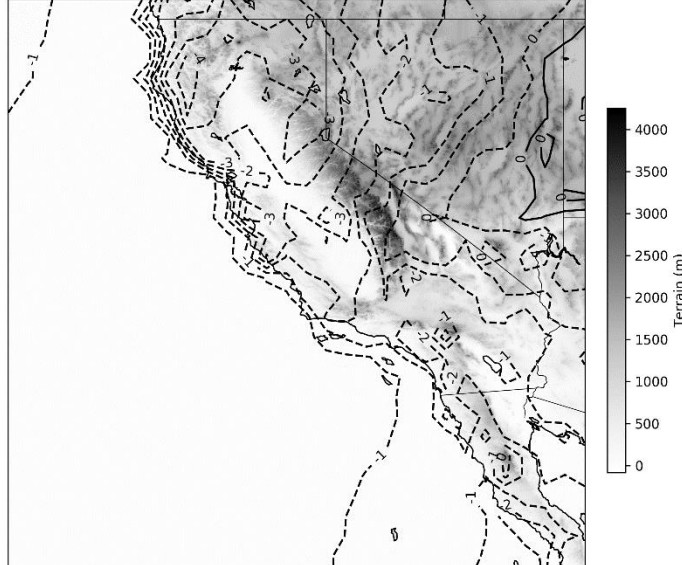

**Figure 9.** North American Regional Reanalysis 2 m air temperature (°C) difference between cases where low-level jet speeds exceeding 12 m s$^{-1}$ and cases where it is below 7 m s$^{-1}$. Positive values indicate warmer surface temperatures for strong jets.


### 3.4. Vertical Turbulent Mixing and the LLJ

As seen in Figure 4, an average low-level jet height between 200-400 m is seen, which corresponds approximately with the average observed stable NBL depth. Likely due to the shear induced by the LLJ, turbulence is seen to be vigorous at night with TKE values about 50 % of what is observed during the daytime during convective conditions. However, TKE is seen to increase toward the surface, contrary to what would be expected in the presence of an elevated jet. Banta et al. (2006) refers to this as a "traditional" TKE profile.

The physical significance of mixing overnight in relation to the air pollution problem remains somewhat of an open question. On the one hand Beaver & Palazoglu (2009) suggest that a stronger Fresno Eddy circulation is associated with higher ozone pollution potential. On the other hand, several previous studies examining different parts of the world have proposed that mixing induced by nocturnal jets may decrease ozone levels the following day (Hu et al., 2013; Neu et al., 1995). Greater coupling between the NBL and RL could reduce the amount stored in the RL reservoir rendering cleaner air the following day. This relationship between the eddy diffusivity values found in our study and regional mean surface ozone from the CARB network is analyzed, and serves as both an additional check on the relative validity of the calculated $K_z$ values as well as a test of this proposed hypothesis.

The thermals generated by solar heating after sunrise initiate a fumigation process where as the daytime boundary layer develops, the ozone that was in the RL will be mixed downward. The change in surface ozone concentration (d[O$_3$]/dt) due to fumigation peaks at around 8 am PST and continues until about 10 am PST. The relationship of our estimated eddy diffusivities with ozone during the fumigation period is strongest at 10 am PST, after the bulk of the

vertical mixing due to the boundary layer growth entraining into the RL has occurred ($r^2$=0.291, p=0.070). The relationships between eddy diffusivities and the maximum 1-hour ozone, 24 hour average ozone, and MDA8 were also in the predicted direction, with the strongest relationship found for the MDA8 ($r^2$=0.463, p=0.015).

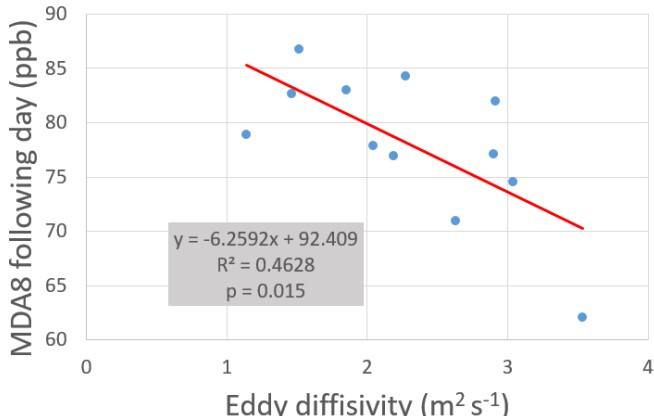

**Figure 10.** Correlation between overnight eddy diffusivity and maximum daily 8 hour average ozone (MDA8) the
following day. All values are averages of 11 CARB surface network stations that are within the flight region.

Because this analysis consisted of only 12 flights, we decided to explore a larger data set that might support the hypothesis that a stronger LLJ reduces ozone the following day. 7 years of low-level jet speeds obtained from the Visalia sounder from 2010 – 2016 is combined with the CARB surface network ozone monitoring site at Visalia N Church St (36.3325° N, 119.2908° W, 30 m elevation) for analysis. Only calendar days 152 through 273 (June –
September) is included. The low level jet, hypothesized to be the main contribution to the variability in overnight mixing between the RL and NBL, is compared with MDA8 observed the following day, shown in Figure 11. It can be seen that a stronger nocturnal low-level jet is correlated, albeit weakly, with lower ozone the following day ($r^2$=0.181, p < $10^{-5}$). A single outlier was removed where the LLJ exceeded 25 m s$^{-1}$. This is in line with our hypothesis that the low level jet will lead to stronger mixing, which leads to more residual layer ozone depletion.

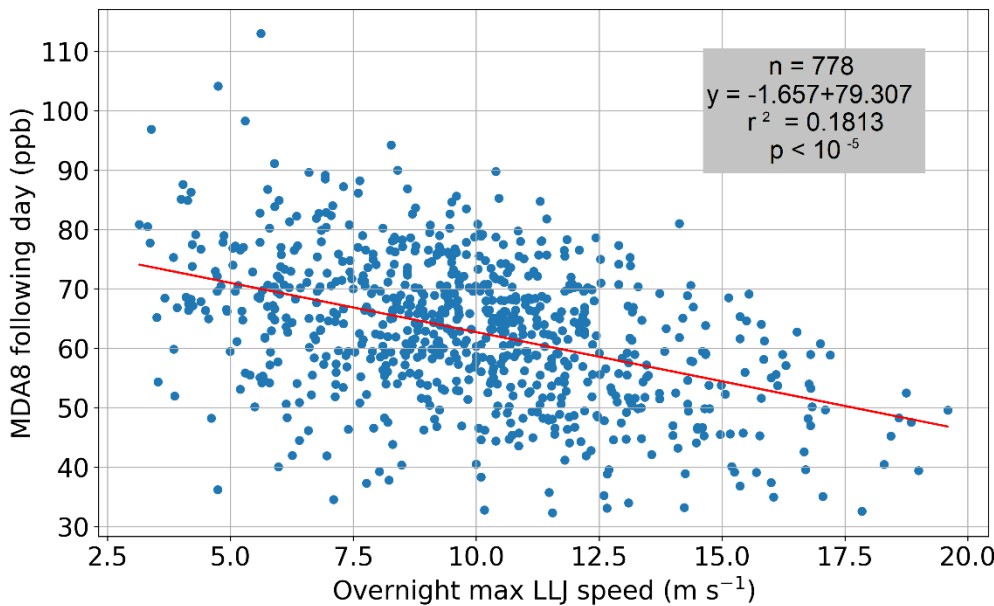


**Figure 11.** Correlation between nocturnal low level jet speed and the following day's MDA8 in Visalia, CA, for Calendar days 152-273 from 2010-2016.

In addition to a stronger LLJ mixing down more ozone, a further possibility is that the deposition velocity of ozone may be enhanced by a reduction of aerodynamic resistance under a stronger LLJ. The dry deposition of any gas may be characterized by a series of resistances (Wesely, 1989):

$$v_d = \frac{1}{r_a + r_b + r_c} \quad (6)$$

Where $r_a$ is the aerodynamic resistance, $r_b$ is the viscous sub-layer resistance, and $r_c$ is the surface (canopy) resistance. Figure 4 in Padro (1996) suggests that for ozone at night, $r_a \sim r_c \sim 250$ s m$^{-1}$. $r_b$ is likely non-zero (Massman et al., 1994) but will be neglected here because it is unknown. Combining the aerodynamic resistance due to mass transfer ($r_a = u\ u_*^{-2}$, where $u_*^2$ is the momentum flux) and parameterizing the momentum flux as a function of 10-meter wind speed $U_{10}$ and the bulk transfer coefficient for heat $C_H$ ($u_*^2 = C_H\ U_{10}^2$) we roughly approximate $r_a$ as:

$$r_a \sim \frac{1}{C_H\ U_{10}} \quad (7)$$

In the 7 years of LLJ data at Visalia, The 10-meter wind speed is correlated with the jet strength ($r^2 = 0.309$, $p < 10^{-5}$). On average, $U_{10}$ was 1 m s$^{-1}$ for 5 m s$^{-1}$ jets, and 2.5 m s$^{-1}$ for 15 m s$^{-1}$ jets. An average $U_{10}$ of 1.75 m s$^{-1}$ would imply that $C_H \sim 2.3 \times 10^{-3}$. A sensitivity analysis indicates that this difference in $U_{10}$ between strong and weak jets would result in a 40 % change in $v_d$. We thus conclude that the LLJ likely plays a significant role in modulating the dry deposition rate, where a strong jet decreases $r_a$ and thus increases $v_d$, further contributing to a loss of ozone overnight. It is important to note that what we have presented is only a rough estimate of the variability of $r_a$, and thus future studies will need to measure these parameters with more precision in order to better estimate the degree to which the LLJ can modulate dry deposition in the SJV.

### 3.5. Eddy Diffusivity and other estimates of Turbulence

Here we attempt to build confidence in the eddy diffusivity estimates by analyzing additional metrics of turbulence. We find that nocturnally and spatially averaged TKE in the NBL ranges from 0.35 and 1.02 m$^2$ s$^{-2}$, which is very comparable to values obtained in other NBL studies (Banta et al., 2006; Lenschow et al., 1988). Table 1 shows the TKE, LLJ speed, as well as the ratio of the streamwise variance to LLJ speed ($\sigma_u/U_x$) for each night. The average value of $\sigma_u/U_x$ in this study is 0.11, approximately double what was reported in Banta et al. (2006), although we did not attempt to remove buoyancy waves from our data. There is no detectable relationship between our calculated NBL TKE and eddy diffusivities, LLJ speed, or MDA8 the following day.

Our budget method of estimating turbulent dispersion differs from some other attempts that have been made for stably stratified environments. Clayson and Kantha (2008) applied a technique that has been used in oceans to the free troposphere, where turbulence is sparse and intermittent, much like the NBL. Their method involves using high-resolution soundings to estimate a length scale of overturning eddies, known as the Thorpe scale (Thorpe, 2005), which is then used to obtain estimates of turbulent dissipation rate, and subsequently eddy diffusivity. This is done by relating the Thorpe scale to the Ozmidov scale, where if the Brunt-Vaisala frequency ($N_{BV}$) is known, TKE dissipation rate ($\varepsilon$) can be estimated. Eddy diffusivity can then be estimated as a product of the TKE dissipation and $N^{-2}$:

$$K_z = \gamma \varepsilon N_{BV}^{-2} \quad (8)$$

Where $\gamma$ is the mixing efficiency, which can vary between 0.2 and 1 (Fukao et al., 1994). From the nocturnal power spectra (Fig. 1) we use a Kolmogorov fit to estimate $\varepsilon$, which is determined to be approximately $4.8 \times 10^{-6}$ m$^2$ s$^{-3}$ for the overall altitude range of our nighttime flights (surface to $\sim$3000 m), but a median of $3.0 \times 10^{-4}$ m$^2$ s$^{-3}$ is observed in the NBL. Using the average NBL Brunt–Väisälä frequency of 0.023 Hz and a mixing efficiency of 0.6 results in an eddy diffusivity of 0.34 m$^2$ s$^{-1}$, which is about three times smaller than the lower end of our range (1.1 – 3.5 m$^2$ s$^{-1}$). A recent study of vertical mixing based on scalar budgeting of Radon-222 in the stable boundary by Kondo et al. (2014) estimated 7-day average overnight diffusivities of 0.05 – 0.13 m$^2$ s$^{-1}$, which is an order of magnitude below our estimates inferred from the O$_x$ budget. However, Wilson (2004) conducted a meta-analysis of radar-based estimates of eddy diffusivity in the free troposphere, which is also a generally stable environment, and found a general range of 0.3 – 3 m$^2$ s$^{-1}$. Pisso and Legras (2008) estimated diffusivities of about 0.5 in the lower stratosphere during Rossby wave-induced intrusions of mid-latitude air into the subtropical region. A modeling study by Hegglin et al. (2005) reports diffusivities of 0.45 – 1.1 m$^2$ s$^{-1}$ in the lower stratosphere with an average Brunt–Väisälä frequency of 0.021 Hz, indicating a similar turbulent environment to ours. Finally, Lenschow et al. (1988) analyzed flight data in the NBL over rolling terrain in Oklahoma, and found eddy diffusivities for heat ($K_h$) of $\sim$0.25 m$^2$ s$^{-1}$ for the upper half of the NBL, and $\sim$1 m$^2$ s$^{-1}$ for the lower half. To our knowledge, the latter is the most comparable observational finding within the NBL to our range of diffusivities. Nevertheless, the variability in the reported values leads to the inevitable conclusion that vertical diffusivity in very stable environments is poorly understood, and further research is necessary to illuminate its phenomenology. More specifically, while it is possible that the diffusivity measurements in this study are slightly large, it is also possible that the LLJ and other mesoscale wind features of the complex terrain account for stronger nocturnal mixing in the SSJV compared to other stable environments.

Lastly, we estimate the Bulk Richardson number (BRN) on each late night flight within the NBL, using 100 meter bins to estimate wind shear. A range of Richardson numbers between 0.23 and 1.34 is obtained, and the estimates are seen to have a negative relationship with eddy diffusivities, as expected (Fig. 12). While the relationship is not strong, it is important to remember that both parameters are noisy estimates.

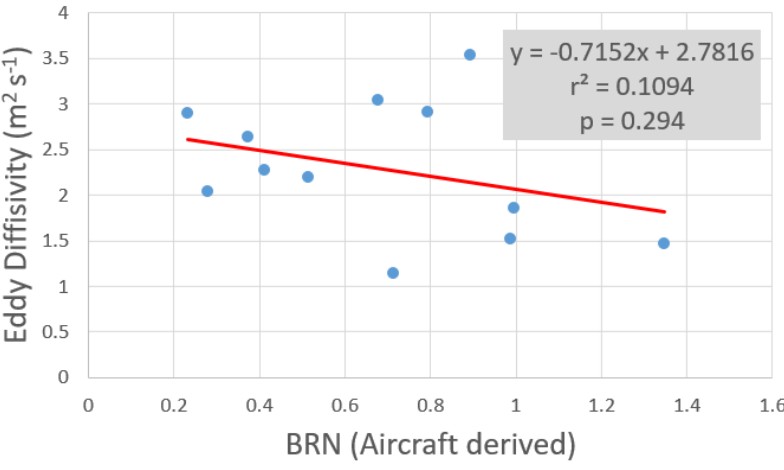

**Figure 12.** Eddy diffusivities and Bulk Richardson Numbers (BRN) derived from aircraft observations.

### 3.6. Nocturnal Elevated Mixed Layers

During the late night flights in stable environments, the flight crew reported many patches of turbulence. While most
of these subjective reports were during low approaches and thus likely attributable to wind shear between the LLJ and
the surface, they noted that some patches corresponded with what appeared to be elevated mixed layers, i.e. layers of
air where virtual potential temperature was observed to decrease with height. Understanding these anomalies may
guide future research toward a deeper phenomenological understanding of overnight mixing and turbulence in the
SSJV.

The time series of all late night flights was scanned for any period where 1) the aircraft maintained an ascent (or
descent) rate of at least 1.4 m s$^{-1}$, and 2) during a given elevation span of 100 m, a virtual potential temperature
decrease with height was observed. The process was repeated for a thickness of 50 m.

The locations of the layers detected, along with their elevation and magnitude, is shown in Figure 13. One feature of
note is that the layers appear to be more prominent over urban areas, such as Fresno, Visalia, and Bakersfield. This
may lead one to suspect that some of these layers are driven by an urban heating effect, however, this seems unlikely
as the unstable layers appear to be above the NBL where there is communication with the surface. It is perhaps more
likely that this is an artifact of more flight time in those areas. Another feature worth noting is that more unstable
layers are observed closer to the Tehachapi pass. One possible explanation for this is that the katabatic flow down the
mountain slopes detrain along the way and are carried over the valley by local advection before mixing with
surrounding air. Given that these layers are found from near the bottom of the residual layer all the way up to 2.5 km,
it is possible that they contribute to the overnight mixing of O$_x$ from the RL to the NBL. Further research, both
observational and modeling-based, is needed to explore this possibility.

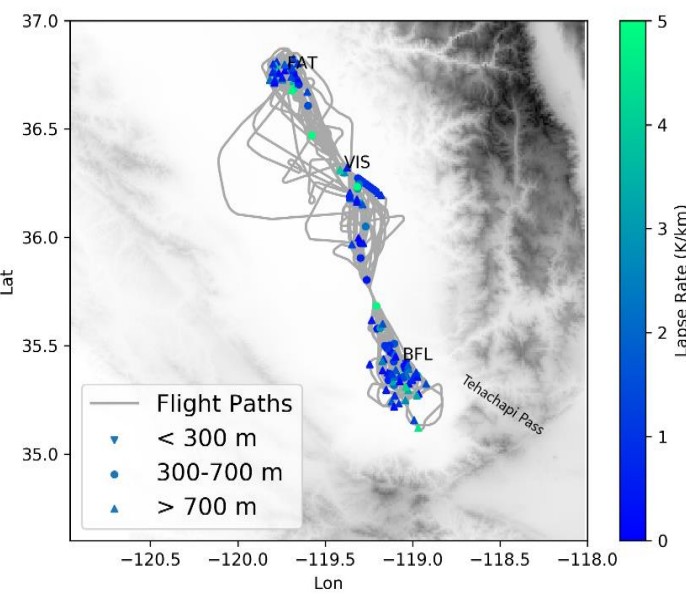

**Figure 13.** Detected nocturnal elevated mixed layers with at least 50 meters thickness, with elevations shown.

The unstable layers are not seen to have more TKE than the rest of the atmosphere, and this may reflect the limitations
of the method used to estimate turbulence from this low-cost wind measurement system. However, this is consistent

with the study by Cho et al. (2003) which found no relationship between turbulence and static stability in the free troposphere. Since the aircraft is moving horizontally a lot faster than it is vertically, one may be concerned that our observations of elevated mixed layers are an artifact of localized temperature gradients that are more prominent in the horizontal dimension. To check this, we plotted the wind quivers in the unstable layers along with the direction of the colder air. The cooler air was not systematically detected in any one direction, which supports the hypothesis that these are true vertical temperature gradients.

To analyze the stability, wind shear, and turbulence from a climatological standpoint, a July-August 2016 composite of the 915 MHz Visalia sounder data is presented in Figure 14. Even in the climatological averages, some nocturnal unstable layers are detectable between 500 and 1500 m.

**Figure 14.** Stability and wind quivers for Visalia 915 MHz sounder, 1 Jul 2016 – 31 Aug 2016.

**4. Conclusions**

We have demonstrated a method for performing a nocturnal scalar budget analysis using aircraft data, and applying it to estimate the effects of turbulence in the stable boundary layer which can be related to air quality problems. Inherently, eddy diffusivity estimates for any given night will have a large uncertainty due to the indirect nature of the measurement and the limited flight durations. However, the overall between-flight consistency and the correlations with both the Richardson number and surface ozone suggest that this method is informative. We obtain eddy diffusivity values between 1.1 and 3.5 $m^2$ $s^{-1}$, which are larger but approximately within the same order of magnitude of values that have been obtained from other studies in the free troposphere, lower stratosphere, and nocturnal boundary layer. A limitation of our study is the lack of sample size, with only 12 pairs of overnight and morning flights. However, we believe this study demonstrates the importance of synoptic and mesoscale features at night within

the context of high ozone episodes, and the utility of this type of focused flight strategy where terms in the scalar budget equation are measured.

The larger set of RASS and ARB surface network data from Visalia, CA establishes a correlation between low level jet speed and the maximum 1-hour ozone the following afternoon for summertime months, further suggesting the link between nocturnal mixing and the following days ozone. Similarly, the correlations between the aircraft-estimated eddy diffusivities and MDA8 the following day also suggest that vertical mixing in the NBL plays an important role in determining ozone concentrations. In particular, we note that 11 of 12 days where the Visalia, CA ozone

concentration exceeded 100 ppb was preceded by a low-level jet speed < 9 m/s. While we cannot determine a causal relationship between a strong low-level jet, stronger mixing, and reduced ozone pollution, we propose that a stronger LLJ leads to greater mixing, which helps deplete the ozone reservoir by bringing it into the stable boundary layer overnight. There it is subject to deposition to the surface, and that dry deposition rate may itself be partially modulated by the strength of the LLJ through reduced aerodynamic resistance resulting in more efficient transport to surfaces

where ozone can deposit. Subsequently, when thermals begin to form after sunrise the following morning, there is less ozone to fumigate downward. While the correlation between nocturnal mixing and ozone the following day is not always strong, it is an important link that may have consequential implications for modeling studies and policy making. For example, our findings highlight the crucial need of models to capture the LLJ and Fresno eddy with sufficient resolution. Policy makers may consider putting more stringent emission limitations on days where synoptic and

mesoscale patterns appear to favor a lack of overnight mixing.

Of course, in addition to nocturnal mixing, photochemical production of ozone as well as advection will play a major role in the ultimate daytime peak ozone levels observed, which may be why the correlation between nighttime turbulence and afternoon ozone is not always high. Airborne measurements from flights over Bakersfield, CA showed an average photochemical production as high as 6.8 ppb $h^{-1}$, with an average advection of -0.8 ppb $h^{-1}$, though on any

given day advection tended to be more comparable in magnitude to photochemical production (Trousdell et al., 2016). In that study they have demonstrated that on days with very high ozone that pose hazards to human and agricultural health, the ozone abundance is dependent on elevated ozone in the mornings that serve to catalyze photochemical production through the afternoon. Future modeling studies may directly investigate these factors, which may help elucidate the causal mechanisms of high ozone events.

We have also suggested that the fate of the $NO_3$ plays an important role in the nocturnal $O_x$ budget chemical loss term, and thus likely impacts the following day's maximum ozone concentration. The loss of the nitrate radical at night can occur from $N_2O_5$ hydrolysis, reaction with VOCs, or a very rapid reaction with small NO concentrations, and there is considerable uncertainty regarding which reactions dominate without direct measurements of $NO_3$. Thus, the lifetime of $NO_3$ can range from seconds to several minutes, which affects the chemical loss term in the scalar budget equation.

It is thus crucial to measure the lifetime of $NO_3$ in future studies that analyze the NBL ozone or $O_x$ budget. We also suggest more direct measurements of aerodynamic resistance and ozone deposition at the surface by eddy covariance in conjunction with future airborne studies.

**Data Availability**

All of the aircraft data used in this analysis can be found at
https://www.esrl.noaa.gov/csd/groups/csd3/measurements/cabots/

**Author Contribution**

IF designed the research study and DC, JS, and SC carried it out. DC and SC designed the scalar budget code and DC carried out the analysis. Other analyses were performed by DC, IF, JS, NF, and JT. DC prepared and submitted the manuscript.

**Competing Interests**

The authors declare that they have no conflicts of interest.

**Acknowledgements**

This project was supported by a grant sponsored by the California Air Resource Board Research Division Contract No. 14-308. We thank Steven S. Brown, William P. Dube, and Nick Wagner of the NOAA Chemical Sciences
Division, Allen Goldstein of the University of California at Berkeley, and Drew Gentner of Yale University for freely sharing their data from the CalNex mission. We also thank Laura Bianco of the NOAA Physical Sciences Division for sharing the Chowchilla boundary layer height data. I. Faloona was supported in part by the California Agricultural Experiment Station, Hatch project CA-D-LAW-2229-H.

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
