# Peer review of "Residual Layer Ozone, Mixing, and the Nocturnal Jet in California's San Joaquin Valley"

_Atmospheric Chemistry and Physics, 2018_

## Referee Comment (RC1) · Anonymous Referee #1 · 9 Oct 2018

Caputi et al. is a thorough and well-designed analysis of the relationship between nocturnal turbulence and next-day ground-level ozone concentration in the southern San Joaquin Valley in California, one of the most polluted parts of the United States. This study presents much needed constraints on this relationship with a variety of observations - nocturnal airborne measurements, surface monitoring and sounder data, and reanalysis data. The authors construct an ozone (or Ox) budget for the nocturnal boundary layer using airborne observations, and use this to infer turbulent diffusivity. They explore some of the implications of their assumptions in constructing that budget (i.e., nitrate chemistry, the effects of recirculation on advection), and perform an error analysis of the budget, clearly stating that their calculations are quite uncertain.

Their budget-derived estimates of turbulent diffusivity are on the higher end of the

few existing previous estimates for other parts of the world, atmosphere, &/or time of day, but largely within the bounds of previous estimates. The authors speculate that this could be due to the unique topography of the region. The authors also calculate mean eddy diffusivity independently from the Ox budget using TKE dissipation and the Brunt-Vaisala frequency. They do find that this other estimate suggests a much smaller mean turbulent diffusivity, but don't really discuss the implications of this finding. Their budget-inferred eddy diffusivities largely correlate to estimates of the bulk Richardson number, which is expected and builds confidence in at least the estimates' variation.

The authors find a correlation between nocturnal boundary layer eddy diffusivity and next day ozone concentration using airborne measurements and regional mean surface network observations, respectively. As this correlation is only based on 12 data points, they leverage observations from a long-term monitoring site in Visalia, CA that both a sounder and an ozone sensor. The authors find a correlation between overnight maximum lower-level jet speed and next-day afternoon ozone concentration at Visalia, building further support for their hypothesis that strong nocturnal turbulence influenced by the low-level jet depletes ambient nocturnal ozone, and leads to cleaner next-day conditions. The authors also suggest that more efficient nocturnal ozone dry deposition under strong turbulence further acts to deplete nocturnal ozone.

Although I do think this paper merits publication in ACP, I would like to see substantial revisions before publication. However, these revisions for the most part have to do with improving structure and clarity of the manuscript. As is, the paper pretty severely lacks cohesion. I found it challenging to understand the goals, results, and implications of most sections. The abstract, introduction, and conclusion focus on the relationship between nocturnal turbulence and next day ozone, but there is quite a lot of supplemental analysis investigating the assumptions going into the Ox budget calculation, the uncertainties with respect to the inferred eddy diffusivity, etc. These parts could be much better integrated with the rest of the text. A clear articulation of the goals of each section at the beginning of the section, and a more detailed roadmap of the investigation in the introduction could be helpful. I like that the results and discussion are combined, but in many sections there is no discussion of the implications of the results, and they are not discussed in the conclusions. There are several figures that are barely discussed and I urge the authors to reconsider whether they should be included in the paper. The paper would strongly benefit from a streamlining of the analysis.

Also, the order of the figures should be the order that they are mentioned in the text. I would also like to see more explicit referencing in the text to the figure that is being discussed. The figures could also be more publication-quality. I would like to see the information on the plots themselves also included in captions, acronyms spelled out in captions, explicit references to the data sources in the captions. For most of the plots, I would also like to see larger text and larger symbols. The jet color scheme used on many figures is hard to interpret. For the maps, I find the underlying image of topography distracting and not helpful. If the authors really want to show the topography then one map of only the topography would be sufficient.

For the correlation between afternoon maximum hourly ozone and overnight maximum lower level jet speed at Visalia (Figure 14), the correlation is quite low. The fitted line doesn't look like it's capturing the pattern well, and looks like it's strongly influenced by the one really high jet speed. If this data point is removed, how does the relationship change?

—- Line-by-line comments.

Lines 56-57: Will the authors please include the point about dry deposition in a separate sentence? Also, the way the part about deposition is phrased too much does not really suggest that there is much uncertainty to this estimate, but there it is quite uncertain (see comments below)

Line 58: Would "more" be better than "stronger" here?

Line 63: "infer" instead of "measure"

[Figure]

Line 73: I find "occasion" as a verb to be non-intuitive

Lines 96-99: Will the authors refer to the stable layer as the NBL for consistency? This part is quite dense, especially for readers not fluent in boundary layer meteorology

Lines 101-104: I'm not seeing why the last two sentences are needed here. I would urge the authors to be as concise as possible here, again for readers not as fluent in BL meteorology

Line 110: Replace the "is" in "is important" to "is likely important". Also, both is plural, so "is" should be "are"

Lines 112- 128: I struggled with this paragraph, which feels out of place. It's not clear why the authors start to talk about the Fresno Eddy. One option would be to move this paragraph to Section 3.3. Another option would be to more clearly direct the reader as to why they are introducing it (i.e. that it challenges their analysis). Also, will the authors please briefly introduce ozone production potential?

Lines 129-140: I find this paragraph a bit awkward, especially the first sentence with the term "acknowledge". It seems like this sentence should be followed by a discussion of assumptions made, but this does not seem to be the case. The authors then proceed to mostly talk about daytime conditions, then say nitrate chemistry and dry deposition cannot be ignored. Why even talk about daytime? I would suggest saying that the focus of this work is nighttime and previous work has focused on daytime. The discussion of daytime doesn't feel meaningful, and it's confusing for the reader. Also, I'm confused about the point of mentioning nitrate chemistry and dry deposition here in this way. Do the authors examine these processes in detail later on? Perhaps framing it like that would help.

Line 152: Does "this ozone difference" refer to the day-to-day difference in ozone concentration? Please specify

Line 157: Do the authors average over a large area? The limitations would only be

overcome if so, right?

Line 161: Does "in this area" refer to Taiwan, or SSJV?

In general, the introduction is pretty dense. I feel that most readers would find some sort of schematic useful.

Lines 194-196: Do the authors think that their "somewhat arbitrary" cutoff has a substantial influence on their results?

Lines 199-204: Again, do the authors think that this assumption has a substantial influence on results?

Lines 241-242: Seems like this sentence is unnecessary

Line 247: Please cut "tracked by", it's confusing. The ultimate fate of nitrate? Please specify

Line 259: Please specify the field site and time examined in Padro 1996

Line 271: "A blend of these three methods" is too vague. Please specify the method

Line 290: Do the authors mean that NO2 and O3 are by far the dominant species of Ox? Please specify

Lines 319-386: This is a lot of information. I found this section very confusing and long-winded. Will the authors please break this paragraph up? It would be helpful if the authors stated the goal of this analysis upfront and more clearly stated what the assumptions are, the bases for making them, and how they feed into calculating the net reaction of R1-R6 as a constant multiple of R2.

Lines 323-234: But the authors just said that their airborne measurements are supported by the ground-level measurement network? What is the measurement network used? Do the authors not trust that it provides values that should be regionally representative?

Lines 327: This "channel of NO3" meaning R6?

Line 330: What are the "VOC reactions in our analysis"? So does this finding mean that the authors ignore R6?

Lines 319-330: So what's the conclusion here? It looks like the authors are finding a basis for including R6, but also a basis for not including R6.

Line 344: "Out of respect for" should be "Based on"

Line 348: Can "channel" be "reaction"? I find "channel" confusing and a bit colloquial

Line 352: Why is temperature shown in Figure 5 if it is not discussed? Also, in the caption of Figure 5 the acronyms of the airports should be spelled out.

Table 2: What do the authors mean that values may not match literature values? How is the extrapolation and valley average done? It seems like this info should be somewhere in the paper or supplementary material.

Lines 390: Will the authors better explain what the linear regression here is for, and how it is done?

Line 403: What is the similar environment? Please specify

Line 404: I don't think the authors have specified yet that the SSJV is an agricultural region.

Line 405: What's the basis of using these papers, over other ozone deposition papers? Half of these papers are not listed in the references list. There are also additional papers on CODE (California Ozone Deposition Experiment) that the authors may find helpful - for example, Massman 1994, Padro et al. 1994, Grantz et al. 1997. The authors should specify whether they are looking at an average of the CODE sites, or one in particular (there is a vineyard, cotton field, ...).

Line 409: Will the authors at least spell out that 2.5 cm/s is likely much higher than the

deposition velocity for NO2 should be, and perhaps cite some previous work here?

Line 410: Is the vertical flux divergence used in the last term or the last two terms?

Lines 412-3: Will the authors better explain what the linear regression here is for, and how it is done?

Lines 423-4: By surplus of Ox do the authors mean where Ox indicated by the purple line is greater than Ox indicated by the black line? Please specify this. Also please specify in the caption which of the terms have been inferred (and refer to section on calculation) and which have been observed.

Line 429: How is the error propagation calculated? At least refer to Section 3.2

Table 3: What exactly is the error estimate? At least refer to Section 3.2

Line 433: Please cut "Another way to frame . . . NBL"

Line 434: Please cut "Further". (In my opinion, doing this and the above suggestion would make this part more digestible).

Line 438: Do the authors mean NO2 is less than O3 by 10-20 ppb here?

Line 445: There should be an introductory sentence here, instead of starting with a specific component's error calculation.

Line 455-6: I would cut the term "conservative". What basis do the authors have for this value judgement? It seems little, especially in terms of the ozone deposition velocity

Section 3.3: This section is confusing because the authors say that the presence of Fresno Eddy could be problematic for their analysis. Then, they say that the predominant circulation during their flights is similar to Fresno Eddy, but then they say any recirculation has a minimal impact on their results (lines 492-3). A lot of the analysis on Fresno Eddy could be cut. . .especially because it's found to be irrelevant. This would help with clarity and flow. Additionally, can the authors split Section 3.3 in two?

One section on Fresno Eddy, and one on the low-level jet?

Lines 468-72: Are Zhong et al. 2004 describing the Fresno Eddy conditions, or other prevailing conditions? Please specify.

Line 473: The authors need to more clearly specify that they are suggesting there are Fresno Eddy conditions during their flights.

Lines 480-2: I don't really know what the takeaway here is.

Figure 7: What is shown in the background of the plots? It's hard to see the yellow and light blue colors on top of the grey. I recommend using a different color scheme and/or thicker lines.

Line 494: I would repeat the hypothesis more in full here (i.e., the effects of the nocturnal jet on the next day's ozone levels; "contribute to the variability of ozone" is a bit vague).

Line 494-5: Again, "explored some of the meteorological factors that are absent from the current literature" is vague. Further, why would the authors only explore unexplored factors?

Line 498: "in 100m bin space" is too colloquial

Lines 506-525: This paragraph is confusing. The authors should state up front what they are investigating here.

Line 506: Explicitly say which thresholds correspond to strong and weak jets

Line 506: What is "it" here? The trough? Please specify

Line 512: Why the mention of Fresno Eddy here? Are the authors trying to attribute eastward trough to Fresno Eddy not happening? Please clarify

Line 516: What are "those" conditions?

Lines 516-526: It seems like this should be a paragraph on it's own, and better linked

with the mention around Line 512 of Fresno Eddy. Referring to "LLJ" generally in this paragraph here is particularly confusing because in the preceding lines the authors were talking about weak vs. strong LLJ.

Lines 522-527: I'm not exactly sure why the authors feel the need to compromise here.

Lines 523-524: Previously the authors had said the Fresno Eddy and the LLJ are not the same thing, but here the authors seem to be referring to them interchangeably.

Lines 527: What is in addition to synoptic forcing?

Lines 532: High temperature could also decrease deposition through stomatal pores

Line 534: -> With the NARR climatology.

Figure 9: A map showing the difference in 2m air temperature for stronger vs. weaker LLJ may be more effective. Hard to see the contours. Or maybe just cut the elevation map, and color by temperature contours.

Figure 11 is never referenced, but I think it should be on Line 545. Figure 11 is interesting, but very tangential, and I think the figure and the short discussion of it should be cut.

Line 551: "Another look at . . ." is not a very helpful way of introducing what the authors are doing here. What are the authors trying to investigate here? Also, what is overnight layering?

Section 3.4 What's the rationale for including the discussion of Figure 12 in the previous section, as opposed to at the beginning of this one? Seems like it would flow better in Section 3.4.

Line 562: "several previous studies examining different parts of the world"

Line 567: Will the author please make it more clear that their hypothesis is stated on lines 564-5? Line 566: Specify regional mean ozone from monitoring stations in a

certain network Lines 568: Here are the authors examining ozone at the monitoring stations or measured on the aircraft? Please specify Line 574: Why would the relationship be strongest for MDA8? How much stronger is the relationship for MDA8 vs. max hourly, 24 hour average? If it's a lot stronger, is MDA8 roughly representing ozone at the same hours each day? Examining this could be insightful.

Also, why is this relationship stronger for MDA8 than that observed during the fumigation period?

Line 578: It would help the reader to briefly restate the hypothesis. Lines 580-3: Wait, why not MDA8 here?

Figure 13 and 14: Please be consistent in terms of ozone on the y vs. x axis.

Lines 593-5: Why would Rb be 0 at night? This doesn't make much sense to me. Is this stated in the Padro 1996? Rb is not included in Padro 1996 Figure 4. In Massman [1994] Rb is estimated to be nonzero for the CODE vineyard. I recommend specifying that not only Ra is modeled in Massman [1994] but Rc is too (it's not a residual of observed vd and estimated Ra and Rb). Then I might just say here that modeled Ra and Rc are similar at night and Rb is unknown, rather than zero. It's also important to note that this is only one way of estimating Ra ($u/u\_*^2$) and estimates at night are likely highly uncertain. Lines 600-3: How would taking changes in Ra into account in the budget calculation change the eddy diffusivity estimate?

Section 3.5: It would be helpful if the authors introduced the goal of their analysis in this section upfront.

Line 607: Why should the authors values be comparable to Banta et al. 2006 and Lenschow et al. 1988? Please specify. Line 610: Did Banta et al. try to remove buoyancy waves? Line 610-1: Why? What is the implication of this finding? Line 624: "lower end of the range inferred from the Ox budget". It would be helpful here if the authors re-stated the range of eddy diffusivities that they infer from the Ox budget.

Line 626: "our estimates inferred from the Ox budget" Line 631: "similar turbulent environment to ours" ? Line 634: Specify here that the Lenschow et al. 1988 eddy diffusivity from the lower half of the NBL is the most comparable Lines 645-9: To me flow is better if the order of these two sentences is flipped Line 636: "variability in the reported values" Lines 640-4: What's the point of this analysis? Because this relationship is expected, does this build confidence in the authors' estimate of eddy diffusivity (at least the variability in eddy diffusivity)? If so, it should be explicitly stated. Lines 658-9: Briefly, why would the unstable layers have to extend upward beyond the NBL depth? Line 659-60: Why is this more likely? What's the implication of this? Lines 663-4: Briefly, how would they contribute to overnight mixing?

Figure 16: Why is only 50 m shown? The authors say they examine thickness of 50m and 100m. It is challenging to interpret this plot. Another color scheme, and a zoomed-in map would be better. Also, the font size should be increased. It would be helpful to indicate the location of the Tehachapi pass on the map.

Lines 668-9: Where is this shown? Also, "seen"-> "observed" Line 669: What finding? Line 674-5: It might be more clear to state that the figure does not support the hypothesis that the authors outlined on lines 671-2. Also where is this figure? It would be helpful if the authors specified that it is not shown. Line 675-6: How does this fit into the above discussion? What are the implications of this finding?

Line 687: Cut "slightly" Line 689: "A limitation of our study" Line 690: Cut "being conducted". Also what do the authors mean by pairs? Do they mean morning and evening flights? I would specify this. "pairs" is non-intuitive. Line 690: What demonstrates? Specify what "it" is. Line 691: Seems strange to mention that the authors demonstrate something "within the context of high ozone episodes" when ozone hasn't been mentioned yet in the conclusion. On a similar note, the authors haven't noted in the conclusion that there was a particular focus strategy of the flights, so it's strange to mention it. It's helpful for the reader if the conclusion can really stand alone from the rest of the text. Line 692: Specify where the soundings and surface monitoring data

are from (locations, networks) here Line 692-3: Specify the implication of this finding (tie back to hypothesis) Line 694: What do the authors mean "although in the former analysis"? In the analysis of soundings and surface network data? This could be more clearly articulated, and it should be directly stated that this is not found in the airborne measurements. Line 695-6: "is an important link that may have consequential implications for modeling studies and policy making" is vague and verbose. I think the authors' findings are important for modeling and policy, but this sentence doesn't do much to convince me of it. Line 697: Introduce Visalia Line 698: "infer" -> "determine" Line 701: Spell out that reduced aerodynamic resistance means more efficient transport to surfaces where ozone can deposit Line 704: It would be good to articulate that this may be why the correlation between night turbulence + next day ozone may not always be high. Line 704: "Airborne measurements from flights over Bakersfield, CA showed . . ." Lines 704-6: Spell out the implication of this finding Line 706: In what study? Trousdell et al. 2016? If so, the subject should not be "we", it should be "they" or better, Trousdell et al. (2016) Lines 704-10: I'm not quite following why the discussion of Trousdell et al. 2016 is relevant for the conclusions of this paper. Lines 711-2: "illustrated"-> "suggested"; "which consequently has impacts for"-> "and thus likely impacts" Lines 712-5: But what exactly is so uncertain about nitrate, and why will it affect ozone? There should be a line stating that the authors haven't measured nitrate on their flights, and how/why this leads to uncertainty in their analysis. The authors should re-introduce alpha, and why it's important. I really like how the authors have spelled out that nitrate measurements (specifically the lifetime) are needed in future nocturnal airborne measurement campaigns. Are there any other measurements or techniques that their analysis suggests doing or developing would reduce uncertainty?

---

## Referee Comment (RC2) · Anonymous Referee #2 · 23 Oct 2018

Review of Caputi et al, 2018

The authors hypothesize that a strong low-level nighttime jet more effectively mixes down ozone into the stable nighttime boundary layer where it deposits, resulting in lower ozone the next day. On nights with a weaker jet, the residual layer remains decoupled and results in higher ozone the next day. This paper introduces methods for developing nocturnal scalar budgets from aircraft observations. This hypothesis and support from aircraft contributions is a useful contribution to our understanding of the effect of weather patterns on ozone. One general comment is that the authors could better motivate their statement that air quality models need to better forecast this feature with a brief overview of the current ability of models to simulate the nocturnal low-level jet. Generally the paper would also benefit from clearer presentation of the methodology and results including checking for consistent use of terms and figure referencing. The authors seem to discuss methodology and results intermixed in multiple locations, and a more coherent progression of methodology and results would both shorten and clarify the author's hypothesis and findings. This paper should be published in ACP after addressing these revisions and the comments below.

Specific comments:
1. Page 3, line 125. I don't understand whether the authors are using the Beaver and Palazoglu (2009) paper to support their hypothesis. They initially say that a strong nighttime LLJ reduces ozone the next day, so how does this reconcile with strong nighttime ventilation resulting in high ozone the next day?
2. Page 8, line 259. Can you comment on the validity of using 0.2 cm/s for the ozone dry deposition velocity when you argue that deposition will be enhanced when the nighttime LLJ is strong?
3. Page 14, line 376. Please discuss where the uncertainty on the 1.5 value comes from\, this is unclear from the previous discussion.
4. Page 15, line 410. Do you mean the last term (not the last two)?
5. Page 15, line 423. Please use consistent language to avoid confusion. Do you mean the surplus of Ox observed on the morning flight is inferred to be driven by the "advection" term?
6. Page 16, Table 3. You haven't actually explicitly described yet as far as I can tell the procedure for calculating the storage term.
7. Page 16, line 435. Please clarify which term refers to the observed time rate of change, since no term appears to be double the sum of chemical loss and deposition.
8. Page 17, line 445. How is the error in the nocturnal PBL height included in the error analysis?
9. Page 17, line 477. It would be useful to show on the figure the extent of the SSJV and point out the nocturnal low-level jet.
10. Page 18, line 501. Give the corresponding PST, since that is what is stated in the abstract.
11. Page 18, line 504. I assume you are referring to Figures 8-9 here for the daily average synoptic charts? If so, please add this to the text.

12. Page 19, line 506.  Could you again highlight this offset of the figures so that the reader can easily pick out this 100 km difference?

13. Page 19, line 510.  You don't need Figure 10, just tell us the correlation coefficient and p-value.

14. Page 19, line 531.  Why would higher temperatures increase photochemical production at night?  What about higher temperatures increasing soil NOx? Also, you haven't mentioned PAN at all – wouldn't higher temperatures result in more PAN decomposition to increase Ox?

15. Figure 11 – please explain the legend in the figure caption.

16. Page 22, line 555.  Is there an explanation for the different behavior of TKE here?

17. Page 24, line 602. Can you clarify whether you are still considering a high canopy resistance in the vd calculation?

18. Figure 16 – The font size here is difficult to read.

19. One general comment – the use of eddy diffusivity to evaluate turbulent mixing is most generally applicable to the daytime convective mixed layer.  The authors could better support why this framework is applicable to the stable nocturnal boundary layer. This is better done in 3.5, so possibly referencing this section earlier on would be useful.

---

## Author Comment (AC1) · 12 Nov 2018

We would like to thank reviewer #1 for providing insightful comments on our submission to ACP. As stated in our response to reviewer #2, we are aiming to better partition the methodology and results. We are also making some revisions such that the goals of the study are articulated towards the beginning.

Most of the figures are being adjusted to improve quality and legibility. In particular, we plan to adjust the terrain to be greyscale, higher resolution, and less distracting. However, we believe the terrain in each figure is important because flow characteristics in California are highly influenced by it. We will use larger text for fonts, as requested.

We would like to report that removing the single outlier point of the MDA8 vs. LLJ correlation does not substantially affect the results ($r^2 = 0.167$).

Line-by-line comments:

Lines 56-57: Done.

Line 58: Done.

Line 63: Done.

Line 73: Changed "occasion" to "are associated with".

Lines 96-97: Done.

Lines 101-104: We have removed these last two sentences.

Line 110: Done.

Lines 112-128: Added "The complex nocturnal wind patterns in the SSJV contribute to the challenges of understanding and forecasting ozone pollution in our study region" to the beginning of the paragraph. Also changed "ozone pollution potential" to simply "ozone pollution".

Lines 129-140: Here we are discussing the context of the scalar budget equation in general, although I do understand why discussing daytime studies in detail might be confusing. The discussion about advection is relevant for both daytime and nighttime scalar budgets, but we changed the sentence regarding daytime photochemical production to "Studies performing daytime scalar budgets of ozone (Conley et al., 2011; Lehning et al., 1998; Lenschow et al., 1981; Trousdell et al., 2016) have shown that chemical production is important, and similarly, we expect the chemical loss of ozone to be important at night."

Line 152: Yes, changed to "the aforementioned ozone difference".

Line 157: The scalar budget technique we present covers a large swath of the SSJV, and thus the terms in the budget equation can be taken as averages of the entire region for which the budget is performed.

Line 161: Changed to "ozone problems in southern Taiwan"

Lines 194-196: Changing the cutoff will result in different TKE values, but the night to night variability should not be affected by this. The TKE analysis is mostly supplemental to the main thesis and would not change our conclusions. This is an issue that arises in any stable boundary layer study.

Lines 199-204: The similarity relationships are employed as a best approximation and we acknowledge that the uncertainty in our TKE estimates are high. Again, we do not use the TKE estimates for anything critical to our conclusions.

Lines 241-242: Removed.

Line 247: Changed to "then computed by the reaction …, and the ultimate fate of nitrate will affect…"

Line 259: Done.

Line 271: Changed to "all three of these methods were used in tandem."

Line 290: Changed to "as these are by far the dominant species of $O_x$."

Lines 319-386: To clarify the aim of this paragraph, we added "Thus, determining the dominant loss of nitrate is crucial for our analysis" to the end of the previous paragraph (line 318). We started a new paragraph on line 327 ("There is a further question), 347 ("With longer lifetimes"), and 355 ("Given the obvious").

Lines 323-324: Changed to "However, both the ground network and aircraft observations may be biased high to the regional average because of their proximity to…"

Line 327: Yes.

Line 330: We have clarified that we mean VOC reactions in Table 2. For these calculations, we are only considering the VOC channel of nitrate loss (R5) in order to answer the question of whether or not R5 is important.

Lines 319-330: As stated in line 355, we conclude that R6 should not be ignored. We added this statement in line 330 for clarity.

Line 344: Done.

Line 348: Done.

Line 352: The temperatures of Figure 5 are later referenced in lines 480-482.

Table 2: We found that often, the measurements in the studies were taken in specific areas such as crop fields. Since the aim of this analysis was merely to get a reasonable estimate, we used our meteorological knowledge to estimate whether a valley-averaged concentration may be slightly higher or lower than what was reported in the study. We added this to the text.

Line 390: It is our opinion that the linear regression was concisely summarized here.

Line 403: Specified that this study was done in a flat grass field.

Line 404: Replaced "these agricultural regions" with "the SSJV".

Line 405: Corrected the reference list to include Meszaros et al. (2009) and Pederson et al. (1995). We found the Lin et al. (2010) reference to be the most helpful in that it summarized past estimates in Table 3, and it specifically focused on nocturnal dry deposition values.

Line 409: Done.

Line 410: Discussed in our response to reviewer #2.

Lines 412-413: Changed to "A linear regression through the 20 m resolution vertical $O_x$ profile is used to determine $dOx/dz$ (for the last term in equation 1) in the upper…"

Lines 423-424: Discussed in our response to reviewer #2.

Line 429: Done.

Table 3: Done.

Line 433: Done.

Line 434: Done.

Line 438: Yes. Changed to "$O_3$ is less than $NO_2$ by…"

Line 445: Added "Here we estimate the uncertainty for each term in the budget equation, as well as the ultimately calculated eddy diffusivities." as an introductory sentence.

Line 455-456: Done.

Section 3.3: As addressed in some of the following comments, we have attempted to clarify our discussion of the Fresno Eddy and where it fits in to this work. We firmly believe that a clear discussion of the Fresno Eddy is absolutely necessary to retain because it is constantly referred to in air quality discussions of the SJV, but not clearly understood. It is a major conclusion of the paper that we sample and describe the Fresno Eddy in a new and better way, which we believe can help illuminate future studies. We have tried to clarify the discussion where possible, but maintain that the low-level jet is *part and parcel* of the Fresno Eddy, therefore separating the two into distinct sections in the manuscript only perpetuates the misleading distinction.

Lines 468-472: Changed to "Zhong et al. (2004) uses a series of 915 MHz radio acoustic sounding systems to analyze low-level winds in the SSJV. Their Figure 4 shows that at night, …"

Line 473: Changed to "…observations, suggesting the presence of a Fresno eddy during our flights."

Lines 480-482: Here we are stating that Zhong et al. (2004) was presenting a climatological analysis of typical summertime conditions, while our flights were targeting periods of higher ozone, thus the synoptic and mesoscale conditions during our flights might be systematically different from climatological norms.

Figure 7: The color scheme used was the best one we could find in terms of readability. However, we have increased the resolution so that the arrows stand out better.

Lines 494-495: changed to "…variability of maximum daytime ozone concentration, we explored the synoptic patterns that are associated with differing strengths of the LLJ".

Line 498: Changed to "averaged in 100 m vertical bins,…"

Moved "To analyze variability … (N=165 nights)" to the first sentence of the following paragraph (line 506) for better flow.

Line 506: Yes, corrected in text.

Line 512: Added "(and thus LLJ)" after Fresno eddy mention.

Line 516: Changed to "those synoptic conditions".

Lines 516-526: We have made this a separate paragraph.

Lines 522-527: Changed to "As a provisional synthesis of these seemingly conflicting findings"

Lines 523-524: We are suggesting that the LLJ is the strongest branch of a Fresno eddy, thus a strong eddy will produce a strong LLJ. We have attempted to clarify this in the text.

Line 527: Changed to "In addition to the synoptic patterns discussed above"

Line 532: Thank you for pointing this out. We have added this.

Line 534: Done.

Figure 11: Removed along with the discussion of it.

Line 551: Section 3.4 now starts here. We removed figure 12 and instead added the TKE profile to figure 4, and reference that here.

Line 562: Done.

Line 567: Changed "the relationship" to "this relationship" for better flow.

Line 566: Done.

Line 574: Jin et al., JGR, 2013 suggests that the MDA8 occurs fairly consistently between 13 and 14 PST. For the 24 hour average ozone correlation with eddy diffusivity, $r^2 = 0.40$. I believe that the relationship is notably weaker for the fumigation periods due to slight variations in timing of the peak boundary layer growth rate.

Line 578: Changed to "Because this analysis consisted of only 12 flights, we decided to explore a larger data set that might support the hypothesis that a stronger LLJ reduces ozone the following day."

Lines 580-583: The relationship with MDA8 was slightly weaker. We have added this in the text so that we are not biasing our results to only showcase the best correlations.

Figure 13 and 14: Done.

Added suggested literature and stated that $r_b$ is unknown and thus not included in this approximation. The average error of $K_z$ due to the uncertainty of $V_d$ is calculated to be ~0.50 m$^2$ s$^{-1}$, which is included in the original error propagation analysis.

Section 3.5: Done.

Line 607: Specified that these are studies of NBL turbulence.

Line 610: Banta et al. (2006) is a meta analysis of other studies. To the best of my knowledge, buoyancy waves were not removed.

Lines 610-611: While we were hoping that our TKE would have a relationship with ozone the following day, it is a very noisy measurement and we were also using many approximations to estimate it, as outlined in the paper.

Line 624: Done.

Line 626: Done.

Line 631: Done.

Line 634: Done.

Lines 645-649: Disagree with reviewer here because subjective turbulence should be mentioned before delving into it.

Line 636: Done.

Lines 640-644: Yes, the point of this analysis was to build confidence of our eddy diffusivity measurements. We have clarified this in the text.

Lines 658-659: changed to "as the unstable layers appear to be above the NBL where there is communication with the surface."

Lines 659-660: We are stating that although unstable layers are observed more frequently in urban areas compared to rural areas, we may have simply detected them more often there because the aircraft spends more time in urban areas. Hence, the apparent pattern of more unstable layers in urban areas could be insignificant.

Lines 663-664: Absolutely unstable layers in the atmosphere promote the production of turbulence and thus vertical mixing.

Figure 16: We are making the requested adjustments to the figure. Only 50 m is shown in order to reduce the number of figures in this submission, and we did not believe the 100 m thickness plot added anything particularly useful.

Lines 668-669: We did not include this figure for sake of brevity.

Line 669: We are referring to the finding stated in the previous sentence. "Finding" has been removed for better flow.

Lines 674-675: Done.

Lines 675-676: This fits into the above discussion because we are showing the unstable layers appearing in the climatological averages of the 915 MHz profiler. The implications of this are that it lends some additional credibility to their existence.

Line 687: Done.

Line 689: Done.

Line 690: Yes and done. Also changed to "the study demonstrates".

Line 691: Focus strategy of the flight restated in conclusion.

Line 692: Done.

Lines 692-693: Done.

Line 694: Changed sentence to "Similarly, the correlations between the aircraft-estimated eddy diffusivities and MDA8 the following day also suggest that vertical mixing in the NBL plays an important role in determining ozone concentrations."

Lines 695-696: Adding a brief discussion of specific modeling and policy implications.

Line 697: Specified Visalia, CA.

Line 698: Done.

Line 701: Done.

Line 704: Done.

Lines 704-706: We were mainly pointing this out to remind the reader that even though the advection term on average tends to be near zero, it can be large for any particular data point.

Line 706: Done.

Lines 706-710: We are reminding the reader that there is more to the picture than just vertical mixing of ozone at night, since afternoon ozone concentrations are influenced by advection and photochemical production.

Lines 711-712: Done.

Lines 712-715: We have followed these suggestions and are also stating that deposition velocity measurements of ozone using eddy covariance on future campaigns would be helpful.

---

## Author Comment (AC2) · 12 Nov 2018

We would like to thank the reviewer for the useful feedback regarding our submission to ACP, and their recommendation for publication. We agree with the reviewer's opinion that we can better partition the methodology and results of this study. As requested, we will also be glad to include a brief summary of air forecasting models in the literature review.

Specific comments:

1. Beaver and Palazoglu (2009) point to the recirculation effects of the Fresno eddy, which as we state later, may appear to conflict with our hypothesis rather than support it. We are clarifying this aspect in the literature review.
2. We estimate that 0.2 cm s$^{-1}$ is an average value of ozone dry deposition at night in our region, and our stated error is 50%. Thus, the estimated variation due to changes in jet strength (~40%) is within our envelope of uncertainty.
3. The uncertainty of this coefficient is discussed in section 3.2 (lines 450-457). However, we will allude to this when the coefficient is first introduced.
4. While the last term represents the flux at the top of the NBL, the second to last term represents the surface flux. Thus, the flux divergence in the vertical direction is represented by the last 2 terms in equation 1.
5. The surplus of $O_x$ refers to the difference between the projected $O_x$ if there were only chemistry and advection at play, and the actual observed morning $O_x$. Since chemistry and advection has been modeled in this figure, we assume the difference between projected and observed is due to vertical mixing. We will clarify this in the text.
6. $dO_x/dt$ is calculated from the aircraft profile difference between the late night and sunrise flights. We will specify this in the text.
7. Thank you for catching this – this was misstated. We have changed the text to "Further, both losses of Ox added together are about triple the observed time rate of change, and thus the physical and chemical losses are largely (2/3rds) compensated by vertical mixing."
8. The error in the NBL height is included in the error propagation analysis for the eddy diffusivities.
9. We are extending the image further to the south to show the full SSJV. However, as the figure mainly focuses on observations, we avoid adding cartoon schematics of the mesoscale features, which are not fully known.
10. Done.
11. Done.
12. We have drawn a dotted dashed line in the figures to indicate the mean trough axis.
13. Done.
14. Here we are arguing that greater daytime photochemical rates (including those due to increased PAN dissociation) during warmer synoptic periods might be an additional factor that increases surface ozone. This would act in addition to less nocturnal mixing (due to the synoptic conditions favoring high temperatures making a weaker LLJ).
15. We have removed this figure from the manuscript in response to another reviewer, as it is not central to our thesis.
16. This is likely due to there being less shear immediately under the jet compared to the amount of shear in the surface layer.
17. We are assuming that the canopy resistance does not change.

18. We have increased the font size and sharpened the terrain in Figure 16.
19. Eddy diffusivities were the most practical way of estimating the NBL mixing due to the logistics of our study. We have changed the wording in section 3.5 earlier in the paper to make this clear earlier on, as requested.

---

## Author Response (AR1)

**Reviewer #1 Response**

Although I do think this paper merits publication in ACP, I would like to see substantial revisions before publication. However, these revisions for the most part have to do with improving structure and clarity of the manuscript. As is, the paper pretty severely lacks cohesion. I found it challenging to understand the goals, results, and implications of most sections. The abstract, introduction, and conclusion focus on the relationship between nocturnal turbulence and next day ozone, but there is quite a lot of supplemental analysis investigating the assumptions going into the Ox budget calculation, the uncertainties with respect to the inferred eddy diffusivity, etc. These parts could be much better integrated with the rest of the text. A clear articulation of the goals of each section at the beginning of the section, and a more detailed roadmap of the investigation in the introduction could be helpful. I like that the results and discussion are combined, but in many sections there is no discussion of the implications of the results, and they are not discussed in the conclusions. There are several figures that are barely discussed and I urge the authors to reconsider whether they should be included in the paper. The paper would strongly benefit from a streamlining of the analysis

Response: We would like to thank reviewer #1 for providing insightful comments on our submission to ACP. As stated in our response to reviewer #2, we have attempted to better partition the methodology and results. We have also made some revisions such that the goals of the study are articulated towards the beginning.

Most of the figures have been adjusted to improve quality and legibility. In particular, we changed the terrain to be greyscale, higher resolution, and less distracting. However, we believe the terrain in each figure is important because flow characteristics in California are highly influenced by it. Larger text for fonts has been used, as requested.

We presented a new analysis of the MDA8 vs. LLJ correlation that removes the outlier point where LLJ > 25 m s$^{-1}$.

Line-by-line comments:

Lines 56-57: Will the authors please include the point about dry deposition in a separate sentence? Also, the way the part about deposition is phrased too much does not really suggest that there is much uncertainty to this estimate, but there it is quite uncertain (see comments below)

Response: changed to "Here we investigate the hypothesis that on nights with a strong low-level jet (LLJ) ozone in the residual layer is more effectively mixed down into the stable boundary layer. There it is subject to dry deposition to the surface, the rate of which is itself enhanced by the strength of the LLJ, resulting in lower ozone levels the following day"

Line 58: Would "more" be better than "stronger" here?
Line 63: "infer" instead of "measure"
Line 73: I find "occasion" as a verb to be non-intuitive

Lines 96-99: Will the authors refer to the stable layer as the NBL for consistency? This part is quite dense, especially for readers not fluent in boundary layer meteorology

Lines 101-104: I'm not seeing why the last two sentences are needed here. I would urge the authors to be as concise as possible here, again for readers not as fluent in BL meteorology

Line 110: Replace the "is" in "is important" to "is likely important". Also, both is plural, so "is" should be "are"

Response: Changed "occasion" to "are associated with" in line 73. All other semantic changes have been made. The two sentences from lines 101-104 have been removed and the stable layer is now referred to as the NBL.

Lines 112- 128: I struggled with this paragraph, which feels out of place. It's not clear why the authors start to talk about the Fresno Eddy. One option would be to move this paragraph to Section 3.3. Another option would be to more clearly direct the reader as to why they are introducing it (i.e. that it challenges their analysis). Also, will the authors please briefly introduce ozone production potential?

Response: Added "The complex nocturnal wind patterns in the SSJV contribute to the challenges of understanding and forecasting ozone pollution in our study region" to the beginning of the paragraph. Also changed "ozone pollution potential" to simply "ozone pollution".

Lines 129-140: I find this paragraph a bit awkward, especially the first sentence with the term "acknowledge". It seems like this sentence should be followed by a discussion of assumptions made, but this does not seem to be the case. The authors then proceed to mostly talk about daytime conditions, then say nitrate chemistry and dry deposition cannot be ignored. Why even talk about daytime? I would suggest saying that the focus of this work is nighttime and previous work has focused on daytime. The discussion of daytime doesn't feel meaningful, and it's confusing for the reader. Also, I'm confused about the point of mentioning nitrate chemistry and dry deposition here in this way. Do the authors examine these processes in detail later on? Perhaps framing it like that would help.

Response: Here we are discussing the context of the scalar budget equation in general, although I do understand why discussing daytime studies in detail might be confusing. The discussion about advection is relevant for both daytime and nighttime scalar budgets, but we changed the sentence regarding daytime photochemical production to "Studies performing daytime scalar budgets of ozone (Conley et al., 2011; Lehning et al., 1998; Lenschow et al., 1981; Trousdell et al., 2016) have shown that chemical production is important, and similarly, we expect the chemical loss of ozone to be important at night."

Line 152: Does "this ozone difference" refer to the day-to-day difference in ozone concentration? Please specify

Response: Yes, changed to "the aforementioned ozone difference".

Line 157: Do the authors average over a large area? The limitations would only be overcome if so, right?

Response: The scalar budget technique we present covers a large swath of the SSJV, and thus the terms in the budget equation can be taken as averages of the entire region for which the budget is performed.

Line 161: Does "in this area" refer to Taiwan, or SSJV?

Response: Changed to "ozone problems in southern Taiwan"

Lines 194-196: Do the authors think that their "somewhat arbitrary" cutoff has a substantial Influence on their results?

Response: Changing the cutoff will result in different TKE values, but the night to night variability should not be affected by this. The TKE analysis is mostly supplemental to the main thesis and would not change our conclusions. This is an issue that arises in any stable boundary layer study.

Lines 199-204: Again, do the authors think that this assumption has a substantial influence on results?

Response: The similarity relationships are employed as a best approximation and we acknowledge that the uncertainty in our TKE estimates are high. Again, we do not use the TKE estimates for anything critical to our conclusions.

Lines 241-242: Seems like this sentence is unnecessary

Response: Removed.

Line 247: Please cut "tracked by", it's confusing. The ultimate fate of nitrate? Please specify.

Response: Changed to "then computed by the reaction …, and the ultimate fate of nitrate will affect…"

Line 259: Please specify the field site and time examined in Padro 1996.

Response: changed to "Combining those measurements with an estimated 0.2 cm s$^{-1}$ nighttime dry deposition velocity of ozone at night in the SSJV (Padro, 1996), we can indirectly estimate $K_z$."
Line 271: "A blend of these three methods" is too vague. Please specify the method
Response: Changed to "all three of these methods were used in tandem."

Line 290: Do the authors mean that NO2 and O3 are by far the dominant species of Ox? Please specify

Response: Changed to "as these are by far the dominant species of $O_x$."

Lines 319-386: This is a lot of information. I found this section very confusing and long-winded. Will the authors please break this paragraph up? It would be helpful if the authors stated the goal of this analysis upfront and more clearly stated what the assumptions are, the bases for

making them, and how they feed into calculating the net reaction of R1-R6 as a constant multiple of R2.

Response: To clarify the aim of this paragraph, we added "Thus, determining the dominant loss of nitrate is crucial for our analysis" to the end of the previous paragraph (line 318). We started a new paragraph on line 327 ("There is a further question), 347 ("With longer lifetimes"), and 355 ("Given the obvious").

Lines 323-234: But the authors just said that their airborne measurements are supported by the ground-level measurement network? What is the measurement network used? Do the authors not trust that it provides values that should be regionally representative?

Response: Changed to "However, both the ground network and aircraft observations may be biased high to the regional average because of their proximity to…"

Lines 327: This "channel of NO3" meaning R6?

Response: Yes.

Line 330: What are the "VOC reactions in our analysis"? So does this finding mean that the authors ignore R6?

Response: Changed "our analysis" to "Table 2". For these calculations, we are only considering the VOC channel of nitrate loss (R5) in order to answer the question of whether or not R5 is important.

Lines 319-330: So what's the conclusion here? It looks like the authors are finding a basis for including R6, but also a basis for not including R6.

Response: R6 should be included. We have separated the paragraph that addresses R6 from the paragraph that addresses VOCs to avoid confusion.

Changes made:

Reaction (R6) has often been ignored at night under the presumption that local sources of NO are sparse and reaction (R1) will outcompete reaction (R6) (Brown et al., 2007; Stutz et al., 2010); however, at 30 ppb of $O_3$ and 20 ppt of $NO_3$ the lifetimes of NO to (R1) and (R6) are nearly equivalent (~80s). Our measurements indicate ground-level NO of about 0.6 ppb at midnight (SD = 1 ppb), corroborated by the surface air quality network, increasing in the early morning hours to 2-4 ppb. However, both the ground network and aircraft observations may be biased high to the regional average because of their proximity to California Highway 99 and other urban centers (Fig. 3). Nevertheless, the rate of reaction (R6) is 2.6 x 10-11 $cm^3$ $s^{-1}$ $molec^{-1}$ (Sander et al., 2006), extremely rapid relative to the others, such that even 60 ppt of NO, an order of magnitude lower than what our measurements indicate, results in an $NO_3$ lifetime of only 25 seconds. Hence, we conclude that (R6) should not be ignored.

There is then a further question as to whether any VOCs would be able to compete with this channel of NO3 consumption. An investigation into the faster VOC reactions with NO3 per Atkinson et al. (2006) and Gentner et al. (2014a) is presented in Table 2. The estimated lifetime of NO3 due to the VOC reactions in Table 2 is 9.5 seconds, about four times the lifetime of NO3 with respect to the presence of 0.6 ppb of NO (2.5 seconds)

Line 344: "Out of respect for" should be "Based on"
Line 348: Can "channel" be "reaction"? I find "channel" confusing and a bit colloquial
Line 352: Why is temperature shown in Figure 5 if it is not discussed? Also, in the caption of Figure 5 the acronyms of the airports should be spelled out.

Response: Semantic changes have been made. The temperatures of Figure 5 are later referenced in lines 480-482.

Table 2: What do the authors mean that values may not match literature values? How is the extrapolation and valley average done? It seems like this info should be somewhere in the paper or supplementary material.

Response: We found that often, the measurements in the studies were taken in specific areas such as crop fields. Since the aim of this analysis was merely to get a reasonable estimate, we used our meteorological knowledge to estimate whether a valley-averaged concentration may be slightly higher or lower than what was reported in the study.

Changes made:

The measurements in some of the studies above were taken in specific crop fields. Since the aim of this analysis was merely to obtain an order of magnitude estimate, we predicted whether a valley-averaged concentration may be slightly higher or lower than what was reported in the study. Thus, values here may not exactly match literature.

Lines 390: Will the authors better explain what the linear regression here is for, and how it is done?

Response: It is our opinion that the linear regression was concisely summarized here.

Line 403: What is the similar environment? Please specify

Response: Specified that this study was done in a flat grass field.

Line 404: I don't think the authors have specified yet that the SSJV is an agricultural region.

Response: Replaced "these agricultural regions" with "the SSJV".

Line 405: What's the basis of using these papers, over other ozone deposition papers? Half of these papers are not listed in the references list. There are also additional papers on CODE (California Ozone Deposition Experiment) that the authors may find helpful - for example, Massman 1994, Padro et al. 1994, Grantz et al. 1997. The authors should specify whether they are looking at an average of the CODE sites, or one in particular (there is a vineyard, cotton field, …)

Line 409: Will the authors at least spell out that 2.5 cm/s is likely much higher than the deposition velocity for NO2 should be, and perhaps cite some previous work here?

Response: Corrected the reference list to include Meszaros et al. (2009) and Pederson et al. (1995). We found the Lin et al. (2010) reference to be the most helpful in that it summarized past estimates in Table 3, and it specifically focused on nocturnal dry deposition values.

Changes made:

Results from a European field study in a flat grass field corroborates this finding (Pio et al., 2000). We thus estimate a dry deposition velocity of 0.2 cm s$^{-1}$ ± 0.1 cm s$^{-1}$ for ozone at night in the SSJV based on these, as well as other (Pederson et al., 1995; Meszaros et al., 2009; Neirynck et al., 2012; Lin et al., 2010), literature values.
We purposefully ignore NO$_2$ deposition on the basis that crop canopies can be either a small source or sink of NO$_2$ at the surface (Walton et al., 1997). The amount of O$_x$ lost overnight due to deposition would be within our stated uncertainty (± 0.86 ppb h$^{-1}$) as long as $|v_{d\ NO2}| < \sim 2.5$ cm s$^{-1}$, an assumption supported by the literature (Pilegaard et al., 1998; Walton et al., 1997).

Line 410: Is the vertical flux divergence used in the last term or the last two terms?

Response: Yes, it refers to the last two terms.

Lines 412-3: Will the authors better explain what the linear regression here is for, and how it is done?

Response: Changed to "A linear regression through the 20 m resolution vertical O$_x$ profile is used to determine dOx/dz (for the last term in equation 1) in the upper…"

Lines 423-4: By surplus of Ox do the authors mean where Ox indicated by the purple line is greater than Ox indicated by the black line? Please specify this. Also please specify in the caption which of the terms have been inferred (and refer to section on calculation) and which have been observed.

Changes made:

The dashed profiles show the expected profile that would have been observed on the morning flight if only advection (blue), chemical loss (green), or both advection and chemical loss (red) processes were occurring. The observed morning $O_x$ (magenta) is inferred to exceed the predicted morning $O_x$ (red) due to the vertical mixing term in the scalar budget equation.

**Figure 6.** $O_x$ profiles from 2016-06-04 overnight analysis, NBL height (green line), and lower bound to vertical mixing gradient (yellow line). The solid lines are observations and the dashed lines are inferred.

Line 429: How is the error propagation calculated? At least refer to Section 3.2
Table 3: What exactly is the error estimate? At least refer to Section 3.2
Line 433: Please cut "Another way to frame … NBL"
Line 434: Please cut "Further". (In my opinion, doing this and the above suggestion would make this part more digestible).

Response: References to Section 3.2 have been made and the requested cuts have been completed.

Changes made:

**Table 3.** Results from the nocturnal scalar budget for all terms. Estimated error (see section 3.2) in parenthesis.

Of note is the fact that on average the chemical loss is expected to be a little more than twice as large as the physical loss from dry deposition. For dry deposition the average lifetime of ozone is 28 h (200 m / 0.002 m s$^{-1}$), and for chemical loss it is 12 h. Both losses of $O_x$ added together are about triple the observed time rate of change, and thus the physical and chemical losses are largely (~ 2/3) compensated by vertical mixing. Because the RL consistently contains more ozone than the stable NBL, turbulent mixing will result in a transfer of ozone into the NBL. While $NO_2$ is observed to be higher in the NBL than in the RL (by about 3-5 ppbv), it is a much smaller contribution to the $O_x$ ($O_3$ is less than $NO_2$ by anywhere from 10-20 ppbv.)

Line 438: Do the authors mean NO2 is less than O3 by 10-20 ppb here?
Response: Yes. Changed to "$O_3$ is less than $NO_2$ by…"

Line 445: There should be an introductory sentence here, instead of starting with a specific component's error calculation.

Response: Added "Here we estimate the uncertainty for each term in the budget equation, as well as the ultimately calculated eddy diffusivities." as an introductory sentence.

Line 455-6: I would cut the term "conservative". What basis do the authors have for this value judgement? It seems little, especially in terms of the ozone deposition velocity

Response: Done.

Section 3.3: This section is confusing because the authors say that the presence of Fresno Eddy could be problematic for their analysis. Then, they say that the predominant circulation during their flights is similar to Fresno Eddy, but then they say any recirculation has a minimal impact on their results (lines 492-3). A lot of the analysis on Fresno Eddy could be cut, especially because it's found to be irrelevant. This would help with clarity and flow. Additionally, can the authors split Section 3.3 in two? One section on Fresno Eddy, and one on the low-level jet?

Response: As addressed in some of the following comments, we have attempted to clarify our discussion of the Fresno Eddy and where it fits in to this work. We firmly believe that a clear discussion of the Fresno Eddy is absolutely necessary to retain because it is constantly referred to in air quality discussions of the SJV, but not clearly understood. It is a major conclusion of the paper that we sample and describe the Fresno Eddy in a new and better way, which we believe can help illuminate future studies. We have tried to clarify the discussion where possible, but maintain that the low-level jet is *part and parcel* of the Fresno Eddy, therefore separating the two into distinct sections in the manuscript only perpetuates the misleading distinction.

Lines 468-72: Are Zhong et al. 2004 describing the Fresno Eddy conditions, or other prevailing conditions? Please specify.

Response: Changed to "Zhong et al. (2004) uses a series of 915 MHz radio acoustic sounding systems to analyze low-level winds in the SSJV. Their Figure 4 shows that at night, …"

Line 473: The authors need to more clearly specify that they are suggesting there are Fresno Eddy conditions during their flights.

Response: Changed to "…observations, suggesting the presence of a Fresno eddy during our flights."

Lines 480-2: I don't really know what the takeaway here is.

Response: Here we are stating that Zhong et al. (2004) was presenting a climatological analysis of typical summertime conditions, while our flights were targeting periods of higher ozone, thus the synoptic and mesoscale conditions during our flights might be systematically different from climatological norms.

Figure 7: What is shown in the background of the plots? It's hard to see the yellow and light blue colors on top of the grey. I recommend using a different color scheme and/or thicker lines.
Response: The color scheme used was the best one we could find in terms of readability. However, we have increased the resolution so that the arrows stand out better.

Line 494: I would repeat the hypothesis more in full here (i.e., the effects of the nocturnal jet on the next day's ozone levels; "contribute to the variability of ozone" is a bit vague).

Line 494-5: Again, "explored some of the meteorological factors that are absent from the current literature" is vague. Further, why would the authors only explore unexplored factors?

Response: changed to "…variability of maximum daytime ozone concentration, we explored the synoptic patterns that are associated with differing strengths of the LLJ".

Line 498: "in 100m bin space" is too colloquial

Response: Changed to "averaged in 100 m vertical bins,…"

Lines 506-525: This paragraph is confusing. The authors should state up front what they are investigating here.

Response: Moved "To analyze variability … (N=165 nights)" to the first sentence of the following paragraph (line 506) for better flow.

Line 506: Explicitly say which thresholds correspond to strong and weak jets
Line 506: What is "it" here? The trough? Please specify
Line 512: Why the mention of Fresno Eddy here? Are the authors trying to attribute eastward trough to Fresno Eddy not happening? Please clarify
Line 516: What are "those" conditions?

Changes made:

To analyze variability of the jet strength, daily average synoptic charts from the North American Regional Reanalysis (NARR) are created in Figures 8 and 9 for days when the low-level jet strength was less than 7 m s$^{-1}$ (N=147 nights), and greater than 12 m s$^{-1}$ (N=165 nights). Both the strong and weak low-level jets show a climatological trough pattern, but the mean trough axis is situated about 100 km to the east for the strong cases. We also note that the pressure gradient is at least twice as strong for the stronger low-level jets, and that the synoptic pattern of the weak jets favors southerly geostrophic wind aloft, which directly opposes the up-valley northwesterly thermally driven flow. We also note the positive correlation found between the LLJ strength and the upwelling index (r$^2$ = 0.3018, p < 10$^{-5}$), which is primarily driven by the North Pacific High, which when strong, acts to push the trough farther eastward as seen in Figure 8. These findings are consistent with the Lin and Jao (1995) modeling study where the Fresno Eddy (and thus LLJ) did not form when the synoptic flow over the coastal range was westerly. Beaver and Palazoglu (2009) found that maximum daily 8-hour average ozone (MDA8) exceedances were more frequent in the central and southern San Joaquin Valley when an offshore ridge or onshore high were present, consistent with Figure 8 (right). The results of our study suggest that this may be at least partially explained by the presence of a weaker LLJ under those synoptic conditions.

Lines 516-526: It seems like this should be a paragraph on it's own, and better linked with the mention around Line 512 of Fresno Eddy. Referring to "LLJ" generally in this paragraph here is particularly confusing because in the preceding lines the authors were talking about weak vs. strong LLJ.

Response: We have made this a separate paragraph.

Lines 522-527: I'm not exactly sure why the authors feel the need to compromise here.

Response: Changed to "As a provisional synthesis of these seemingly conflicting findings"

Lines 523-524: Previously the authors had said the Fresno Eddy and the LLJ are not the same thing, but here the authors seem to be referring to them interchangeably.
Lines 527: What is in addition to synoptic forcing?
Lines 532: High temperature could also decrease deposition through stomatal pores
Line 534: -> With the NARR climatology.

Response: We are suggesting that the LLJ is the strongest branch of a Fresno eddy, thus a strong eddy will produce a strong LLJ. We have attempted to clarify this in the text and we have added the discussion of stomatal pores.

Changes made:

Future research may attempt to further establish the degree to which the LLJ and Fresno Eddy are linked, as well as which of these two nocturnal mechanisms will dominate the ozone budget under different synoptic conditions. As a provisional synthesis of these seemingly conflicting findings, we suggest that the Fresno Eddy, when present, will act to recirculate pollutants regardless of the strength of the LLJ (the strongest branch of the eddy). That is, a stronger eddy will not recirculate pollutants any more than a weaker eddy will. Thus, the optimal nighttime dynamics for ozone pollution the following day may consist of a Fresno Eddy just coherent enough to effectively recirculate pollutants, but without its strongest branch too strong as to deplete the RL ozone by vertical mixing.
In addition to the synoptic patters discussed above, slightly lower surface temperatures across the entire region during stronger low-level jets are observed. This could either be a consequence of the synoptic flow (southerly geostrophic flow will generally result in warmer temperatures) or itself be an underlying precursor to the LLJ (a colder delta region will lead to more up-valley thermal forcing resulting in stronger winds that decouple from the surface at night). The higher temperatures associated with the weak nocturnal jets may make for a twofold mechanism for high ozone: the high temperatures either causing increased photochemical production or resulting from increased meteorological stagnation, and a lack of mixing overnight induced by the low level jet causing less depletion of the RL ozone. Warmer nights may also result in less dry deposition of $O_x$ through stomatal pores. It is worth noting that this relationship with temperature is only apparent with the NARR climatology, as ambient overnight low temperature at Visalia yields only a very weak relationship with the jet strength ($r^2 = 0.035$, $p < 10^{-5}$).

Figure 9: A map showing the difference in 2m air temperature for stronger vs. weaker LLJ may be more effective. Hard to see the contours. Or maybe just cut the elevation map, and color by temperature contours.

Response: Figure 9 has been changed as suggested

[Figure]

Figure 11 is never referenced, but I think it should be on Line 545. Figure 11 is interesting, but very tangential, and I think the figure and the short discussion of it should be cut.

Response: Figure 11 removed along with the discussion of it.

Line 551: "Another look at … " is not a very helpful way of introducing what the authors are doing here. What are the authors trying to investigate here? Also, what is overnight layering?

Section 3.4 What's the rationale for including the discussion of Figure 12 in the previous section, as opposed to at the beginning of this one? Seems like it would flow better in Section 3.4.

Response: Section 3.4 now starts here. We removed figure 12 and instead added the TKE profile to figure 4, and reference that here.

Changes made:

As seen in Figure 4, an average low-level jet height between 200-400 m is seen, which corresponds approximately with the average observed stable NBL depth.

[Figure]

Line 562: "several previous studies examining different parts of the world"

Line 567: Will the author please make it more clear that their hypothesis is stated on lines 564-5? Line 566: Specify regional mean ozone from monitoring stations in a certain network
Lines 568: Here are the authors examining ozone at the monitoring stations or measured on the aircraft? Please specify

Changes made:

On the other hand, several previous studies examining different parts of the world have proposed that mixing induced by nocturnal jets may decrease ozone levels the following day (Hu et al., 2013; Neu et al., 1995). Greater coupling between the NBL and RL could reduce the amount stored in the RL reservoir rendering cleaner air the following day. This relationship between the eddy diffusivity values found in our study and regional mean surface ozone from the CARB network is analyzed, and serves as both an additional check on the relative validity of the calculated $K_z$ values as well as a test of this proposed hypothesis.

Line 574: Why would the relationship be strongest for MDA8? How much stronger is the relationship for MDA8 vs. max hourly, 24 hour average? If it's a lot stronger, is MDA8 roughly representing ozone at the same hours each day? Examining this could be insightful. Also, why is this relationship stronger for MDA8 than that observed during the fumigation period?

Response: Jin et al., JGR, 2013 suggests that the MDA8 occurs fairly consistently between 13 and 14 PST. For the 24 hour average ozone correlation with eddy diffusivity, $r^2$ = 0.40. I believe

that the relationship is notably weaker for the fumigation periods due to slight variations in timing of the peak boundary layer growth rate.

Line 578: It would help the reader to briefly restate the hypothesis.

Response: Changed to "Because this analysis consisted of only 12 flights, we decided to explore a larger data set that might support the hypothesis that a stronger LLJ reduces ozone the following day."

Lines 580-3: Wait, why not MDA8 here?

Response: For consistency, we changed the analysis to look at MDA8. The new figure is reported below, and the outlier point is removed.

[Figure]

Figure 13 and 14: Please be consistent in terms of ozone on the y vs. x axis.

Response:

Figure 13 (now figure 10) has been adjusted to meet this request.

[Figure]

Lines 593-5: Why would Rb be 0 at night? This doesn't make much sense to me. Is this stated in the Padro 1996? Rb is not included in Padro 1996 Figure 4. In Massman [1994] Rb is estimated to be nonzero for the CODE vineyard. I recommend specifying that not only Ra is modeled in Massman [1994] but Rc is too (it's not a residual of observed vd and estimated Ra and Rb). Then I might just say here that modeled Ra and Rc are similar at night and Rb is unknown, rather than zero. It's also important to note that this is only one way of estimating Ra (u/u_*^2) and estimates at night are likely highly uncertain. Lines 600-3: How would taking changes in Ra into account in the budget calculation change the eddy diffusivity estimate?

Response: Added suggested literature and stated that $r_b$ is unknown and thus not included in this approximation. The average error of $K_z$ due to the uncertainty of $V_d$ is calculated to be ~0.50 $m^2\,s^{-1}$, which is included in the original error propagation analysis.

Changes made:

Where $r_a$ is the aerodynamic resistance, $r_b$ is the viscous sub-layer resistance, and $r_c$ is the surface (canopy) resistance. Figure 4 in Padro (1996) suggests that for ozone at night, $r_a \sim r_c \sim$ 250 s $m^{-1}$. $r_b$ is likely non-zero (Massman et al., 1994) but will be neglected here because it is unknown.

Section 3.5: It would be helpful if the authors introduced the goal of their analysis in this section upfront.

Line 607: Why should the authors values be comparable to Banta et al. 2006 and Lenschow et al. 1988? Please specify. Line 610: Did Banta et al. try to remove buoyancy waves? Line 610-1: Why? What is the implication of this finding?

Response: Specified that these are studies of NBL turbulence. Banta et al. (2006) is a meta analysis of other studies. To the best of my knowledge, buoyancy waves were not removed. While we were hoping that our TKE would have a relationship with ozone the following day, it is a very noisy measurement and we were also using many approximations to estimate it, as outlined in the paper.

Changes made:

Here we attempt to build confidence in the eddy diffusivity estimates by analyzing additional metrics of turbulence. We find that nocturnally and spatially averaged TKE in the NBL ranges from 0.35 and 1.02 $m^2\ s^{-2}$, which is very comparable to values obtained in other NBL studies (Banta et al., 2006; Lenschow et al., 1988).

Line 624: "lower end of the range inferred from the Ox budget". It would be helpful here if the authors re-stated the range of eddy diffusivities that they infer from the Ox budget. Line 626: "our estimates inferred from the Ox budget" Line 631: "similar turbulent environment to ours"? Line 634: Specify here that the Lenschow et al. 1988 eddy diffusivity from the lower half of the NBL is the most comparable. Line 636: "variability in the reported values"

Response: All changes have been made as requested.

Changes made:

Using the average NBL Brunt–Väisälä frequency of 0.023 Hz and a mixing efficiency of 0.6 results in an eddy diffusivity of 0.34 $m^2\ s^{-1}$, which is about three times smaller than the lower end of our range (1.1 – 3.5 $m^2\ s^{-1}$). A recent study of vertical mixing based on scalar budgeting of Radon-222 in the stable boundary by Kondo et al. (2014) estimated 7-day average overnight diffusivities of 0.05 – 0.13 $m^2\ s^{-1}$, which is an order of magnitude below our estimates inferred from the $O_x$ budget.  However, Wilson (2004) conducted a meta-analysis of radar-based estimates of eddy diffusivity in the free troposphere, which is also a generally stable environment, and found a general range of 0.3 – 3 $m^2\ s^{-1}$. Pisso and Legras (2008) estimated diffusivities of about 0.5 in the lower stratosphere during Rossby wave-induced intrusions of mid-latitude air into the subtropical region. A modeling study by Hegglin et al. (2005) reports diffusivities of 0.45 – 1.1 $m^2\ s^{-1}$ in the lower stratosphere with an average Brunt–Väisälä frequency of 0.021 Hz, indicating a similar turbulent environment to ours. Finally, Lenschow et al. (1988) analyzed flight data in the NBL over rolling terrain in Oklahoma, and found eddy diffusivities for heat ($K_h$) of ~0.25 $m^2\ s^{-1}$ for the upper half of the NBL, and ~1 $m^2\ s^{-1}$ for the lower half. To our knowledge, the latter is the most comparable observational finding within the NBL to our range of diffusivities. Nevertheless, the variability in the reported values leads to the inevitable conclusion that vertical diffusivity in very stable environments is poorly understood, and further research is necessary to illuminate its phenomenology.

Lines 640-4: What's the point of this analysis? Because this relationship is expected, does this build confidence in the authors' estimate of eddy diffusivity (at least the variability in eddy diffusivity)? If so, it should be explicitly stated.

Response: Yes, the point of this analysis was to build confidence of our eddy diffusivity measurements. We have clarified this in the first sentence of this section: "Here we attempt to

build confidence in the eddy diffusivity estimates by analyzing additional metrics of turbulence."

Lines 645-9: To me flow is better if the order of these two sentences is flipped

Response: Disagree with reviewer here because subjective turbulence should be mentioned before delving into it.

Lines 658-9: Briefly, why would the unstable layers have to extend upward beyond the NBL depth?

Response: changed to "as the unstable layers appear to be above the NBL where there is communication with the surface."

Line 659-60: Why is this more likely? What's the implication of this?

Response: We are stating that although unstable layers are observed more frequently in urban areas compared to rural areas, we may have simply detected them more often there because the aircraft spends more time in urban areas. Hence, the apparent pattern of more unstable layers in urban areas could be insignificant.

Lines 663-4: Briefly, how would they contribute to overnight mixing?

Response: Absolutely unstable layers in the atmosphere promote the production of turbulence and thus vertical mixing.

Figure 16: Why is only 50 m shown? The authors say they examine thickness of 50m and 100m. It is challenging to interpret this plot. Another color scheme, and a zoomed in map would be better. Also, the font size should be increased. It would be helpful to indicate the location of the Tehachapi pass on the map.

Response: We have made the requested adjustments to the figure. Only 50 m is shown in order to reduce the number of figures in this submission, and we did not believe the 100 m thickness plot added anything particularly useful.

[Figure]

Lines 668-9: Where is this shown? Also, "seen"-> "observed" Line 669: What finding?
Line 674-5: It might be more clear to state that the figure does not support the hypothesis that the authors outlined on lines 671-2. Also where is this figure? It would be helpful if the authors specified that it is not shown.

Response: We did not include this figure for sake of brevity. We are referring to the finding stated in the previous sentence. "Finding" has been removed for better flow.

Changes made:

However, this is consistent with the study by Cho et al. (2003) which found no relationship between turbulence and static stability in the free troposphere. Since the aircraft is moving horizontally a lot faster than it is vertically, one may be concerned that our observations of elevated mixed layers are an artifact of localized temperature gradients that are more prominent in the horizontal dimension.

Line 675-6: How does this fit into the above discussion? What are the implications of this finding?

Response: This fits into the above discussion because we are showing the unstable layers appearing in the climatological averages of the 915 MHz profiler. The implications of this are that it lends some additional credibility to their existence.

Line 687: Cut "slightly"
Line 689: "A limitation of our study"
Line 690: Cut "being conducted". Also what do the authors mean by pairs? Do they mean morning and evening flights? I would specify this. "pairs" is non-intuitive.

Response: These changes have been made.

Changes made:

A limitation of our study is the lack of sample size, with only 12 pairs of overnight and morning flights. However, we believe this study demonstrates the importance of synoptic and mesoscale features at night within the context of high ozone episodes, and the utility of this type of focused flight strategy where terms in the scalar budget equation are measured.

Line 690: What demonstrates? Specify what "it" is.

Response: Changed to "the study demonstrates".

Line 691: Seems strange to mention that the authors demonstrate something "within the context of high ozone episodes" when ozone hasn't been mentioned yet in the conclusion. On a similar note, the authors haven't noted in the conclusion that there was a particular focus strategy of the flights, so it's strange to mention it. It's helpful for the reader if the conclusion can really stand alone from the rest of the text.

Line 692: Specify where the soundings and surface monitoring data are from (locations, networks) here Line 692-3: Specify the implication of this finding (tie back to hypothesis) Line 694: What do the authors mean "although in the former analysis"? In the analysis of soundings and surface network data? This could be more clearly articulated, and it should be directly stated that this is not found in the airborne measurements. Line 695-6: "is an important link that may have consequential implications for modeling studies and policy making" is vague and verbose. I think the authors' findings are important for modeling and policy, but this sentence doesn't do much to convince me of it. Line 697: Introduce Visalia Line 698: "infer" -> "determine" Line 701: Spell out that reduced aerodynamic resistance means more efficient transport to surfaces where ozone can deposit Line 704: It would be good to articulate that this may be why the correlation between night turbulence + next day ozone may not always be high. Line 704: "Airborne measurements from flights over Bakersfield, CA showed …"

Response: Focus strategy of the flight restated in conclusion. The other requested changes have been made.

Changes:

A limitation of our study is the lack of sample size, with only 12 pairs of overnight and morning flights. However, we believe this study demonstrates the importance of synoptic and mesoscale features at night within the context of high ozone episodes, and the utility of this type of focused flight strategy where terms in the scalar budget equation are measured.

The larger set of RASS and ARB surface network data from Visalia, CA establishes a correlation between low level jet speed and the maximum 1-hour ozone the following afternoon for summertime months, further suggesting the link between nocturnal mixing and the following

days ozone. Similarly, the correlations between the aircraft-estimated eddy diffusivities and MDA8 the following day also suggest that vertical mixing in the NBL plays an important role in determining ozone concentrations. In particular, we note that 11 of 12 days where the Visalia, CA ozone concentration exceeded 100 ppb was preceded by a low-level jet speed < 9 m/s. While we cannot determine a causal relationship between a strong low-level jet, stronger mixing, and reduced ozone pollution, we propose that a stronger LLJ leads to greater mixing, which helps deplete the ozone reservoir by bringing it into the stable boundary layer overnight. There it is subject to deposition to the surface, and that dry deposition rate may itself be partially modulated by the strength of the LLJ through reduced aerodynamic resistance resulting in more efficient transport to surfaces where ozone can deposit. Subsequently, when thermals begin to form after sunrise the following morning, there is less ozone to fumigate downward. While the correlation between nocturnal mixing and ozone the following day is not always strong, it is an important link that may have consequential implications for modeling studies and policy making. For example, our findings highlight the crucial need of models to capture the LLJ and Fresno eddy with sufficient resolution. Policy makers may consider putting more stringent emission limitations on days where synoptic and mesoscale patterns appear to favor a lack of overnight mixing.

Of course, in addition to nocturnal mixing, photochemical production of ozone as well as advection will play a major role in the ultimate daytime peak ozone levels observed, which may be why the correlation between nighttime turbulence and afternoon ozone is not always high. Airborne measurements from flights over Bakersfield, CA showed an average photochemical production as high as 6.8 ppb h-1, with an average advection of -0.8 ppb h-1, though on any given day advection tended to be more comparable in magnitude to photochemical production (Trousdell et al., 2016).

Lines 704-6: Spell out the implication of this finding.

Response: We were mainly pointing this out to remind the reader that even though the advection term on average tends to be near zero, it can be large for any particular data point.

Line 706: In what study? Trousdell et al. 2016? If so, the subject should not be "we", it should be "they" or better, Trousdell et al. (2016) Lines 704-10: I'm not quite following why the discussion of Trousdell et al. 2016 is relevant for the conclusions of this paper. Lines 711-2: "illustrated"-> "suggested"; "which consequently has impacts for"-> "and thus likely impacts"

Response: Here we are reminding the reader that there is more to the picture than just vertical mixing of ozone at night, since afternoon ozone concentrations are influenced by advection and photochemical production.

Changes made:

In that study they have demonstrated that on days with very high ozone that pose hazards to human and agricultural health, the ozone abundance is dependent on elevated ozone in the mornings that serve to catalyze photochemical production through the afternoon. Future

modeling studies may directly investigate these factors, which may help elucidate the causal mechanisms of high ozone events.

We have also suggested that the fate of the NO3 plays an important role in the nocturnal Ox budget chemical loss term, and thus likely impacts the following day's maximum ozone concentration.

Lines 712-5: But what exactly is so uncertain about nitrate, and why will it affect ozone? There should be a line stating that the authors haven't measured nitrate on their flights, and how/why this leads to uncertainty in their analysis. The authors should re-introduce alpha, and why it's important. I really like how the authors have spelled out that nitrate measurements (specifically the lifetime) are needed in future nocturnal airborne measurement campaigns. Are there any other measurements or techniques that their analysis suggests doing or developing would reduce uncertainty?

Response: We have followed these suggestions and are also stating that deposition velocity measurements of ozone using eddy covariance on future campaigns would be helpful.

Changes made:

We have also suggested that the fate of the $NO_3$ plays an important role in the nocturnal $O_x$ budget chemical loss term, and thus likely impacts the following day's maximum ozone concentration. The loss of the nitrate radical at night can occur from $N_2O_5$ hydrolysis, reaction with VOCs, or a very rapid reaction with small NO concentrations, and there is considerable uncertainty regarding which reactions dominate without direct measurements of $NO_3$. Thus, the lifetime of $NO_3$ can range from seconds to several minutes, which affects the chemical loss term in the scalar budget equation. It is thus crucial to measure the lifetime of $NO_3$ in future studies that analyze the NBL ozone or $O_x$ budget. We also suggest more direct measurements of aerodynamic resistance and ozone deposition at the surface by eddy covariance in conjunction with future airborne studies.

**Reviewer #2 Response**

Reviewer comment: The authors hypothesize that a strong low-level nighttime jet more effectively mixes down ozone into the stable nighttime boundary layer where it deposits, resulting in lower ozone the next day. On nights with a weaker jet, the residual layer remains decoupled and results in higher ozone the next day. This paper introduces methods for developing nocturnal scalar budgets from aircraft observations. This hypothesis and support from aircraft contributions is a useful contribution to our understanding of the effect of weather patterns on ozone. One general comment is that the authors could better motivate their statement that air quality models need to better forecast this feature with a brief overview of the current ability of models to simulate the nocturnal low-level jet. Generally the paper would also benefit from clearer presentation of the methodology and results including checking for consistent use of terms and figure referencing. The authors seem to discuss methodology and results intermixed in multiple locations, and a more coherent progression of

methodology and results would both shorten and clarify the author's hypothesis and findings. This paper should be published in ACP after addressing these revisions and the comments below.

Response: We would like to thank the reviewer for the useful feedback regarding our submission to ACP, and their recommendation for publication. We have attempted to better partition the methodology and results in the revised manuscript by outlining the plan for the paper in the introduction. We have also included a brief discussion of modeling in the literature review.

Excerpt of changes:

Our goal is to test whether more nocturnal mixing between the residual layer and stable boundary layer, induced by wind-shear turbulence beneath a strong low level jet, will effectively "deplete" ozone in the residual layer, making less available to fumigate the following morning and seed further photochemical production. We will proceed with this in three ways: first, we introduce a method for analysing nocturnal scalar budgets of flight data, which is similar to that of the daytime scalar budgets, and attempt to estimate the eddy diffusivity of Ox in the NBL on each night of the field campaign (sections 3.1 and 3.2). Second, we analyse synoptic conditions around the LLJ, and look at a broader dataset of LLJ strength and the following afternoon's ozone concentrations (sections 3.3 and 3.4). Lastly, we look at other metrics of NBL turbulence in our campaign data such as Turbulent Kinetic Energy (TKE), Bulk Richardson Number (BRN), and elevated mixed layers in order to bolster confidence in our findings (sections 3.5 and 3.6).

Specific comments:

1. Page 3, line 125. I don't understand whether the authors are using the Beaver and Palazoglu (2009) paper to support their hypothesis. They initially say that a strong nighttime LLJ reduces ozone the next day, so how does this reconcile with strong nighttime ventilation resulting in high ozone the next day?

Response: Beaver and Palazoglu (2009) point to the recirculation effects of the Fresno eddy, which as we state later, may appear to conflict with our hypothesis rather than support it. Changes made:

Beaver and Palazoglu (2009) found that ozone pollution in the central San Joaquin Valley is particularly high on days where the preceding nocturnal Fresno Eddy is strong, even when strong ventilation is occurring. They also found that the early morning downslope flow through the Tehachapi pass is a strong predictor of ozone pollution in the SSJV. However, mixing induced by nocturnal jets has been shown to decrease ozone levels the following day in other parts of the world (Hu et al., 2013; Neu et al., 1995), so one might suspect that a Fresno eddy

that creates a particularly strong LLJ on a given night may decrease ozone the following day if the recirculation of ozone does not compensate for the loss due to vertical mixing.

2. Page 8, line 259. Can you comment on the validity of using 0.2 cm/s for the ozone dry deposition velocity when you argue that deposition will be enhanced when the nighttime LLJ is strong?

Response: We estimate that 0.2 cm s$^{-1}$ is an average value of ozone dry deposition at night in our region, and our stated error is 50%. Thus, the estimated variation due to changes in jet strength (~40%) is within our envelope of uncertainty.

3. Page 14, line 376. Please discuss where the uncertainty on the 1.5 value comes from/ this is unclear from the previous discussion.

Response: The uncertainty of this coefficient is discussed in section 3.2 (lines 450-457). Manuscript now refers readers to this section when the value of alpha is first introduced.

4. Page 15, line 410. Do you mean the last term (not the last two)?

Response: While the last term represents the flux at the top of the NBL, the second to last term represents the surface flux. Thus, the flux divergence in the vertical direction is represented by the last 2 terms in equation 1.

5. Page 15, line 423. Please use consistent language to avoid confusion. Do you mean the surplus of Ox observed on the morning flight is inferred to be driven by the "advection" term?

Response: The surplus of $O_x$ refers to the difference between the projected $O_x$ if there were only chemistry and advection at play, and the actual observed morning $O_x$. Since chemistry and advection has been modeled in this figure, we assume the difference between projected and observed is due to vertical mixing.

Changes made:

The dashed profiles show the expected profile that would have been observed on the morning flight if only advection (blue), chemical loss (green), or both advection and chemical loss (red) processes were occurring. The observed morning Ox (magenta) is inferred to exceed the predicted morning Ox (red) due to the vertical mixing term in the scalar budget equation.

6. Page 16, Table 3. You haven't actually explicitly described yet as far as I can tell the procedure for calculating the storage term.

Response: $dO_x/dt$ is calculated from the aircraft profile difference between the late night and sunrise flights.

Additions made:

The overnight average Ox profile was subtracted from the Sunrise profile and divided by the time difference between the midpoints of each flight to compute the storage term.

7. Page 16, line 435. Please clarify which term refers to the observed time rate of change, since no term appears to be double the sum of chemical loss and deposition.

Response: Thank you for catching this – this was misstated. We have changed the text to "Further, both losses of Ox added together are about triple the observed time rate of change, and thus the physical and chemical losses are largely (2/3rds) compensated by vertical mixing."

8. Page 17, line 445. How is the error in the nocturnal PBL height included in the error analysis?

Response: The error in the NBL height is included in the error propagation analysis for the eddy diffusivities.

9. Page 17, line 477. It would be useful to show on the figure the extent of the SSJV and point out the nocturnal low-level jet.

Response: We have extended the image further to the south to show the full SSJV. However, as the figure mainly focuses on observations, we avoid adding cartoon schematics of the mesoscale features, which are not fully known.

[Figure]

10. Page 18, line 501. Give the corresponding PST, since that is what is stated in the abstract.

Response: Done.

11. Page 18, line 504. I assume you are referring to Figures 8-9 here for the daily average synoptic charts? If so, please add this to the text.

Response: Done.

Changes made:

To analyze variability of the jet strength, daily average synoptic charts from the North American Regional Reanalysis (NARR) are created in Figures 8 and 9 for days when the low-level jet strength was less than 7 m s$^{-1}$ (N=147 nights), and greater than 12 m s$^{-1}$ (N=165 nights).

12. Page 19, line 506. Could you again highlight this offset of the figures so that the reader can easily pick out this 100 km difference?

Response: We have drawn a dotted dashed line in the figures to indicate the mean trough axis.

[Figure]

13. Page 19, line 510. You don't need Figure 10, just tell us the correlation coefficient and p-value.

Response: Figure 10 removed.

14. Page 19, line 531. Why would higher temperatures increase photochemical production at night? What about higher temperatures increasing soil NOx? Also, you haven't mentioned PAN at all – wouldn't higher temperatures result in more PAN decomposition to increase Ox?

Response: Here we are arguing that greater daytime photochemical rates (including those due to increased PAN dissociation) during warmer synoptic periods might be an additional factor that increases surface ozone. This would act in addition to less nocturnal mixing (due to the synoptic conditions favoring high temperatures making a weaker LLJ).

15. Figure 11 – please explain the legend in the figure caption.

Response: We have removed this figure from the manuscript in response to another reviewer, as it is not central to our thesis.

16. Page 22, line 555. Is there an explanation for the different behavior of TKE here?

Response: This is likely due to there being less shear immediately under the jet compared to the amount of shear in the surface layer.

17. Page 24, line 602. Can you clarify whether you are still considering a high canopy resistance in the vd calculation?

Response: We are assuming that the canopy resistance does not change.

18. Figure 16 – The font size here is difficult to read.

Response: See adjusted figure below

[Figure]

19. One general comment – the use of eddy diffusivity to evaluate turbulent mixing is most generally applicable to the daytime convective mixed layer. The authors could better support why this framework is applicable to the stable nocturnal boundary layer. This is better done in 3.5, so possibly referencing this section earlier on would be useful.

Response: Eddy diffusivities were the most practical way of estimating the NBL mixing due to the logistics of our study. We have highlighted other literature where eddy diffusivity is estimated in stable, non-convective environments.

[revised manuscript text omitted]

---

## Referee Report (RR1)

While the quality of this manuscript has improved, the authors have not adequately responded to my initial review (Reviewer #1). After responding to the following concerns, I think the paper will be suitable for publication in ACP.

General comments
· The paper is excessively wordy and there are grammar errors. I urge the authors to use less words whenever possible and en dashes in their compound adjectives to enhance readability and check their grammar.
· There needs to be more consistent use of terms and abbreviations. For example, sometimes the authors use "RL" and sometimes "residual layer"
· The abstract should be treated as separate from the paper and the abbreviations and terms should be re-defined.
· "pairs of flights" needs to be introduced as early as the abstract. It's not a given that pairs of flights means night and following morning.
· More information on the regressions should be given. The authors regress ozone on x, y, and z, and then calculate the partial derivatives? What is the error and amount of variability explained by the regression? How many data points go into the regression?
· Again, I urge the authors to shorten and clarify their discussion of nitrate, as it is hard to follow.
· It would be very useful for the reader if section names were a bit more detailed (e.g., articulated findings, or objectives).
· I still find it challenging to interpret Figure 9 with the topography on the map. I urge the authors to reconsider including the topography on this figure.

Line-by-line comments
Line 82: Insert ", which is " before known more generally
Line 102-130: This introductory paragraph on the Fresno Eddy is extremely long and still seems out of context. Please better contextualize this discussion.
Line 115: Many readers may not know what a Froude number is. Please briefly define
Line 114-5: "act as a barrier to the jet" is not clear; please rephrase
Line 115: By "eddy feature" do the authors mean the eddy? Please clarify in text
Line 140-2: This is helpful to the reader, but it seems quite strange to have this description without a prior introduction of the study in the introduction
Line 148: typo
Line 149: "ozone problems" is too colloquial
Line 150-5: This paragraph would benefit from an sentence introducing that the authors are going to start talking about modeling. The authors need to more directly state that models don't capture the nocturnal circulation motivates their study in the text.
Line 151: "—" should be "-"; please check elsewhere that the authors use of "—" vs. "-" is correct
Line 157-69: This is too long and the motivation from daytime studies is a bit convoluted. I brought this up previously but I don't feel like the authors sufficiently addressed my concern (or convinced me that the discussion is necessary). Can the authors simply say that most studies focus on the day, and thus our understanding of the nocturnal ozone budget and mixing on ozone air pollution more generally is limited?
Line 171: Why are there quotes on depletes?
Line 175: Please give the audience context for "broader dataset" - what dataset are the authors using? Also, please say the goal of this analysis here.
Line 177: "bolster" has a negative connotation in my opinion

Line 198: "lab" -> "laboratory"

Line 200: I'm not sure that it's ok to cite papers in preparation

Line 231: I'm not sure that it's to cite papers that have been submitted

Line 233-4: "If time permitted on … , we typically completed … or flew …"

Line 236: "Residual Layer ozone project" has not been defined or acknowledged previously, please revise; "ground tracks" seems colloquial; please give acronyms for the sites that are used on the plot in the caption (as well as which network a given site is a part of). Why are only some of them labeled on the figure? Please label them all. Additionally, please move the label closer to the "x" - not always easy to tell which "x" goes with which label

Line 250: please cut "aforementioned"

Line 252: the objective aims to use? "to address this objective, we use a method …"

Line 263: "the flight volume" is a bit colloquial - please rephrase

Line 269: where is the storage term in equation 1?

Line 272: it would be helpful if the authors had a line here saying something like "in the following sections, we detail the methods for estimating the terms in equation 1"

Line 276-298: Why isn't this paragraph its own section (to estimate h)?

Line 290: I think it would be clearer to state "late night and morning flight pairs"

Line 303: I think it is confusing to say that O3, NO2 and NO3 are grouped together for Ox here. The authors clarify that their definition of Ox is different from that conventionally used in the following lines but I think some general restructuring of this part would help with clarity.

Line 335: are 30 ppb of O3 and 20 ppt of NO3 hypothetical values for SSJV? Please clarify in the text

Line 338: which surface air quality network? I asked this previously; please specify in the text.

Line 343: Will the authors directly link this nitrate lifetime with the implications for ozone here?

Line 373: cut "obvious"

Line 376: cut "and best accounts …. dominant."

Line 378: cut "very"

Line 379: cut "highly"

Line 283: ampersand should not be used here after Table 2

Line 392: give acronyms used in figure in figure caption

Line 395: "2nd" -> "second"

Line 403: "would be" -> "are"

Line 410: "the 1-second Ox data"

Line 414: this is not a sentence

Line 420: the authors' field campaign or that of Padro? Please clarify in the text

Line 422: does Padro conclude this or do the authors infer this? Please clarify in the text

Line 426: cut "purposefully"

Line 436: cut "on any given night"

Line 436-7: "likely accounts … in Ox"

Line 461: Why is this worth noting? Is this observed in a figure? Otherwise seems extraneous to include this.

Line 481: Why is uncertainty in deposition computed in this way? It would only make sense to me if the authors are considering a deposition flux here. Do the authors mean the deposition flux (rather than the deposition velocity) here? If so, please specify.

Line 489: can the authors refer the reader to where they did this previously (e.g., the section)?
Line 501: In the level? Cut level?
Line 504: Zhong et al. (2004)
Line 508: Cut "It is noted that"
Lines 512-4: Can the authors more closely link with line with the previous finding (i.e., that this is additional evidence supporting a minimal influence of advection)
Line 520: "is"-> "are"
Line 524: Cut "that is"
Line 525: Define acronym
Line 528: "To analyze variability of the jet strength" does not give me much insight as to what the authors are trying to do here. Please more clearly lay out the goal. Also, in the following paragraph, will the authors please refer to the figures that they are referencing more.
Line 535: "where"-> "that showed that"
Line 539: "were" -> "was"
Line 542: Cut "thing"
Line 543: Please clarify in the text what the authors mean by essentially
Line 541-552: This entire paragraph needs to be re-worked for clarity
Line 550: By optimal, do the authors mean the best for good air quality? Please revise
Line 555: Refer to Figure 9?
Line 560-2: Why is this worth noting? What is the implication of this finding? Please include in text
Line 575: "50% of daytime values during convective conditions"?
Line 576: "TKE increases"
Line 578: "air pollution problem" is too colloquial
Line 580: Again, what is higher ozone pollution potential?
Line 583-5: Suggestion to break this into two sentences. "relative validity" doesn't make much sense
Line 590: Is the growth entraining into the RL? Suggest re-phrasing
Line 592: Instead of saying "were in the predicted direction" can the authors just say the direction of the relationship?
Line 596: "we explored"
Line 600: "is"-> "are"
Line 604: "This" is confusing here, because the authors were just talking about the outlier
Line 614: "is neglected", "combining an estimate of aerodynamic resistance"
Line 618: "the"
Lines 619-20: "The difference in U10 … assuming an average U10 of …"
Line 626: "will need to"-> "should"
Line 635: "for oceans and the free troposphere"
Line 642: cut "where"
Line 643-4: So do the authors use the median or the average…?
Line 648: "is"=>"are"
Line 664-5: sentence is too colloquial
Line 693: cut "a lot"
Line 693-4: "the observations of elevated mixed layers may be"
Line 695: "to confirm that this is not the case, we examine"
Line 698: "they"
Line 700: What are the implications of this finding?

**Line 704: Mention ozone?**
**Line 705: again, please change "air quality problems"**
**Line 707: correlations between what and both Richardson number and ozone? Specify**
**Line 713: the context of high ozone episodes is hardly discussed in the text**
**Line 717: "next-day ozone"?**
**Line 719: "11 out of 12 days WHEN ozone concentration exceeded 100 ppm over Visala were preceded"**
**Line 722: the ozone reservoir where? Please specify in text**
**Line 723-4: suggestion to separate this into two sentences**

**Below I copied and pasted some of my initial reviews (black), along with the authors' response (green), and my response (black, bold). I ask that the authors also respond to these comments.**

Line 157: Do the authors average over a large area? The limitations would only be overcome if so, right?

Response: The scalar budget technique we present covers a large swath of the SSJV, and thus the terms in the budget equation can be taken as averages of the entire region for which the budget is performed.

**Will the authors more clearly articulate in the text, somewhere close to the beginning, that they are examining a large area of the SSJV? This should be closely linked with the authors' introduction of the Fresno Eddy.**

Line 259: Please specify the field site and time examined in Padro 1996.

Response: changed to "Combining those measurements with an estimated 0.2 cm s$^{-1}$ nighttime dry deposition velocity of ozone at night in the SSJV (Padro, 1996), we can indirectly estimate $K_z$."

**My interpretation of Padro 1996 is that they examine several field sites in the SSJV - which one do the authors examine? Please specify in the text**

Line 271: "A blend of these three methods" is too vague. Please specify the method
Response: Changed to "all three of these methods were used in tandem."

**A "blend" / "in tandem" is too vague. How do the authors combine them? Please specify in the text**

Table 2: What do the authors mean that values may not match literature values? How is the extrapolation and valley average done? It seems like this info should be somewhere in the paper or supplementary material.

Response: We found that often, the measurements in the studies were taken in specific areas such as crop fields. Since the aim of this analysis was merely to get a reasonable estimate, we used our meteorological knowledge to estimate whether a valley-averaged concentration may be slightly higher or lower than what was reported in the study.

Changes made:

The measurements in some of the studies above were taken in specific crop fields. Since the aim of this analysis was merely to obtain an order of magnitude estimate, we predicted whether a valley-averaged

concentration may be slightly higher or lower than what was reported in the study. Thus, values here may not exactly match literature.

**I think back-of-the-envelope calculations are fine here, but the authors need to describe the method. Their description is too hand wavy. Somewhere in the text the authors should describe the land use characterization of the SSJV to give context to the several references to agriculture (e.g., is only a little of the SSJV agriculture?)**

Line 403: What is the similar environment? Please specify

Response: Specified that this study was done in a flat grass field.

**Now it needs to be more clear that this is a land use type (or climate?) representative of the SSJV.**

Lines 423-4: By surplus of Ox do the authors mean where Ox indicated by the purple line is greater than Ox indicated by the black line? Please specify this. Also please specify in the caption which of the terms have been inferred (and refer to section on calculation) and which have been observed.

Changes made:

The dashed profiles show the expected profile that would have been observed on the morning flight if only advection (blue), chemical loss (green), or both advection and chemical loss (red) processes were occurring. The observed morning $O_X$ (magenta) is inferred to exceed the predicted morning $O_X$ (red) due to the vertical mixing term in the scalar budget equation.

**Figure 6.** $O_X$ profiles from 2016-06-04 overnight analysis, NBL height (green line), and lower bound to vertical mixing gradient (yellow line). The solid lines are observations and the dashed lines are inferred.

**Ok, but now it is not exactly clear why Figure 6 is included in the paper. What should the reader be taking away from this snapshot figure? Please better integrate this figure and the discussion of it into the text.**

Line 445: There should be an introductory sentence here, instead of starting with a specific component's error calculation.

Response: Added "Here we estimate the uncertainty for each term in the budget equation, as well as the ultimately calculated eddy diffusivities." as an introductory sentence.

**In my opinion "ultimately calculated" leaves room for confusion. Please rephrase**

Section 3.3: This section is confusing because the authors say that the presence of Fresno Eddy could be problematic for their analysis. Then, they say that the predominant circulation during their flights is similar to Fresno Eddy, but then they say any recirculation has a minimal impact on their results (lines 492-3). A lot of the analysis on Fresno Eddy could be cut, especially because it's found to be irrelevant. This would help with clarity and flow. Additionally, can the authors split Section 3.3 in two? One section on Fresno Eddy, and one on the low-level jet?

Response: As addressed in some of the following comments, we have attempted to clarify our discussion of the Fresno Eddy and where it fits in to this work. We firmly believe that a clear discussion of the

Fresno Eddy is absolutely necessary to retain because it is constantly referred to in air quality discussions of the SJV, but not clearly understood. It is a major conclusion of the paper that we sample and describe the Fresno Eddy in a new and better way, which we believe can help illuminate future studies. We have tried to clarify the discussion where possible, but maintain that the low-level jet is *part and parcel* of the Fresno Eddy, therefore separating the two into distinct sections in the manuscript only perpetuates the misleading distinction.

**I still think the discussion of the Fresno Eddy feels tangential. I urge the authors to better articulate "It is a major conclusion of the paper that we sample and describe the Fresno Eddy in a new and better way, which we believe can help illuminate future studies" in their paper (upfront, and in the conclusions).**

Lines 480-2: I don't really know what the takeaway here is.

Response: Here we are stating that Zhong et al. (2004) was presenting a climatological analysis of typical summertime conditions, while our flights were targeting periods of higher ozone, thus the synoptic and mesoscale conditions during our flights might be systematically different from climatological norms.

**Ok, so can the authors more clearly state this rather than what they currently have (which feels tangential)?**

Lines 516-526: It seems like this should be a paragraph on it's own, and better linked with the mention around Line 512 of Fresno Eddy. Referring to "LLJ" generally in this paragraph here is particularly confusing because in the preceding lines the authors were talking about weak vs. strong LLJ.

Response: We have made this a separate paragraph.

**Again, it seems like the authors have only responded to half of my concern.**

Lines 593-5: Why would Rb be 0 at night? This doesn't make much sense to me. Is this stated in the Padro 1996? Rb is not included in Padro 1996 Figure 4. In Massman [1994] Rb is estimated to be nonzero for the CODE vineyard. I recommend specifying that not only Ra is modeled in Massman [1994] but Rc is too (it's not a residual of observed vd and estimated Ra and Rb). Then I might just say here that modeled Ra and Rc are similar at night and Rb is unknown, rather than zero. It's also important to note that this is only one way of estimating Ra (u/u_*^2) and estimates at night are likely highly uncertain. Lines 600-3: How would taking changes in Ra into account in the budget calculation change the eddy diffusivity estimate?

Response: Added suggested literature and stated that $r_b$ is unknown and thus not included in this approximation. The average error of $K_z$ due to the uncertainty of $V_d$ is calculated to be ~0.50 $\text{m}^2\,\text{s}^{-1}$, which is included in the original error propagation analysis.

Changes made:

Where $r_a$ is the aerodynamic resistance, $r_b$ is the viscous sub-layer resistance, and $r_c$ is the surface (canopy) resistance. Figure 4 in Padro (1996) suggests that for ozone at night, $r_a \sim r_c \sim 250$ s $\text{m}^{-1}$. $r_b$ is likely non-zero (Massman et al., 1994) but will be neglected here because it is unknown.

**Seems to me like it is important to spell out "The average error of Kz due to the uncertainty of vd is calculated to be ~0.50 m2 s-1, which is included in the original error propagation analysis" in the text close to this discussion**

Line 607: Why should the authors values be comparable to Banta et al. 2006 and Lenschow et al. 1988? Please specify. Line 610: Did Banta et al. try to remove buoyancy waves? Line 610-1: Why? What is the implication of this finding?

Response: Specified that these are studies of NBL turbulence. Banta et al. (2006) is a meta analysis of other studies. To the best of my knowledge, buoyancy waves were not removed. While we were hoping that our TKE would have a relationship with ozone the following day, it is a very noisy measurement and we were also using many approximations to estimate it, as outlined in the paper.

Changes made:

Here we attempt to build confidence in the eddy diffusivity estimates by analyzing additional metrics of turbulence. We find that nocturnally and spatially averaged TKE in the NBL ranges from 0.35 and 1.02 $m^2 s^{-2}$, which is very comparable to values obtained in other NBL studies (Banta et al., 2006; Lenschow et al., 1988).

**Can the authors please clarify in the text why they are mentioning that they did not remove buoyancy waves? I would suggest saying something like "differences between the studies may reflect Banta et al. 2006 removing buoyancy waves" if this is what the authors are implying**

**Please answer my question about the implication of the finding (now Lines 632-3**

Line 659-60: Why is this more likely? What's the implication of this?

Response: We are stating that although unstable layers are observed more frequently in urban areas compared to rural areas, we may have simply detected them more often there because the aircraft spends more time in urban areas. Hence, the apparent pattern of more unstable layers in urban areas could be insignificant.

Lines 663-4: Briefly, how would they contribute to overnight mixing?

Response: Absolutely unstable layers in the atmosphere promote the production of turbulence and thus vertical mixing.

**Please incorporate the authors' response into the main text**

Line 675-6: How does this fit into the above discussion? What are the implications of this finding?

Response: This fits into the above discussion because we are showing the unstable layers appearing in the climatological averages of the 915 MHz profiler. The implications of this are that it lends some additional credibility to their existence.

**Please incorporate the authors' response into the main text**

Line 691: Seems strange to mention that the authors demonstrate something "within the context of high ozone episodes" when ozone hasn't been mentioned yet in the conclusion. On a similar note, the authors haven't noted in the conclusion that there was a particular focus strategy of the flights, so it's strange to mention it. It's helpful for the reader if the conclusion can really stand alone from the rest of the text.

Line 692: Specify where the soundings and surface monitoring data are from (locations, networks) here Line 692-3: Specify the implication of this finding (tie back to hypothesis) Line 694: What do the authors mean "although in the former analysis"? In the analysis of soundings and surface network data? This could be more clearly articulated, and it should be directly stated that this is not found in the airborne measurements. Line 695-6: "is an important link that may have consequential implications for modeling studies and policy making" is vague and verbose. I think the authors' findings are important for modeling and policy, but this sentence doesn't do much to convince me of it. Line 697: Introduce Visalia Line 698: "infer" -> "determine" Line 701: Spell out that reduced aerodynamic resistance means more efficient transport to surfaces where ozone can deposit Line 704: It would be good to articulate that this may be why the correlation between night turbulence + next day ozone may not always be high. Line 704: "Airborne measurements from flights over Bakersfield, CA showed ..."

Response: Focus strategy of the flight restated in conclusion. The other requested changes have been made.

Changes:

A limitation of our study is the lack of sample size, with only 12 pairs of overnight and morning flights. However, we believe this study demonstrates the importance of synoptic and mesoscale features at night within the context of high ozone episodes, and the utility of this type of focused flight strategy where terms in the scalar budget equation are measured.

The larger set of RASS and ARB surface network data from Visalia, CA establishes a correlation between low level jet speed and the maximum 1-hour ozone the following afternoon for summertime months, further suggesting the link between nocturnal mixing and the following days ozone. Similarly, the correlations between the aircraft-estimated eddy diffusivities and MDA8 the following day also suggest that vertical mixing in the NBL plays an important role in determining ozone concentrations. In particular, we note that 11 of 12 days where the Visalia, CA ozone concentration exceeded 100 ppb was preceded by a low-level jet speed < 9 m/s. While we cannot determine a causal relationship between a strong low-level jet, stronger mixing, and reduced ozone pollution, we propose that a stronger LLJ leads to greater mixing, which helps deplete the ozone reservoir by bringing it into the stable boundary layer overnight. There it is subject to deposition to the surface, and that dry deposition rate may itself be partially modulated by the strength of the LLJ through reduced aerodynamic resistance resulting in more efficient transport to surfaces where ozone can deposit. Subsequently, when thermals begin to form after sunrise the following morning, there is less ozone to fumigate downward. While the correlation between nocturnal mixing and ozone the following day is not always strong, it is an important link that may have consequential implications for modeling studies and policy making. For example, our findings highlight the crucial need of models to capture the LLJ and Fresno eddy with sufficient resolution. Policy makers may consider putting more stringent emission limitations on days where synoptic and mesoscale patterns appear to favor a lack of overnight mixing.

Of course, in addition to nocturnal mixing, photochemical production of ozone as well as advection will play a major role in the ultimate daytime peak ozone levels observed, which may be why the correlation between nighttime turbulence and afternoon ozone is not always high. Airborne measurements from flights over Bakersfield, CA showed an average photochemical production as high as 6.8 ppb h-1, with an

average advection of -0.8 ppb h-1, though on any given day advection tended to be more comparable in magnitude to photochemical production (Trousdell et al., 2016).

Lines 704-6: Spell out the implication of this finding.

Response: We were mainly pointing this out to remind the reader that even though the advection term on average tends to be near zero, it can be large for any particular data point.

**Changing "within the context of" —> "for", "establishes"-> "shows", "the following days" -> "next-day", "a lack of overnight"-> "weak nocturnal" would be helpful**

Line 706: In what study? Trousdell et al. 2016? If so, the subject should not be "we", it should be "they" or better, Trousdell et al. (2016) Lines 704-10: I'm not quite following why the discussion of Trousdell et al. 2016 is relevant for the conclusions of this paper. Lines 711-2: "illustrated"-> "suggested"; "which consequently has impacts for"-> "and thus likely impacts"

Response: Here we are reminding the reader that there is more to the picture than just vertical mixing of ozone at night, since afternoon ozone concentrations are influenced by advection and photochemical production.

Changes made:

In that study they have demonstrated that on days with very high ozone that pose hazards to human and agricultural health, the ozone abundance is dependent on elevated ozone in the mornings that serve to catalyze photochemical production through the afternoon. Future

modeling studies may directly investigate these factors, which may help elucidate the causal mechanisms of high ozone events.
We have also suggested that the fate of the NO3 plays an important role in the nocturnal Ox budget chemical loss term, and thus likely impacts the following day's maximum ozone concentration.

**I find the discussion of Trousdell et al. 2016 tangential (and thus confusing for the reader). I agree that it is important to point out that photochemical production may lead to the weak correlation. This is could be spelled out concisely after "While the correlation between nocturnal mixing and ozone the following day is not always strong, …". On a similar note (in terms of re-structuring this section), I recommend cutting "it is an important link that may have consequential implications for modeling studies and policy making" because it is vague and wordy and the following sentences illustrate this point well.**

Lines 712-5: But what exactly is so uncertain about nitrate, and why will it affect ozone? There should be a line stating that the authors haven't measured nitrate on their flights, and how/why this leads to uncertainty in their analysis. The authors should re-introduce alpha, and why it's important. I really like how the authors have spelled out that nitrate measurements (specifically the lifetime) are needed in future nocturnal airborne measurement campaigns. Are there any other measurements or techniques that their analysis suggests doing or developing would reduce uncertainty?

Response: We have followed these suggestions and are also stating that deposition velocity measurements of ozone using eddy covariance on future campaigns would be helpful.

Changes made:

We have also suggested that the fate of the NO3 plays an important role in the nocturnal $O_X$ budget chemical loss term, and thus likely impacts the following day's maximum ozone concentration. The loss of the nitrate radical at night can occur from N2O5 hydrolysis, reaction with VOCs, or a very rapid reaction with small NO concentrations, and there is considerable uncertainty regarding which reactions dominate without direct measurements of NO3. Thus, the lifetime of NO3 can range from seconds to several minutes, which affects the chemical loss term in the scalar budget equation. It is thus crucial to measure the lifetime of NO3 in future studies that analyze the NBL ozone or $O_X$ budget. We also suggest more direct measurements of aerodynamic resistance and ozone deposition at the surface by eddy covariance in conjunction with future airborne studies.

**Direct measurements of aerodynamic resistance are not really feasible at this point so I would recommend slightly rephrasing. Additionally, it's not really clear whether the authors want airborne ozone eddy covariance fluxes, or ground-based ozone eddy covariance fluxes.**

---

## Author Response (AR2)

We would like to thank reviewer #1 for reading and re-reading our manuscript and providing additional, conscientious feedback. Below are our point-by-point responses.

**General comments**
• The paper is excessively wordy and there are grammar errors. I urge the authors to use less words whenever possible and en dashes in their compound adjectives to enhance readability and check their grammar.
• There needs to be more consistent use of terms and abbreviations. For example, sometimes the authors use "RL" and sometimes "residual layer"
• The abstract should be treated as separate from the paper and the abbreviations and terms should be re-defined.

We have attempted to reduce the wordiness of the manuscript where possible, and utilized all acronyms consistently.

• "pairs of flights" needs to be introduced as early as the abstract. It's not a given that pairs of flights means night and following morning.

Done.

• More information on the regressions should be given. The authors regress ozone on x, y, and z, and then calculate the partial derivatives? What is the error and amount of variability explained by the regression? How many data points go into the regression?

Separate regressions were performed for every flight. We have clarified this in the text and summarized the key parameters ($r^2$, n, slope errors, etc.).

• Again, I urge the authors to shorten and clarify their discussion of nitrate, as it is hard to follow.

We have accepted most of the reviewer's suggested edits for this section. However, these key processes of nitrate loss have not been explicitly stated in other literature, so we feel that it is necessary to go through them in a thorough, step-by-step format. This justifies the importance of nitrate loss for the nocturnal ozone budget as one of our key conclusions of this study.

We made one additional minor change where we state that the VOC pathway of nitrate loss can consume either 1 or 2 $O_x$ molecules.

• It would be very useful for the reader if section names were a bit more detailed (e.g., articulated findings, or objectives).

Done.

• I still find it challenging to interpret Figure 9 with the topography on the map. I urge the authors to reconsider including the topography on this figure.

We have removed the topography for this particular figure and colorized the contours.

**Line-by-line comments**

**Line 82: Insert ", which is " before known more generally**

Done.

**Line 102-130: This introductory paragraph on the Fresno Eddy is extremely long and still seems out of context. Please better contextualize this discussion.**

We have more clearly stated up front (in the introduction, abstract, and conclusions) our aim in discussing the Fresno Eddy. Namely, we point out that the LLJ is a branch of the Fresno Eddy, which induces nocturnal vertical mixing. This mixing may counteract the effect of the eddy recirculating ozone and its precursors.

Additionally, In accordance with the reviewer's suggestion we have chosen to move much of the detailed discussion of the Fresno Eddy formation from the introduction section to the section on the Fresno Eddy and LLJ.

**Line 115: Many readers may not know what a Froude number is. Please briefly define**
**Line 114-5: "act as a barrier to the jet" is not clear; please rephrase**

After further consideration, we have removed the specific Froude number threshold since the Lin and Jao (1995) model did not initialize the flow parallel to the Tehachapi mountains. Changed to "The Tehachapi Mountains will topographically block the flow of the LLJ (Lin and Jao, 1995)." And this is now in section 3.3.

**Line 115: By "eddy feature" do the authors mean the eddy? Please clarify in text**

Removed "feature".

**Line 140-2: This is helpful to the reader, but it seems quite strange to have this description without a prior introduction of the study in the introduction**

We moved this statement to the last paragraph of the introduction where the general scientific layout and objectives of this study are outlined.

**Line 148: typo**

Cut "currently made".

**Line 149: "ozone problems" is too colloquial**

This sentence was removed in order to reduce wordiness of the paper.

**Line 150-5: This paragraph would benefit from a sentence introducing that the authors are going to start talking about modeling. The authors need to more directly state that models don't capture the nocturnal circulation motivates their study in the text.**

Added "Owing to the complex topography and stable stratification overnight, the dynamics of the NBL and RL in California are difficult to model."

**Line 151: "—" should be "-"; please check elsewhere that the authors use of "—" vs. "-" is correct.**

**Done.**

**Line 157-69: This is too long and the motivation from daytime studies is a bit convoluted. I brought this up previously but I don't feel like the authors sufficiently addressed my concern (or convinced me that the discussion is necessary). Can the authors simply say that most studies focus on the day, and thus our understanding of the nocturnal ozone budget and mixing on ozone air pollution more generally is limited?**

**We have removed the detail about the role of horizontal advection in past daytime budget studies, to improve the flow of this paragraph. We do, however, believe that the references to past daytime budget studies is appropriate in the introduction to give readers the ability to investigate the different uses of this technique. Because the nitrate chemistry is so central to the interpretation of this study's results we believe it is important to maintain the general outline of the major chemical pathways in the introduction.**

**Line 171: Why are there quotes on depletes?**

**Removed quotations.**

**Line 175: Please give the audience context for "broader dataset" - what dataset are the authors using? Also, please say the goal of this analysis here.**

**Changed to "Second, to determine whether our findings can be generalized to climatological timescales we analyze synoptic conditions around the LLJ, and look at a broader dataset of LLJ strength and the following afternoon's ozone concentrations using Radio Acoustic Sounding System (RASS) and California Air Resources Board (CARB) ground network data (sections 3.3 and 3.4)"**

**Line 177: "bolster" has a negative connotation in my opinion**

**Changed to "further support"**

**Line 198: "lab" -> "laboratory"**

**Done.**

**Line 200: I'm not sure that it's ok to cite papers in preparation**
**Line 231: I'm not sure that it's to cite papers that have been submitted**

**Citations removed.**

**Line 233-4: "If time permitted on … , we typically completed … or flew …"**

**Done.**

**Line 236: "Residual Layer ozone project" has not been defined or acknowledged previously, please revise; "ground tracks" seems colloquial; please give acronyms for the sites that are used on the plot in the caption (as well as which network a given site is a part of). Why are only some of them labeled on the figure? Please label them all.**

Additionally, please move the label closer to the "x" - not always easy to tell which "x" goes with which label

"ground tracks" changed to "flight paths". Airports were labeled with ICAO identifiers, and ground sites were marked with an "x". We recognize that this may have been confusing, so we adjusted the figure and caption to more clearly differentiate the airports from the ground sites.

Line 250: please cut "aforementioned"
Line 252: the objective aims to use? "to address this objective, we use a method …"

Done.

Line 263: "the flight volume" is a bit colloquial - please rephrase

Removed "within the flight volume" as we feel the description is adequate without that information.

Line 269: where is the storage term in equation 1?

Clarified as "The storage (left hand side) term"

Line 272: it would be helpful if the authors had a line here saying something like "in the following sections, we detail the methods for estimating the terms in equation 1"
Line 276-298: Why isn't this paragraph its own section (to estimate h)?
Line 290: I think it would be clearer to state "late night and morning flight pairs"

Done.

Line 303: I think it is confusing to say that O3, NO2 and NO3 are grouped together for Ox here. The authors clarify that their definition of Ox is different from that conventionally used in the following lines but I think some general restructuring of this part would help with clarity.

Below is the modified paragraph:

The chemical loss term in Equation 1 is expected to be an important component of the NBL $O_x$ budget. Both $NO_2$ and $NO_3$ are able to regenerate ozone in the presence of sunlight and participate in the same sequence of reactions, which are normally grouped together into a family of species referred to as odd oxygen ($O_x = O_3+NO_2+2NO_3+3N_2O_5$) (Brown et al., 2006; Wood et al., 2004); however, since we did not measure $NO_3$ and $N_2O_5$, in this study we estimate $O_x$ as merely the sum of $O_3+NO_2$ because these are expected to exceed to concentrations of the other $O_x$ species by 1-2 orders of magnitude (Brown et al., 2003; Smith et al.,1995). Considering $O_x$ is useful for our study because the family is conserved in the rapid oxidation of NO by $O_3$ (R1 below) yielding $NO_2$, which is quickly photolyzed to regenerate $O_3$ once the sun rises as part of the standard daytime photostationary state.

Line 335: are 30 ppb of O3 and 20 ppt of NO3 hypothetical values for SSJV? Please clarify in the text

**They are typical values observed in the SSJV. Clarified in the text.**

**Line 338: which surface air quality network? I asked this previously; please specify in the text.**

**Specified CARB network in text.**

**Line 343: Will the authors directly link this nitrate lifetime with the implications for ozone here?**
**Added "Hence, we conclude that (R6) should not be ignored in general as it may ultimately reduce the chemical loss rate of $O_x$."**

**Line 373: cut "obvious"**
**Line 376: cut "and best accounts …. dominant."**

**Done.**

**Line 378: cut "very"**
**Line 379: cut "highly"**

**"Very important" changed to "critical". "Highly" cut.**

**Line 283: ampersand should not be used here after Table 2**
**Line 392: give acronyms used in figure in figure caption**
**Line 395: "2nd" -> "second"**
**Line 403: "would be" -> "are"**
**Line 410: "the 1-second Ox data"**

**Done.**

**Line 414: this is not a sentence**

**Changed to "Per convention, u is the mean x-component (zonal) wind and v is the mean y-component (meridional) wind."**

**Line 420: the authors' field campaign or that of Padro? Please clarify in the text**
**Line 422: does Padro conclude this or do the authors infer this? Please clarify in the text**

**Changed to "There are reports of ozone deposition in the area of our field campaign from a 1994 study using the eddy covariance technique (Padro, 1996). The findings of their study suggest nocturnal ozone deposition velocities are several times smaller than their daytime counterparts, but we infer that the overall process is still important for the budget in the NBL because of the smaller mixed layer depth (Eq. 1)."**

**Line 426: cut "purposefully"**
**Line 436: cut "on any given night"**
**Line 436-7: "likely accounts … in Ox"**

**Done.**

**Line 461: Why is this worth noting? Is this observed in a figure? Otherwise seems extraneous to include this.**

Moved to section 3.4 where this is better contextualized.

**Line 481: Why is uncertainty in deposition computed in this way? It would only make sense to me if the authors are considering a deposition flux here. Do the authors mean the deposition flux (rather than the deposition velocity) here? If so, please specify.**

Yes, we meant deposition flux, which we have now stated in the text.

**Line 489: can the authors refer the reader to where they did this previously (e.g., the section)?**
**Line 501: In the level? Cut level?**
**Line 504: Zhong et al. (2004)**
**Line 508: Cut "It is noted that"**

Done.

**Lines 512-4: Can the authors more closely link with line with the previous finding (i.e., that this is additional evidence supporting a minimal influence of advection)**

Added "which further supports the idea that the influence of advection on our scalar budget analysis is minimal."

**Line 520: "is"-> "are"**
**Line 524: Cut "that is"**

Done.

**Line 525: Define acronym**

Changed standard deviation acronym to σ

**Line 528: "To analyze variability of the jet strength" does not give me much insight as to what the authors are trying to do here. Please more clearly lay out the goal. Also, in the following paragraph, will the authors please refer to the figures that they are referencing more.**

Changed to "To analyze possible synoptic influences of the jet strength" and included more references to figures.

**Line 535: "where"-> "that showed that"**
**Line 539: "were" -> "was"**
**Line 542: Cut "thing"**

Done.

**Line 543: Please clarify in the text what the authors mean by essentially**
**Line 541-552: This entire paragraph needs to be re-worked for clarity**
**Line 550: By optimal, do the authors mean the best for good air quality? Please revise**

**The updated paragraph is pasted below:**

Although the LLJ and Fresno Eddy are not synonymous, we propose that the northwesterly LLJ can be the dominant feature of the eddy's northerly flow component. This leads to an important question about the role of the Fresno Eddy in modulating the daily ozone peak. Beaver and Palazoglu (2009) purport that ozone levels in the central SJV are particularly high on days when the morning southerly wind at Parlier, a site about midway between Fresno and Visalia, is strong, concluding that recirculation from the downslope branch of the Fresno Eddy significantly controls the day's buildup of ozone. However, mixing induced by LLJs in other parts of the world has been shown to decrease ozone levels the following day (Hu et al., 2013; Neu et al., 1995). Thus, it may be the case that a Fresno Eddy associated with a particularly strong LLJ may decrease ozone the following day if the recirculation of ozone and its precursors does not overcompensate for overnight losses due to vertical mixing down to the surface. We suggest that the Fresno Eddy, when present, will act to recirculate pollutants regardless of the strength of the LLJ. That is, a stronger eddy will not recirculate pollutants any more than a weaker one will. Thus, the nighttime dynamical conditions that will lead to the greatest ozone levels the following day may consist of a Fresno Eddy just coherent enough to effectively recirculate pollutants, but without an associated LLJ so strong as to deplete the RL ozone by vertical mixing. There is currently no established link in the literature between the Fresno Eddy and LLJ strength. Thus, future research should investigate which of these two nocturnal mechanisms (recirculation from the eddy or RL depletion by vertical mixing) will dominate the ozone budget on any given night, taking into consideration the different possible structures and timing of the Fresno Eddy as well as the synoptic conditions that engender them.

**Line 555: Refer to Figure 9?**

**Done.**

**Line 560-2: Why is this worth noting? What is the implication of this finding? Please include in text**

**We have removed this brief discussion in the interest of simplifying this section.**

**Line 575: "50% of daytime values during convective conditions"?**
**Line 576: "TKE increases"**

**Done.**

**Line 578: "air pollution problem" is too colloquial**

**Removed "problem".**

**Line 580: Again, what is higher ozone pollution potential?**

**Removed "potential".**

**Line 583-5: Suggestion to break this into two sentences. "relative validity" doesn't make much sense**

Text now reads "On the other hand, greater coupling between the NBL and RL, induced by turbulence generated by the LLJ, could reduce the amount of ozone stored in the RL reservoir rendering cleaner air the following day. To test this hypothesis, the relationship between the eddy diffusivity values found in our study and regional mean surface ozone from the CARB network is analyzed."

Line 590: Is the growth entraining into the RL? Suggest re-phrasing

Changed to "after the bulk of the fumigation has occurred".

Line 592: Instead of saying "were in the predicted direction" can the authors just say the direction of the relationship?
Line 596: "we explored"
Line 600: "is"-> "are"

Done.

Line 604: "This" is confusing here, because the authors were just talking about the outlier

Clarified as "This overall relationship supports our hypothesis that the LLJ leads to stronger mixing, which in turn leads to more RL ozone depletion."

Line 614: "is neglected", "combining an estimate of aerodynamic resistance"
Line 618: "the"
Lines 619-20: "The difference in U10 … assuming an average U10 of …"
Line 626: "will need to"-> "should"

Done.

Line 635: "for oceans and the free troposphere"

Clayson and Kantha (2008) applied a method that has previously been used in the oceans *to* the free troposphere, so the original wording is more correct. Clarified this in the text as "Clayson and Kantha (2008) applied a technique that has been *previously* used in oceans to the free troposphere, where turbulence is sparse and intermittent, much like the NBL."

Line 642: cut "where"

Done.

Line 643-4: So do the authors use the median or the average…?

Median – specified this in text.

Line 648: "is"=>"are"

Done.

**Line 664-5: sentence is too colloquial**

Changed to "The weak correlation is probably the result of the limited data set coupled with the challenging nature of both the eddy diffusivity and BRN measurements."

**Line 693: cut "a lot"**
**Line 693-4: "the observations of elevated mixed layers may be"**
**Line 695: "to confirm that this is not the case, we examine"**
**Line 698: "they"**

Done.

**Line 700: What are the implications of this finding?**

Added "Even in the two month averages, some nocturnal unstable layers are detectable between 500 and 1500 m, which further supports the existence of persistent elevated mixed layers that may contribute to overnight mixing of pollutants in the lower troposphere over the valley."

**Line 704: Mention ozone?**

Mentioned $O_x$.

**Line 705: again, please change "air quality problems"**

Changed sentence to "We have demonstrated a method for performing a nocturnal Ox budget analysis using aircraft data, and applied it to estimate the effects of turbulent mixing in the NBL, which can be used to help understand many air quality issues in the SJV."

**Line 707: correlations between what and both Richardson number and ozone? Specify**

Specified eddy diffusivities.

**Line 713: the context of high ozone episodes is hardly discussed in the text**

Changed to, "… and highlights the significant influence that synoptic and mesoscale meteorological conditions can have on the overnight destruction of ozone, thereby impacting the following day's peak concentrations."

**Line 717: "next-day ozone"?**
**Line 719: "11 out of 12 days WHEN ozone concentration exceeded 100 ppm over Visala were preceded"**

Done.

**Line 722: the ozone reservoir where? Please specify in text**

Specified RL.

**Line 723-4: suggestion to separate this into two sentences**

Text now reads "There it is subject to dry deposition at the surface, wherein the deposition velocity itself may be modulated by the strength of the LLJ. Because the near-surface winds are accelerated by an overlying jet, a stronger LLJ reduces the aerodynamic resistance resulting in more efficient transport to surfaces and stomata where ozone can be taken up."

Below I copied and pasted some of my initial reviews (black), along with the authors' response (green), and my response (black, bold). I ask that the authors also respond to these comments.

Line 157: Do the authors average over a large area? The limitations would only be overcome if so, right?
Response: The scalar budget technique we present covers a large swath of the SSJV, and thus the terms in the budget equation can be taken as averages of the entire region for which the budget is performed.
Will the authors more clearly articulate in the text, somewhere close to the beginning, that they are examining a large area of the SSJV? This should be closely linked with the authors' introduction of the Fresno Eddy.

Done (see line 119/120).

Line 259: Please specify the field site and time examined in Padro 1996.
Response: changed to "Combining those measurements with an estimated 0.2 cm s-1 nighttime dry deposition velocity of ozone at night in the SSJV (Padro, 1996), we can indirectly estimate $Kz$."
My interpretation of Padro 1996 is that they examine several field sites in the SSJV - which one do the authors examine? Please specify in the text

Changed to "Combining those measurements with an estimated 0.2 cm s$^{-1}$ nighttime dry deposition velocity of ozone in the SSJV (an average from a study over cotton, grass, mixed deciduous forest, and vineyard field sites by Padro, 1996), we can indirectly estimate Kz. In the following sections, we detail the methods for estimating the terms in Equation 1"

Line 271: "A blend of these three methods" is too vague. Please specify the method
Response: Changed to "all three of these methods were used in tandem."
A "blend" / "in tandem" is too vague. How do the authors combine them? Please specify in the text

Changed paragraph pasted below that attempts to clarify how these methods are combined:

Profiles of wind speed, potential temperature, NO$_2$, and O$_3$ from each night and morning flight were analyzed to make a best guess of the NBL height, h. Figure 4 shows the average scalar profiles from all 15 late night flights to illustrate the typical gradients in the lower portion of the atmosphere. One method of determining h is to observe the lowest elevation where $\partial\theta/\partial z$ becomes close to adiabatic, as the layer below that physically represents air that is in thermodynamic communication with the radiatively cooled surface (Stull, 1988). Another method is to use the level of wind maximum, or LLJ height, when one is present. We found that both of these estimates typically yielded similar values of h. On nights where

there was significant disagreement between the two different estimates, the vertical jump (or sharpest gradient) of $O_x$ in the height region of the NBL-RL interface was considered, as this likely points to a region of maximum mixing. In such cases, we averaged the height where the steepest gradient was observed with the estimates obtained from the other two methods. It should be noted that some subjectivity was involved for determining a final value of h for each night because wind maxima and thermal gradients were not always clearly defined in the profiles. All of the aforementioned factors lead to an estimated uncertainty of ±100 m for all of the NBL heights obtained. The average conditions from the late night and morning flights are presented in Table 1.

Table 2: What do the authors mean that values may not match literature values? How is the extrapolation and valley average done? It seems like this info should be somewhere in the paper or supplementary material.
Response: We found that often, the measurements in the studies were taken in specific areas such as crop fields. Since the aim of this analysis was merely to get a reasonable estimate, we used our meteorological knowledge to estimate whether a valley-averaged concentration may be slightly higher or lower than what was reported in the study.
Changes made:
The measurements in some of the studies above were taken in specific crop fields. Since the aim of this analysis was merely to obtain an order of magnitude estimate, we predicted whether a valley-averaged concentration may be slightly higher or lower than what was reported in the study. Thus, values here may not exactly match literature.

I think back-of-the-envelope calculations are fine here, but the authors need to describe the method. Their description is too hand wavy. Somewhere in the text the authors should describe the land use characterization of the SSJV to give context to the several references to agriculture (e.g., is only a little of the SSJV agriculture?)

Table 2 has been updated to specify the methods behind the (rough) estimates with footnotes. Also specified that the SJV contains about 5 million acres (~20,000 km$^2$) of irrigated land.

Line 403: What is the similar environment? Please specify
Response: Specified that this study was done in a flat grass field.
Now it needs to be more clear that this is a land use type (or climate?) representative of the SSJV.

Changed to "Based on an abundance of observations of nocturnal ozone dry deposition velocities reported in the literature over a broad variety of grassland and agricultural surfaces similar to those found in the SSJV (Pederson et al., 1995; Pio et al., 2000; Meszaros et al., 2009; Neirynck et al., 2012; Lin et al., 2010), all ranging between about 0.1 – 0.3 cm s$^{-1}$, we estimate a dry deposition velocity of 0.2 cm s$^{-1}$ (± 0.1 cm s$^{-1}$) for our purposes."

Lines 423-4: By surplus of Ox do the authors mean where Ox indicated by the purple line is greater than Ox indicated by the black line? Please specify this. Also please specify in the caption which of the terms have been inferred (and refer to section on calculation) and which have been observed.
Changes made:
The dashed profiles show the expected profile that would have been observed on the morning flight if only advection (blue), chemical loss (green), or both advection and chemical loss (red)

processes were occurring. The observed morning Ox (magenta) is inferred to exceed the predicted morning Ox (red) due to the vertical mixing term in the scalar budget equation.
**Figure 6.** Ox profiles from 2016-06-04 overnight analysis, NBL height (green line), and lower bound to vertical mixing gradient (yellow line). The solid lines are observations and the dashed lines are inferred.

**Ok, but now it is not exactly clear why Figure 6 is included in the paper. What should the reader be taking away from this snapshot figure? Please better integrate this figure and the discussion of it into the text.**

**Added "The contribution of vertical mixing to the budget can be visualized as an inferred difference between $O_x$ profiles that are observed and $O_x$ profiles that are predicted from other terms in Equation 1. Figure 6 shows an example of [this]…" and we moved this figure and its associated discussion to section 2.2.5, as we feel it is more appropriate there after considering the reviewer's comment.**

Line 445: There should be an introductory sentence here, instead of starting with a specific component's error calculation.
Response: Added "Here we estimate the uncertainty for each term in the budget equation, as well as the ultimately calculated eddy diffusivities." as an introductory sentence.
**In my opinion "ultimately calculated" leaves room for confusion. Please rephrase**

**Removed "ultimately calculated".**

Section 3.3: This section is confusing because the authors say that the presence of Fresno Eddy could be problematic for their analysis. Then, they say that the predominant circulation during their flights is similar to Fresno Eddy, but then they say any recirculation has a minimal impact on their results (lines 492-3). A lot of the analysis on Fresno Eddy could be cut, especially because it's found to be irrelevant. This would help with clarity and flow. Additionally, can the authors split Section 3.3 in two? One section on Fresno Eddy, and one on the low-level jet?
Response: As addressed in some of the following comments, we have attempted to clarify our discussion of the Fresno Eddy and where it fits in to this work. We firmly believe that a clear discussion of the Fresno Eddy is absolutely necessary to retain because it is constantly referred to in air quality discussions of the SJV, but not clearly understood. It is a major conclusion of the paper that we sample and describe the Fresno Eddy in a new and better way, which we believe can help illuminate future studies. We have tried to clarify the discussion where possible, but maintain that the low-level jet is *part and parcel* of the Fresno Eddy, therefore separating the two into distinct sections in the manuscript only perpetuates the misleading distinction.

**I still think the discussion of the Fresno Eddy feels tangential. I urge the authors to better articulate "It is a major conclusion of the paper that we sample and describe the Fresno Eddy in a new and better way, which we believe can help illuminate future studies" in their paper (upfront, and in the conclusions).**

**See earlier response to lines 102-130.**

Lines 480-2: I don't really know what the takeaway here is.
Response: Here we are stating that Zhong et al. (2004) was presenting a climatological analysis of typical summertime conditions, while our flights were targeting periods of higher ozone, thus

the synoptic and mesoscale conditions during our flights might be systematically different from climatological norms.

**Ok, so can the authors more clearly state this rather than what they currently have (which feels tangential)?**

**Now stated directly in the text.**

Lines 516-526: It seems like this should be a paragraph on it's own, and better linked with the mention around Line 512 of Fresno Eddy. Referring to "LLJ" generally in this paragraph here is particularly confusing because in the preceding lines the authors were talking about weak vs. strong LLJ.
Response: We have made this a separate paragraph.
**Again, it seems like the authors have only responded to half of my concern.**

**In the new manuscript, we have attempted to clarify the linkage between the LLJ and Fresno Eddy, and why they are both being discussed as a single entity.**

Lines 593-5: Why would Rb be 0 at night? This doesn't make much sense to me. Is this stated in the Padro 1996? Rb is not included in Padro 1996 Figure 4. In Massman [1994] Rb is estimated to be nonzero for the CODE vineyard. I recommend specifying that not only Ra is modeled in Massman [1994] but Rc is too (it's not a residual of observed vd and estimated Ra and Rb). Then I might just say here that modeled Ra and Rc are similar at night and Rb is unknown, rather than zero. It's also important to note that this is only one way of estimating Ra (u/u_*^2) and estimates at night are likely highly uncertain.
Lines 600-3: How would taking changes in Ra into account in the budget calculation change the eddy diffusivity estimate?
Response: Added suggested literature and stated that rb is unknown and thus not included in this approximation. The average error of Kz due to the uncertainty of Vd is calculated to be ~0.50 m2 s-1, which is included in the original error propagation analysis.
Changes made:
Where *ra* is the aerodynamic resistance, *rb* is the viscous sub-layer resistance, and *rc* is the surface (canopy) resistance. Figure 4 in Padro (1996) suggests that for ozone at night, *ra* ~ *rc* ~ 250 s m-1. *rb* is likely non-zero (Massman et al., 1994) but will be neglected here because it is unknown.
**Seems to me like it is important to spell out "The average error of Kz due to the uncertainty of vd is calculated to be ~0.50 m2 s-1, which is included in the original error propagation analysis" in the text close to this discussion**

**Done (lines 644/645).**

Line 607: Why should the authors values be comparable to Banta et al. 2006 and Lenschow et al. 1988? Please specify. Line 610: Did Banta et al. try to remove buoyancy waves? Line 610-1: Why? What is the implication of this finding?
Response: Specified that these are studies of NBL turbulence. Banta et al. (2006) is a meta analysis of other studies. To the best of my knowledge, buoyancy waves were not removed. While we were hoping that our TKE would have a relationship with ozone the following day, it is a very noisy measurement and we were also using many approximations to estimate it, as outlined in the paper.
Changes made:
Here we attempt to build confidence in the eddy diffusivity estimates by analyzing additional metrics of turbulence. We find that nocturnally and spatially averaged TKE in the NBL ranges

from 0.35 and 1.02 m2 s-2, which is very comparable to values obtained in other NBL studies (Banta et al., 2006; Lenschow et al., 1988).

**Can the authors please clarify in the text why they are mentioning that they did not remove buoyancy waves? I would suggest saying something like "differences between the studies may reflect Banta et al. 2006 removing buoyancy waves" if this is what the authors are implying**

**Please answer my question about the implication of the finding (now Lines 632-3)**

After contacting the lead author we have verified that Banta et al. (2006) did not remove buoyancy waves.

Text now reads "The average value of $\sigma_u/U_x$ in this study is 0.11, approximately double what was reported in Banta et al. (2006). There is no detectable relationship between our calculated NBL TKE and eddy diffusivities, LLJ speed, or MDA8 the following day, which implies that the eddy diffusivities calculated from the scalar budget analysis may be a better measure of nocturnal mixing strength than TKE."

Line 659-60: Why is this more likely? What's the implication of this?
Response: We are stating that although unstable layers are observed more frequently in urban areas compared to rural areas, we may have simply detected them more often there because the aircraft spends more time in urban areas. Hence, the apparent pattern of more unstable layers in urban areas could be insignificant.

Lines 663-4: Briefly, how would they contribute to overnight mixing?
Response: Absolutely unstable layers in the atmosphere promote the production of turbulence and thus vertical mixing.

**Please incorporate the authors' response into the main text**

Done (lines 698/699).

Line 675-6: How does this fit into the above discussion? What are the implications of this finding?
Response: This fits into the above discussion because we are showing the unstable layers appearing in the climatological averages of the 915 MHz profiler. The implications of this are that it lends some additional credibility to their existence.

**Please incorporate the authors' response into the main text**

Done (lines 728/729).

Line 691: Seems strange to mention that the authors demonstrate something "within the context of high ozone episodes" when ozone hasn't been mentioned yet in the conclusion. On a similar note, the authors haven't noted in the conclusion that there was a particular focus strategy of the flights, so it's strange to mention it. It's helpful for the reader if the conclusion can really stand alone from the rest of the text. Line 692: Specify where the soundings and surface monitoring data are from (locations, networks) here

Line 692-3: Specify the implication of this finding (tie back to hypothesis) Line 694: What do the authors mean "although in the former analysis"? In the analysis of soundings and surface network data? This could be more clearly articulated, and it should be directly stated that this is not found in the airborne measurements. Line 695-6: "is an important link that may have consequential implications for modeling studies and policy making" is vague and verbose. I think the authors' findings are important for modeling and policy, but this sentence doesn't do much to convince me of it. Line 697: Introduce Visalia Line 698: "infer" -> "determine" Line 701: Spell out

that reduced aerodynamic resistance means more efficient transport to surfaces where ozone can deposit Line 704: It would be good to articulate that this may be why the correlation between night turbulence + next day ozone may not always be high. Line 704: "Airborne measurements from flights over Bakersfield, CA showed ..."

Response: Focus strategy of the flight restated in conclusion. The other requested changes have been made.
Changes:
A limitation of our study is the lack of sample size, with only 12 pairs of overnight and morning flights. However, we believe this study demonstrates the importance of synoptic and mesoscale features at night within the context of high ozone episodes, and the utility of this type of focused flight strategy where terms in the scalar budget equation are measured.
The larger set of RASS and ARB surface network data from Visalia, CA establishes a correlation between low level jet speed and the maximum 1-hour ozone the following afternoon for summertime months, further suggesting the link between nocturnal mixing and the following days ozone. Similarly, the correlations between the aircraft-estimated eddy diffusivities and MDA8 the following day also suggest that vertical mixing in the NBL plays an important role in determining ozone concentrations. In particular, we note that 11 of 12 days where the Visalia, CA ozone concentration exceeded 100 ppb was preceded by a low-level jet speed < 9 m/s. While we cannot determine a causal relationship between a strong low-level jet, stronger mixing, and reduced ozone pollution, we propose that a stronger LLJ leads to greater mixing, which helps deplete the ozone reservoir by bringing it into the stable boundary layer overnight. There it is subject to deposition to the surface, and that dry deposition rate may itself be partially modulated by the strength of the LLJ through reduced aerodynamic resistance resulting in more efficient transport to surfaces where ozone can deposit. Subsequently, when thermals begin to form after sunrise the following morning, there is less ozone to fumigate downward. While the correlation between nocturnal mixing and ozone the following day is not always strong, it is an important link that may have consequential implications for modeling studies and policy making. For example, our findings highlight the crucial need of models to capture the LLJ and Fresno Eddy with sufficient resolution. Policy makers may consider putting more stringent emission limitations on days where synoptic and mesoscale patterns appear to favor a lack of overnight mixing. Of course, in addition to nocturnal mixing, photochemical production of ozone as well as advection will play a major role in the ultimate daytime peak ozone levels observed, which may be why the correlation between nighttime turbulence and afternoon ozone is not always high. Airborne measurements from flights over Bakersfield, CA showed an average photochemical production as high as 6.8 ppb h-1, with an average advection of -0.8 ppb h-1, though on any given day advection tended to be more comparable in magnitude to photochemical production (Trousdell et al., 2016).

Lines 704-6: Spell out the implication of this finding.

Response: We were mainly pointing this out to remind the reader that even though the advection term on average tends to be near zero, it can be large for any particular data point.
**Changing "within the context of" —> "for", "establishes"-> "shows", "the following days" -> "next-day", "a lack of overnight"-> "weak nocturnal" would be helpful**

**Done.**

Line 706: In what study? Trousdell et al. 2016? If so, the subject should not be "we", it should be "they" or better, Trousdell et al. (2016) Lines 704-10: I'm not quite following why the discussion of Trousdell et al. 2016 is relevant for the conclusions of this paper. Lines 711-2: "illustrated"-> "suggested"; "which consequently has impacts for"-> "and thus likely impacts"

Response: Here we are reminding the reader that there is more to the picture than just vertical mixing of ozone at night, since afternoon ozone concentrations are influenced by advection and photochemical production.
Changes made:
In that study they have demonstrated that on days with very high ozone that pose hazards to human and agricultural health, the ozone abundance is dependent on elevated ozone in the mornings that serve to catalyze photochemical production through the afternoon. Future modeling studies may directly investigate these factors, which may help elucidate the causal mechanisms of high ozone events. We have also suggested that the fate of the NO3 plays an important role in the nocturnal Ox budget chemical loss term, and thus likely impacts the following day's maximum ozone concentration.

**I find the discussion of Trousdell et al. 2016 tangential (and thus confusing for the reader). I agree that it is important to point out that photochemical production may lead to the weak correlation. This is could be spelled out concisely after "While the correlation between nocturnal mixing and ozone the following day is not always strong, …". On a similar note (in terms of re-structuring this section), I recommend cutting "it is an important link that may have consequential implications for modeling studies and policy making" because it is vague and wordy and the following sentences illustrate this point well.**

**We have followed these suggestions.**

Lines 712-5: But what exactly is so uncertain about nitrate, and why will it affect ozone? There should be a line stating that the authors haven't measured nitrate on their flights, and how/why this leads to uncertainty in their analysis. The authors should re-introduce alpha, and why it's important. I really like how the authors have spelled out that nitrate measurements (specifically the lifetime) are needed in future nocturnal airborne measurement campaigns. Are there any other measurements or techniques that their analysis suggests doing or developing would reduce uncertainty?
Response: We have followed these suggestions and are also stating that deposition velocity measurements of ozone using eddy covariance on future campaigns would be helpful.
Changes made:
We have also suggested that the fate of the NO3 plays an important role in the nocturnal Ox budget chemical loss term, and thus likely impacts the following day's maximum ozone concentration. The loss of the nitrate radical at night can occur from N2O5 hydrolysis, reaction with VOCs, or a very rapid reaction with small NO concentrations, and there is considerable uncertainty regarding which reactions dominate without direct measurements of NO3. Thus, the lifetime of NO3 can range from seconds to several minutes, which affects the chemical loss term in the scalar budget equation. It is thus crucial to measure the lifetime of NO3 in future studies that analyze the NBL ozone or Ox budget. We also suggest more direct measurements of aerodynamic resistance and ozone deposition at the surface by eddy covariance in conjunction with future airborne studies.

**Direct measurements of aerodynamic resistance are not really feasible at this point so I would recommend slightly rephrasing. Additionally, it's not really clear whether the authors want airborne ozone eddy covariance fluxes, or ground-based ozone eddy covariance fluxes.**

**Specified that these are suggestions for future field campaigns. Text now reads "We also suggest more direct estimates of aerodynamic resistance and ozone deposition at the surface by ground-based eddy covariance flux measurements in conjunction with future airborne studies."**

**Residual Layer Ozone, Mixing, and the Nocturnal Jet in California's San Joaquin Valley**

Dani J. Caputi[1], Ian Faloona[1], Justin Trousdell[1], Jeanelle Smoot[1], Nicholas Falk[1], Stephen Conley[2]

[1]Department of Land, Air, and Water Resources, University of California Davis, Davis, 95616, USA
[2]Scientific Aviation, Inc., Boulder, 80301, USA

*Correspondence to*: Dani J. Caputi (djcaputi@ucdavis.edu)

**Abstract:** The San Joaquin  Valley of California is known for excessive  ozone air pollution owing to local production combined with terrain-induced flow patterns that channel air in from the highly-populated San Francisco Bay area and stagnate it against the surrounding mountains. During the summer, ozone violations of the National Ambient Air Quality Standards (NAAQS) are notoriously common, with the San Joaquin Valley having an average of 115 violations of the  current 70 ppb standard each year between 2012 and 2016. Because regional photochemical production peaks with actinic radiation, most studies focus on the daytime, and consequently the nocturnal chemistry and dynamics that contribute to these summertime high ozone events are not as well elucidated. Here we investigate the hypothesis that on nights with a strong low-level jet (LLJ), ozone in the residual layer (RL) is more effectively mixed down into the nocturnal boundary layer (NBL) where it is subject to dry deposition to the surface, the rate of which is itself enhanced by the strength of the LLJ, resulting in lower ozone levels the following day. Conversely, nights with a weaker LLJ will sustain RLs that are more decoupled from the surface, retaining more ozone overnight, and thus lead to more fumigation of ozone  the following mornings, giving rise to higher ozone concentrations the following afternoon. The relative importance of this effect, however, is strongly dependent on the net chemical overnight loss of $O_x$ (here $[O_x] \equiv [O_3] + [NO_2]$) which we show is highly uncertain without knowing the ultimate chemical fate of the nitrate radical ($NO_3$). We analyse aircraft data from a study sponsored by the California Air Resources Board (CARB) aimed at quantifying the role of RL ozone in the high ozone episode events in this area. By formulating nocturnal scalar budgets based on pairs of consecutive flights: the first around midnight and the second just after sunrise the following day, we estimate the rate of vertical mixing between the RL and the NBL, and thereby infer eddy diffusion coefficients in the top half of the NBL. The average depth of the NBL observed on the 12 pairs of flights of this study was 210 (± 50) m. Of the average -1.3 ppb h$^{-1}$ loss of  $O_x$  in the NBL during the overnight hours from midnight to 06:00 PST, -0.2 ppb h$^{-1}$ was found to be due to horizontal advection, -1.2 ppb h$^{-1}$ due to dry deposition, -2.7 ppb h$^{-1}$ to chemical loss via nitrate production, and +2.8 ppb h$^{-1}$ from mixing into the NBL from the RL overnight. Based on the observed gradients of $O_x$ in the top half of the NBL, these mixing rates yield eddy diffusivity estimates ranging from 1.1  3.5 m$^2$ s$^{-1}$, which  are found to inversely correlate with the following afternoon's ozone levels,  providing support for our hypothesis. The diffusivity values are approximately an order of magnitude larger than the few others reported in the extant literature for the NBL, which further suggests that the vigorous nature of nocturnal mixing in this region, due to the LLJ,  may have an important control on daytime ozone levels. Additionally, we propose that the LLJ is a branch of what is colloquially referred to as the Fresno Eddy, which has been previously proposed to recirculate pollutants. However, vertical mixing from the LLJ may counteract this effect, which highlights the importance of studying the LLJ and Fresno Eddy as a single interactive system. The synoptic conditions that are associated with strong LLJs are found to contain deeper troughs along the California coastline . The LLJ observed during this study had an average centreline height of 340 m,  average speed of 9.9 m s$^{-1}$ (SD = 3.1 m s$^{-1}$), and a typical peak timing around 23:00 PST. Seven years of 915 MHz radio-acoustic sounding system and surface air quality network data show an inverse correlation between the jet strength and ozone the following day, further suggesting that air quality models need to forecast the strength of the LLJ in order to more accurately predict ozone violations.

**1. Introduction**

The main source of air for California's Southern San Joaquin Valley (SSJV) is incoming maritime flow from the San Francisco Bay area, which gets accelerated toward the southern end of the valley as a consequence of the valley-mountain circulation (Rampanelli et al., 2004; Schmidli and Rottuno, 2010). The local sources of ozone precursors are scattered along this primary inflow path to the SSJV. The ozone build-up in the SSJV results from both the large amount of local upwind sources and the Tehachapi Mountains to the south which block the flow, preventing advection out of the region (Dabdub et al., 1999; Pun et al., 2000). Because of this tendency for the air to stagnate, both daytime and nocturnal mesoscale dynamics are likely important in the phenomenology of ozone pollution in this area.

100     Under typical fair weather conditions over the continents, thermals are generated near the surface beginning shortly after sunrise, buoyantly forcing a convectively mixed layer, which is known more generally as the  daytime atmospheric boundary layer (ABL). As solar heating increases  the Earth's surface  throughout the day, this layer reaches its maximum height by late afternoon, typically between 700 and 900 m in the SJV during summer months (Bianco et al., 2011). Around sunset, when the solar heating abates, the convective thermals shut off and  no longer power

105 turbulent mixing in the boundary layer. The result of the subsequent radiative cooling of the ground throughout the night forms a stable, nocturnal boundary layer (NBL), typically extending between 100 and 500 m (Stull, 1988) above the surface. The erstwhile convective layer from the daytime, after spinning down and no longer actively mixing, functions as a residual reservoir for pollutants and other trace gases from daytime emissions and photochemical production. This layer overlying the NBL is known as the residual layer (RL).

110     During both daytime and nighttime, mixing can occur between the boundary layer and the layer of air above. In the daytime over land in clear sky conditions, this process of entrainment is driven by convective thermals that penetrate into the laminar free troposphere above,  and then sink back into the convective layer, and may be augmented by wind shear near the top of the boundary layer (Conzemius and Fedorovich, 2006). Entrainment has been shown to be a significant factor for near-surface air quality, and more generally for scalar budgets, as

115 the two interacting layers  often have different trace gas concentrations (Lehning et al., 1998; Trousdell et al., 2016; Vilà-Guerau de Arellano et al., 2011). At night, another type of gas exchange can occur between the aforementioned NBL and the RL by shear-induced mixing. Extensive observations of the structure of the NBL indicate that a localized wind maximum near the top of the NBL, known as a  low-level jet (LLJ), is often present (Banta et al., 2002; Garratt, 1985; Kraus et al., 1985). This LLJ is able to

120 drive sheer production of turbulencethereby  promoting the mixing between these layers despite the stable stratification. In this study, we suggest that the LLJ in the SSJV is part of the northerly flow component of what is colloquially referred to as the Fresno Eddy. As we attempt to show,

125 The Fresno Eddy can drive both vertical mixing and regional horizontal advection.

130 ~~to gravitational potential energy of flow encountering a barrieris lower than about 0.2 (Lin and Jao, 1995). The eddy feature is formed during the hours before dawn when this northwesterly flow interacts with southeasterly nocturnal downslope flow coming from the high southern Sierra Nevada Mountains, although there is some question as to the extent to which the southeasterly flow observed in the morning hours is merely the result of a topographic deflection and recirculation of the nocturnal jet. The Coriolis force helps to circulate this flow; however, a mesoscale low is not~~

135

sunset, while the katabatic drainage flow peaks shortly before dawn, so these two components of the Fresno eddy are (Bianco et al., 2011), suggesting that shear-induced downward mixing of RL ozone in this region may be particularly strong. Beaver and Palazoglu (2009) found that ozone pollution in the central San Joaquin Valley is particularly high following day (Aneja et al. 2000; Zhang and Rao, 1999). Using SODAR data from the Swiss plateau, Neu et al. (1995) estimated that about 75 % of the contribution to the differencefollowing day's early afternoon ozone was due to vertical the NBL depletion. This study was done in complex terrain of Switzerland and primarily used SODAR data. They of time the wind maximaum at night were observedas below 150 m, and the aforementioned early afternoon ozone Coupling of the RL and NBL via intermittent turbulence has also been shown to correlate with overnight ozone spikes at ground-level monitoring stations (Salmond and McKendry, 2005). Because of the complexity of intermittent nocturnal turbulence, the spatial and temporal distributions of these spikes are unknown, and thus it is not known the extent to which these ozone spikes help to deplete the residual layerRL ozone or contribute to the following day's from Southern Taiwan also found that residual layerRL ozone plays an important role in the following day's ozone with fumigation of this ozone into the developing daytime boundary layer accounting for 19 48% of the variance daily maximum (Lin 20122008). As the ozone problems in Southern Taiwan are not heavily driven by local sources, a

    Owing to the complex topography and stable stratification overnight, the dynamics of the NBL and RL in California are difficult to model. Bao et al. (2008) reports that while the Weather Research and Forecasting (WRF) model is able to qualitatively capture the LLJ, systematic errors up to 2 m $\cdot$s$^{-1}$ are observed, with root mean square errors of 4 — 5 m $\cdot$s$^{-1}$. Above 2000 m, a similar magnitude of errors in the model's ability to forecast wind is observed, and since the LLJ is influenced by this upper level synoptic forcing, there is a need for more systematic study of the background synoptic conditions associated with strong and weak LLJs. The authors also note that apart from the 915 MHz Radio Acoustic Sounding Systems (RASS), observations of the LLJ in the SSJV are lacking in spatial coverage. This further highlights the need for an observational-based study of low level winds in the SSJV during high ozone episodes.

    At the core of our observational method, we acknowledge recognize that most scalar budgets are driven by horizontal advection, vertical mixing (primarily entrainment), and local emissions/uptake, and net chemical production (including chemical gains and/or losses). Conley et al. (2011) and Faloona et al. (2009) have shown that on any given day, advection can be a relatively large and significant term in the daytime scalar budget. However, when averaged over numerous flight days, the advection is often close to zero. While many previous sStudiesstudies performing of daytime scalar ozone budgets of ozone (Kleinman et al., 1994; Conley et al., 2011; Lehning et al., 1998; Lenschow et al., 1981; Trousdell et al., 2016) have shown that photochemical production is important, and similarly, wea few nocturnal studies have highlighted significant losses of ozone in the dark (Brown et al., 2006; Stutz et al., 2010), we present here the first complete budget to include the mixing and chemistry overnight. expect the chemical loss of ozone to be important at night. The nocturnal ozone chemistry is driven primarily by its well-known reaction radical. The nitrate radical has many different loss pathways including can combining with $NO_2$ to equilibrate with (which can undergo hydrolysis on surfaces),s, reacting with VOCshydrocarbons, and or rapidly reacting with NO to $
[revised manuscript text omitted]

[Figure]

**Figure 3.** Flight paths of all aircraft deployments in this field campaign (green). Airports where low approaches were conducted (red triangles) and ground ozone monitors (blue crosses) are shown. From north to south, the airports are Fresno Yosemite International Airport (FAT), Visalia Municipal Airport (VIS), Delano Municipal Airport (DLO), and Bakersfield Meadows Field Airport (BFL). From north to south, the CARB  ground ozone network stations (blue crosses)  Fresno-Sierra Skypark #2, Clovis-N Villa Avenue, Fresno-Garland, Fresno-Drummond Street, Parlier, Visalia-N Church Street, Hanford-S Irwin Street, Shafter-Walker Street, Bakersfield-5558 California Avenue, Edison, Bakersfield Municipal Airport.

The nocturnal scalar budget analyses presented here utilizes all late night (~ 21:45 – 00:00 PST) flights in which a subsequent flight was conducted the following morning (~ 06:15 – 08:30 PST). The dates (before midnight PST) of the late-night flights for the 12 overnight periods are shown in Table 1. Additionally, late night flights without a subsequent morning flight were flown on 12 September 2015 and 26 July 2016, and morning flights without a preceding late night flight were flown on 10 September 2015, 24 July 2016, 12 August 2016, and 14 August 2016. These additional flights are included in the analyses here that refer exclusively to either the late night or morning flights, but were not used for the scalar budgets.

**2.2. Scalar Budget Analysis**

Here we aim to test the importance of the  nocturnal mixing on the ozone budget in this region by applying a scalar budgeting technique to the aircraft data in order to estimate an eddy diffusivity between the

NBL and the RL. To address this objective, we use a similar method that has been presented with daytime scalar budgets (Conley et al., 2011; Faloona et al., 2009; Trousdell et al., 2016) to further demonstrate the overall practicality of this methodology.

The nocturnal budget equation is formulated by the Reynolds-averaged conservation equation for a scalar – in this case $O_x$ – in a turbulent medium. $O_x$ is defined here as $NO_2+O_3$ in order to avoid the effects of titration of $O_3$ by NO. If not depleted by chemical oxidation to $NO_3$ and further reaction products, $NO_2$ will photolyze the following day to reproduce ozone in photostationary state, so it can act as an overnight reservoir of ozone. The chemical loss of $O_x$ is then computed by the reaction between $O_3$ and $NO_2$ to form nitrate, and the ultimate fate of nitrate will affect the overall $O_x$ loss. In the stable nighttime environment we will treat the mixing between the RL and NBL by using an eddy diffusivity. The NBL $O_x$ budget can thus be represented as:

$$\frac{\partial [O_x]}{\partial t} = -\alpha k_{O3+NO2}[O_3][NO_2] - \bar{u}\frac{\overline{\Delta[O_x]}}{\Delta x} - \bar{v}\frac{\overline{\Delta[O_x]}}{\Delta y} + \frac{-[O_3]_{SFC}*|v_d|}{h} + \frac{K_z\frac{\Delta[O_x]}{\Delta z}}{h} \qquad (1)$$

Where the term on the left represents the change in concentration with respect to time . The leftmost term on the right side of Eq. 1 represents the net loss of $O_x$ due to chemical reaction of the resultant $NO_3$ and contains an unknown constant of proportionality, $\alpha$, which depends on the subsequent reaction pathway of $NO_3$, and can range from 0 – 3. For reasons later discussed, $\alpha$ is assumed to be ~ 1.5 for this analysis. The next two terms represent changes due to advection by the horizontal wind, followed by terms representing the dry deposition of ozone to the surface, and finally the vertical turbulent mixing term that uses the vertical gradient and the eddy diffusivity, $K_z$ – a number that encapsulates the strength of the overnight mixing. The storage (left hand side) term, chemical loss, advection, surface ozone, and NBL height can be calculated using the aircraft data. Combining those measurements with an estimated 0.2 cm s$^{-1}$ nighttime dry deposition velocity of ozone  in the SSJV (an average from a study over cotton, grass, mixed deciduous forest, and vineyard field sites by (Padro, 1996), we can indirectly estimate $K_z$. In the following sections, we detail the methods for estimating the terms in Equation 1.

[Figure]

**Figure 4.** Mean and ±1 standard deviation (swathes) of potential temperature, ozone, NO, NO$_2$,  wind speed, and turbulent kinetic energy (mean only) from all late-night flights.

**2.2.1. NBL Height**

[revised manuscript text omitted]

[1] Drew Gentner of Yale University, personal communication.

[2] No measurements reported in the SSJV, an order of magnitude estimate is made based on typical aerosol concentrations.

[3] Arey et al. (1990) reported 70 ppt in an orange grove. We estimate 50 ppt as a SJV average.

**Table 2.** Estimations of VOC reactions with nitrate in the summertime nocturnal boundary layer for the SSJV.
Reaction rates from Atkinson & Arey (1998), Table 2  and Atkinson (2006).

[Figure]

**Figure 5.** Diurnal plots of temperature and relative humidity during flight days of the Residual Layer Ozone campaign (individual days = grey lines, campaign average = blue lines), compared to 1 June — 30 September 2015 and 2016 averages (red lines) at the Fresno (FAT), Visalia (VIS), and Bakersfield (BFL) airports Automated Weather Observing System (AWOS) network. Hours are in Pacific Standard Time (PST).

Consequently, we calculate the net reaction (R1-R6) for the nocturnal chemical loss rate of $O_x$ as a constant multiple of (R2). The 2nd second order rate equation for the net chemical loss of $O_x$ is calculated by:

$$\left.\frac{dO_x}{dt}\right|_{chemical\ loss} = -\alpha k_{O3+NO2}[O_3][NO_2] \qquad \textbf{(3)}$$

Where α can range from 0 — 3, and per the discussion above, is estimated to be $1.5 \pm 0.5$ (uncertainty discussed in section 3.2). To estimate a value for the second order rate constant ($k_{O3+NO2}$), we start with the temperature dependent function for this reaction (Sander et al., 2006):

$$k_{O3+NO2} = 1.2(10^{-13}) * e^{\frac{-2450}{T}} \qquad \textbf{(4)}$$

Where T is the temperature in Kelvin. For the domain being analyzed, an instantaneous value of $k_{O3+NO2}$ is determined at each data point. These values of $k_{O3+NO2}$ are then averaged to obtain a constant value for the given night. It should be noted that small errors in the value of $k$ that are within the order of our temperature fluctuations were found to not have a measurable impact on the chemical loss term. To estimate the chemical loss of $O_x$, the initial 20 m altitude bins for $NO_2$ and $O_3$ are taken from the late night and morning profiles. In each bin, the concentrations are linearly interpolated between the late night and morning values, so that there is an estimation of the current average concentration within that bin at every time during the night.

**2.2.3. Horizontal Advection by Mean Wind**

The advection term in Equation 1 is calculated by first collecting all 1-second $O_x$ data points for the late night and morning flights separately. For each flight, a multiple linear regression is fit through the 1-second $O_x$ data for latitude (y), longitude (x), and altitude (z), allowing estimations for the horizontal gradients of $O_x$ ($\partial[O_x]/\partial x$ and $\partial[O_x]/\partial y$) in the horizontal advection term. The $r^2$ values of the regressions ranged from 0.25 to 0.69, and the number of data that they contained ranged from 2813 to 5323. Typical values of the horizontal $O_x$ gradients were of order $0.1 \pm 0.02$ ppb km$^{-1}$. To compute the total advection term within the NBL on a given flight, these gradients are combined the mean wind speeds :

$$Advection_{Ox} =- \left[\left(\frac{\partial[O_x]}{\partial x} * \bar{u}\right) + \left(\frac{\partial[O_x]}{\partial y} * \bar{v}\right)\right] \qquad \textbf{(5)}$$

 Per convention, $u$ is the mean $x$-component (zonal) wind and $v$ is the mean $y$-component (meridional) wind. The same procedure is repeated for the morning flights, and the advection terms from the late night and morning flights are averaged together.

**2.2.4. Dry deposition of $O_x$**

Deposition of ozone is presumed to be an important sink of $O_x$ at the surface, the flux of which can be parameterized as the product of the surface ozone values (measured directly from the aircraft) and the deposition
455  velocity for ozone. There are reports of ozone deposition in the area of  our field campaign from a 1994 study using the eddy covariance technique (Padro, 1996). The findings of their study suggest nocturnal ozone deposition velocities are several times smaller than the daytime counterpart, but we infer that the overall process is still important for the budgetin the NBL because of the smaller mixed layer depth (Eq. 1).
460   Based on an abundance of observations of nocturnal ozone dry deposition velocities reported in the literature over a broad variety of grassland and agricultural surfaces similar to those found in the SSJV literature values~~.
465  deposition on the basis that crop canopies can be either a small source or sink of $NO_2$ at the surface (Walton et al., 1997). The amount of $O_x$ lost overnight due to deposition would be within our stated uncertainty (± 0.86 ppb h$^{-1}$) as long as $|v_{d\ NO2}| < \sim 2.5$ cm s$^{-1}$, an assumption supported by the literature (Pilegaard et al., 1998; Walton et al., 1997).

**2.2.5. Vertical Turbulent Mixing between the NBL and the RL**

Finally, a vertical flux divergence for $O_x$ must be estimated for Equation 1, which is represented by the last two terms.
470  For the top part of the NBL, the flux of $O_x$ can be interpreted as an eddy diffusivity ($K_z$) multiplied by the vertical gradient of $O_x$ between the NBL and RL.  For each flight, a linear regression through the 1-second $O_x$  data within the NBL-RL interface is used to determine $\partial[O_x]/\partial z$ (for the last term in Equation 1) in the upper portion of the NBL that appeared to contain the strongest $O_x$ gradient. The average r$^2$ value of the 24 regressions was 0.11, and the number of data points that they contained ranged from 116 to
475  2166. Typical values of the vertical $O_x$ gradients were $\sim 0.07 \pm 0.04$ ppb m$^{-1}$. The layers used for the regression fit were 100 - 200 m thick and did not extend below 70 m AGL  to avoid capturing the region where the $O_x$ sink due to surface deposition and/or reaction with freshly emitted NO likely accounts for the vertical gradient in $O_x$ (Fig. 6). The eddy diffusivity can now be solved for with all of the other terms estimated.

[Figure]

480

**Figure 6.** $O_x$ profiles from 2016-06-04 overnight analysis, NBL height (green line), and lower bound to vertical mixing gradient (yellow line). The solid lines are observations and the dashed lines are calculated based on expected changes due to horizontal advection (blue), chemical loss (green), and the sum of the two (red).

**3. Results and Discussion**

485   **3.1. Overnight Mixing and the  Scalar Budget Results**

[revised manuscript text omitted]

**3.3 The Fresno Eddy and LLJ**

The formation of the Fresno Eddy begins when the daytime northwesterly -valley wind continues into the late evening, decoupling from the surface and forming a LLJ (Davies 2000). The Tehachapi Mountains will

525 typically topographically block the flow of the LLJ (Lin and Jao, 1995). The eddy is formed during the hours before dawn when this northwesterly flow interacts with southeasterly nocturnal downslope flow coming from the high

southern Sierra Nevada Mountains, although there is some question as to the extent to which the southeasterly flow observed in the morning hours is merely the result of a topographic deflection and recirculation of the nocturnal jet. The Coriolis force helps to circulate this flow; however, a mesoscale low is not thought to develop (Bao et al. 2007,
530    Lin and Jao, 1995). We note that the valley flow peaks around midnight, while the katabatic drainage flow peaks shortly beforenear dawn, so these two components of the Fresno eddyFresno Eddy are not time-coherent. The initial northwesterly wind and a topographic blockage are both critical for determining whether or not the eddy will form on a given night (Lin and Jao, 1995).

One complicating factor that remains for this particularour scalar budget analysis is the presence of the Fresno
535    eddy and its influence that this eddy will have on our measurements of advection. If an eddy is recirculating a scalar quantity, using a simple linear fit model as we did in section 2.2.3 to estimate advection would be questionable, especially if the flight area only covered a small portion of the larger mesoscale circulation. Zhong et al. (2004) uses a series of 915 MHz RASS to analyze low-level winds in the SSJV. Their Figure 4 shows that at night, the northwesterly low level jetLLJ is formed in the San Joaquin ValleySJV, and a weak katabatic southerly flow is
540    observed in the foothills to the east at the Trimmer site. As the night progresses, the eddy becomes more coherent as the northwesterly jet relaxes while the southerly flow strengthens and expands westward. After daybreak, the eddy appears to deform and disintegrate with much of the SSJV experiencing a strong southerly wind.

[revised manuscript text omitted]

jets, and that the synoptic pattern of the weak jets favors a southerly geostrophic wind aloft, which directly opposes the up-valley northwesterly thermally driven flow. We also  find  a positive correlation found between the LLJ strength and the upwelling index ($r^2 = 0.3018$, $p < 10^{-5}$), calculated by NOAA's Pacific Fisheries Environmental Lab at 33N, 119W (https://www.pfeg.noaa.gov/products/PFEL/modeled/indices/upwelling/NA/upwell_menu_NA.html). The indices are primarily driven by the strength and position of the North Pacific High, which, when strong, acts to push the 700 mb trough farther eastward as seen in Figure 8b, and is associated with lower sea surface temperatures and thus enhanced thermal forcing of the coupled sea breeze and valley wind. These findings are consistent with the Lin and Jao (1995) modeling study  that showed that the Fresno Eddy (and  LLJ) did not form when the synoptic flow over the coastal range was westerly. Beaver and Palazoglu (2009) found that maximum daily 8-hour average ozone (MDA8) exceedances were more frequent in the central and southern San Joaquin Valley when an offshore ridge or onshore high  was present, consistent with Figure 8a. The results of our study suggest that this may be at least partially explained by the presence of a weaker LLJ under those synoptic conditions.

 Although the LLJ and Fresno Eddy are not synonymous, we propose that the northwesterly LLJ can be the  dominant feature of the  eddy's northerly flow component. This leads to an important question about the role of the Fresno Eddy in modulating the daily ozone peak. Beaver and Palazoglu (2009) purport that ozone levels in the central SJV are particularly high on days when the morning southerly wind at Parlier, a site about midway between Fresno and Visalia, is strong, concluding that recirculation from the downslope branch of the Fresno Eddy significantly controls the day's buildup of ozone. However, mixing induced by LLJs in other parts of the world has been shown to decrease ozone levels the following day (Hu et al., 2013; Neu et al., 1995). Thus, it may be the case that a Fresno Eddy associated with a particularly strong LLJ may decrease ozone the following day if the recirculation of ozone and its precursors does not overcompensate for overnight losses due to vertical mixing down to the surface. We suggest that the Fresno Eddy, when present, will act to recirculate pollutants regardless of the strength of the LLJ. That is, a stronger eddy will not recirculate pollutants any more than a weaker one will. Thus, the nighttime dynamical conditions that will lead to the greatest ozone levels the following day may consist of a Fresno Eddy just coherent enough to effectively recirculate pollutants, but without an associated LLJ so strong as to deplete the RL ozone by vertical mixing. There is currently no established link in the literature between the Fresno Eddy and LLJ strength. Thus, future research should investigate which of these two nocturnal mechanisms (recirculation from the eddy or RL depletion by vertical mixing) will dominate the ozone budget on any given night, taking into consideration the different possible structures and timing of the Fresno Eddy as well as the synoptic conditions that engender them.~~Although the LLJ and Fresno Eddy are not exactly the same thing, rathersynonymous, the LLJ is part of the northwesterly flow that is an important precursor to the Fresno Eddy. When the eddy is presentwhen the Fresno Eddy is present, the northwesterly LLJ is essentially typically the strongest branch of the eddy. This leads to an important question about the role of the Fresno Eddy in modulating the next-day ozone. Beaver and Palazoglu (2009) found that ozone pollution in the central SJV is particularly high on days where the preceding nocturnal Fresno Eddy is strong, concluding that recirculation from the Fresno Eddy contributes to a buildup of ozone. However, mixing induced by~~

620

In addition to the synoptic patterns discussed above, slightly lower surface temperatures across the entire region are observed during stronger LLJs  (Fig. 9). This could either be a consequence of the synoptic flow (southerly geostrophic flow will generally  bring warm air advection ) or itself be an underlying precursor to the LLJ. In the latter, a ~2 K  greater temperature difference between the

625 delta region and the SSJV for strong LLJs (seen in Fig. 9) will lead to more up-valley thermal forcing resulting in stronger winds that decouple from the surface at night. The higher temperatures associated with the weak nocturnal jets may make for a twofold mechanism for high ozone: the high temperatures either causing increased photochemical production or resulting from increased meteorological stagnation, and a lack of mixing overnight induced by the LLJ causing less depletion of the RL ozone. Warmer nights may also result in less dry deposition of $O_x$ through

630 stomatal pores.

[Figure]

(a)                                         (b)

**Figure 8.** North American Regional Reanalysis 700 mb Geopotential Height (m) for low-level jet speeds

635  less than 7 m s$^{-1}$ (a) and greater than 12 m s$^{-1}$ (b).

[Figure]

**Figure 9.** North American Regional Reanalysis 2 m air temperature (°C) difference between cases where low-level jet speeds exceeding 12 m s$^{-1}$ and cases where it is below 7 m s$^{-1}$ at 01 PST. Positive values indicate warmer surface temperatures for strong jets.

640

**3.4. Vertical  Mixing  and Next-Day Ozone**

As seen in Figure 4, an average LLJ height  of 200-400 m is seen, which corresponds approximately with the average  NBL depth. Likely due to the shear induced by the LLJ, turbulence is seen to be vigorous at night with TKE values about 50 % of daytime values during convective conditions. Further, TKE increases toward the surface, a condition that Banta et al. (2006) refers to as a "traditional" stable boundary layer.

The thermals generated by solar heating after sunrise initiate a fumigation process where as the daytime boundary layer develops, the ozone that was in the RL is mixed downward. The change in surface ozone concentration (d[O$_3$]/dt) due to fumigation peaks at around 08:00 PST and continues until about 10:00 PST. The relationship between our estimated eddy diffusivities and ozone during the fumigation period is strongest at 10:00 am PST, after the bulk of the  fumigation has occurred (r$^2$=0.29, p=0.07). A negative correlation between eddy diffusivities and the maximum 1-hour ozone, 24-hour average ozone, and MDA8 were also found, with the strongest relationship  for the MDA8 (r$^2$=0.46, p=0.015), as shown in Figure 10. This supports our hypothesis that stronger NBL turbulence is associated with lower ozone the following day.

[Figure]

**Figure 10.** Correlation between overnight eddy diffusivity and maximum daily 8 hour8-hour average ozone (MDA8) the following day. All values are averages of 11 CARB surface network stations that are within the flight region.

Because this analysis consisted of only 12 flights, we decided to exploreexplored a larger data set that might support the hypothesis that a stronger LLJ reduces ozone the following day. 7 years of low-level jetLLJ speeds obtained from the Visalia sounder from 2010 –– 2016 is combined with the CARB surface network ozone monitoring site at Visalia N Church St (36.3325° N, 119.2908° W, 30 m elevation) for analysis. Only calendar days 152 through 273 (June –– September) is are included. The low level jetLLJ, hypothesized to be the main contribution to the variability in overnight mixing between the RL and NBL, is compared with MDA8 observed the following day, shown in Figure 11. It can be seen that a stronger nocturnal low level jetLLJ is correlated, albeit weakly, with lower ozone the following day ($r^2$=0.181, $p < 10^{-5}$). A single outlier was removed where the LLJ exceeded 25 m s$^{-1}$. This is in line withThise overall relationship found supports our hypothesis that the low level jetLLJ will leads to stronger mixing, which in turn leads to more residual layerRL ozone depletion.

[Figure]

**Figure 11.** Correlation between nocturnal low level jet speed and the following day's MDA8 in Visalia, CA, for Calendar days 152-273 from 2010-2016.

The physical processes of RL $O_x$ depletion once it mixes down into the NBL is a further question. The main destruction processes of $O_x$ in the NBL are chemical loss and dry deposition. One possibility is that surface sources of $NO_2$ contribute to the excess nocturnal chemical depletion of $O_x$ in the NBL. However, the chemical loss of $O_x$ is not thought to vary significantly between the RL and NBL because the increase of $NO_2$ in the NBL is compensated by the decrease of $O_3$ (see Fig. 4), although this assumes that there are no other chemical differences that alter the reaction fate of nitrate (i.e. α in Eq. 1). Another possibility is that the deposition velocity of ozone may be enhanced by a reduction of aerodynamic resistance under a stronger LLJ. The dry deposition of any gas may be characterized by a series of resistances (Wesely, 1989):

$$v_d = \frac{1}{r_a + r_b + r_c} \quad (6)$$

 where $r_a$ is the aerodynamic resistance, $r_b$ is the viscous sub-layer resistance, and $r_c$ is the surface (or canopy) resistance. Figure 4 in Padro (1996) suggests that for ozone at night, $r_a \sim r_c \sim 250$ s m$^{-1}$. $r_b$ is likely non-zero (Massman et al., 1994) but is typically several times smaller than the other resistances (Georgiadis et al., 1995; Pilegaard et al., 1998), so we assume that $r_a = r_b + r_c = 250$ s m$^{-1}$ to yield our estimated deposition velocity of 0.2 cm s$^{-1}$. Combining an estimate of aerodynamic resistance due to mass transfer ($r_a = \frac{U}{u}$  where $u_*^2$ is the momentum flux) and parameterizing the momentum flux as a function of 10-meter wind speed, $U_{10}$, and a drag coefficient  $C_{D}$  ($u_*^2 = C_{D}$  $U_{10}^2$) we roughly approximate $r_a$ as:

$$r_a \sim \frac{1}{C_{D} \text{ } U_{10}} \quad (7)$$

In the 7 years of LLJ data at Visalia, the 10-meter wind speed is correlated with the jet strength ($r^2 = 0.309$, p < 10$^{-5}$). On average, $U_{10}$ was 1 m s$^{-1}$ for 5 m s$^{-1}$ jets, and 2.5 m s$^{-1}$ for 15 m s$^{-1}$ jets. Assuming an average $U_{10}$ of 1.75 m s$^{-1}$ and $r_a$ of 250 s m$^{-1}$, this would imply that $C_{D}$  $\sim 2.3 \times 10^{-3}$. A sensitivity analysis indicates that the difference U$_{10}$ between strong and weak jets would result in a approximate 40 % change in $v_d$. We thus conclude that the LLJ likely plays a significant role in modulating the dry deposition rate, where a strong jet decreases $r_a$ and thus increases $v_d$, further contributing to a loss of ozone overnight. It is important to note that what we have presented is only a rough estimate of the variability of $r_a$, and thus future studies should measure these parameters with more precision in order to better estimate the degree to which the LLJ can modulate dry deposition in the SJV. The average error of $K_z$ due to the uncertainty in $v_d$ is calculated to be ~0.50 m$^2$ s$^{-1}$, and is included in our  error

**3.5. Eddy Diffusivity and other estimates of Turbulence**

Here we attempt to build confidence in the eddy diffusivity estimates by analyzing additional metrics of turbulence. We find that nocturnally and spatially averaged TKE in the NBL ranges from 0.35 and 1.02 m$^2$ s$^{-2}$, which is very comparable to values obtained in other NBL studies (Banta et al., 2006; Lenschow et al., 1988). Table 1 shows the TKE, LLJ speed, as well as the ratio of the streamwise variance to LLJ speed ($\sigma_u/U_x$) for each night. The average value of $\sigma_u/U_x$ in this study is 0.11, approximately double what was reported in Banta et al. (2006). There is no detectable relationship between our calculated NBL

TKE and eddy diffusivities, LLJ speed, or MDA8 the following day, which implies that the eddy diffusivities calculated from the scalar budget analysis may be a better measure of nocturnal mixing strength than TKE.

[revised manuscript text omitted]

    The locations of the layers greater than 50 m thickness, along with their elevation and lapse
rate, is shown in Figure 13. One feature of note is that the layers appear to be more prominent over urban areas,
such as Fresno, Visalia, and Bakersfield. This may lead one to suspect that some of these layers are driven by an urban
heating effect, however, this seems unlikely as the unstable layers appear mostly above the NBL where there
760  communication with the surface is relatively rapid. TRather, the appearance of these
layers clustering around  urban areas may be the result of a
sampling bias and thus may not be
significant.  Another feature worth
noting is that more unstable layers are observed closer to the Tehachapi pass. One possible explanation for this is that
765 the katabatic flow down the mountain slopes detrain along the way and are carried over the valley by local advection
before mixing with surrounding air. Given that these layers are found from near the bottom of the RL
all the way up to 2.5 km, it is possible that they contribute to the overnight mixing of $O_x$ from the RL to the NBL
by maintaining a fairly well-mixed lower atmosphere over the valley. Further research, both
observational and modeling-based, is needed to explore this possibility.

[Figure]

**Figure 13.** Detected nocturnal elevated mixed layers with at least 50 meters thickness, with elevations shown.

The unstable layers are not  found to have more TKE than the rest of the atmosphere. While,  this may reflect the limitations of the method used to estimate turbulence from this low-cost wind measurement system.  it is consistent with the study by Cho et al. (2003)  that found no relationship between turbulence and static stability in the free troposphere. Interestingly, their analysis of aircraft data collected over the Pacific Ocean up to 8 km altitude found unstable layers in 6 to 25% (depending on the layer thickness definition of 100 m to 10 m) of their profiles above the boundary layer (Cho et al., 2003).  Because the aircraft  moves more than ten times  faster horizontally than  vertically during profiling,  the observations of the elevated mixed layers may be  an artifact of localized temperature gradients that are more prominent in the horizontal dimension. To  confirm that this is not the case, we  examined the wind quivers in the unstable layers along with the direction of the colder air. The cooler air was not systematically detected in any one direction, which supports the hypothesis that  they are true vertical temperature gradients.

To analyze the stability, wind shear, and turbulence from a climatological standpoint, a July-August 2016 composite of the 915 MHz Visalia sounder data is presented in Figure 14. Even in the  two month averages, some nocturnal unstable layers are detectable between 500 and 1500 m, which further supports the existence of  persistent elevated mixed layers  that may contribute to overnight mixing of pollutants in the lower troposphere over the valley.

[Figure]

**Figure 14.** Stability and wind quivers for the Visalia 915 MHz sounder, 1 Jul 2016 –- 31 Aug 2016.

**4. Conclusions**

We have demonstrated a method for performing a nocturnal $O_x$ scalar budget analysis using aircraft data, and applying estimate the effects of turbulent mixinge in the stable boundary layerNBL, which can be related used to help the SSJV. Inherently, eddy diffusivity estimates for any given night will have a large uncertainty due to the indirect nature of the measurement and the limited flight durations. However, the overall between-flight consistency and the correlations of the eddy diffusivities with both the Richardson number and surface ozone suggest that this method is informative. We obtain eddy diffusivity values between 1.1 and 3.5 $m^2$ $s^{-1}$, which are larger but approximately within the same order of magnitude of values that have been obtained from other studies in the free troposphere, lower stratosphere, and nocturnal boundary layerNBL. A One limitation of our study is the lack of sample size, with only morning flights. HoweverNevertheless, we believe this study demonstrates the importance of focused flight strategies measure the individual terms of the scalar budget equation, and highlights the significant influence that synoptic and mesoscale features atmeteorological conditions can have on the over-night within the context offor high ozone day's peak concentrationsf focused flight strategy where terms in the scalar budget equation are measured.

The larger set of RASS and ARB surface network data from Visalia, CA establishes shows a correlation between low level jetLLJ speed and the maximum 1-hour ozoneMDA8 the following afternoon for summertime months, further suggesting the a link between nocturnal mixing and the following daysnext-dayensuing day's ozone levels. In particular, we note that 5 out of 6 days when the Visalia, CA ozone MDA8 exceeded 90 ppb wereas preceded by a weak LLJ (< 7 m $s^{-1}$). Similarly, the correlations between the aircraft-estimated eddy diffusivities and MDA8 the following day also suggest that vertical mixing in the NBL plays an important role in determining controlling ozone concentrations. In particular, we note that 11 of 12 days where the Visalia, CA ozone concentration exceeded 100

ppb was preceded by a low level jet speed < 9 m/s. While we cannot unequivocally determine infer a causal reduced ozone pollutionlevels, we propose a feasible process link between that a stronger LLJ leadings to greater helps deplete the ozone reservoir in the RL by bringing it into the stable boundary layerNBL overnight. There it is atto the surface, whereinand theat dry deposition rate velocity itself may itself be partially modulated by the strength of the LLJ.. This may occur through Because the near-surface winds are accelerated by an overlying jet, a stronger LLJ reducesd the aerodynamic resistance resulting in more efficient transport to surfaces and stomata where ozone can depositbe taken up. Subsequently, when thermals begin to form after sunrise the following morning, there is less ozone to fumigate downward. We propose that the LLJ is a branch of the Fresno eddyFresno Eddy, and the vertical mixing it induces may offset some of the next-day ozone enhancement that results from the eddy recirculating pollutants. While the correlation between nocturnal mixing and ozone the following day is not always strong, it is an important link that may have consequential implications for modeling studies and policy making. For example, oOur findings highlight the crucial need of models to capture the LLJ and Fresno eddyFresno Eddy with sufficient resolution, and p. Policy makers may consider putting more stringent emission limitations on days where synoptic and mesoscale patterns appear to favor a lack of overnightweak nocturnal mixing.

Of course, theThe relative importance of these dynamical effects depends on the exact magnitude of the chemical overnight. We have also suggested that the ultimate fate of the $NO_3$ radical plays a veryn important role in the nocturnal loss term, and thus likely impacts the following day's maximum ozone concentration. The loss of the nitrate radical at night can occur from $N_2O_5$ hydrolysis, reaction with VOCs, or a very rapid reaction with small NO concentrations, and there is considerable uncertainty regarding which reactions dominate without concurrent direct measurements of $N_2O_5$, and VOCs. Thus, the lifetime of $NO_3$ can range from seconds to several minutes, which affects the chemical loss term in the scalar budget equation. It is thereforeus crucial to measure the lifetime of $NO_3$ in future studies that analyze the NBL ozone or $O_x$ budget. We also suggest more direct measurements estimates of aerodynamic resistance ozone deposition at the surface by ground-based eddy covariance flux measurements in conjunction with future airborne studies.

**Data Availability**

[revised manuscript text omitted]